# EXPLORING THE LOSS LANDSCAPE OF REGULARIZED NEURAL NETWORKS VIA CONVEX DUALITY

**Sungyoon Kim**[1]**, Aaron Mishkin**[2]**, Mert Pilanci**[1]
[1]Department of Electrical Engineering, Stanford University
[2]Department of Computer Science, Stanford University
sykim777@stanford.edu, amishkin@cs.stanford.edu,
pilanci@stanford.edu

## ABSTRACT

We discuss several aspects of the loss landscape of regularized neural networks: the structure of stationary points, connectivity of optimal solutions, path with non-increasing loss to arbitrary global optimum, and the nonuniqueness of optimal solutions, by casting the problem into an equivalent convex problem and considering its dual. Starting from two-layer neural networks with scalar output, we first characterize the solution set of the convex problem using its dual and further characterize all stationary points. With the characterization, we show that the topology of the global optima goes through a phase transition as the width of the network changes, and construct examples where the problem may have a continuum of optimal solutions. Finally, we show that the solution set characterization and connectivity results can be extended to different architectures, including two-layer vector-valued neural networks and parallel three-layer neural networks.

## 1 INTRODUCTION

Despite the nonconvex nature of neural networks, training them with local gradient methods finds nearly optimal parameters. Understanding the properties of the loss landscape is theoretically important, as it enables us to depict the learning dynamics of neural networks. For instance, many existing works prove that the loss landscape is "benign" in some sense - i.e. they don't have spurious local minima, bad valleys, or decreasing path to infinity Kawaguchi (2016), Venturi et al. (2019), Haeffele & Vidal (2017), Sun et al. (2020), Wang et al. (2021b), Liang et al. (2022). Such characterization enlightens our intuition on why these networks are trained so well.

As part of understanding the loss landscape, understanding the structure of global optimum has gained much interest. An example is mode connectivity Garipov et al. (2018), where a simple curve connects two global optima in the set of optimal parameters. Another example is analyzing the permutation symmetry that a global optimum has Simsek et al. (2021). Mathematically understanding the global optimum is important as it sheds light on the structure of the loss landscape. They can also motivate practical algorithms that search over neural networks with the same optimal cost Ainsworth et al. (2022), Mishkin & Pilanci (2023), having practical motivations to study.

We shape the loss landscape of regularized neural networks with ReLU activation, mainly analyzing mathematical properties of the global optimum, by considering its convex counterpart and leveraging the dual problem. Our work is inspired by the work of Mishkin & Pilanci (2023), where they characterize the optimal set and stationary points of a two-layer neural network with weight decay using the convex counterpart. They also introduce several important concepts such as the polytope characterization of the optimal solution set, minimal solutions, pruning a solution, and the optimal model fit. Expanding the idea of Mishkin & Pilanci (2023), we show a clear connection between the polytope characterization and the dual optimum. We further derive novel characters of the optimal set of neural networks, the loss landscape, and generalize the result to different architectures.

Finally, it is worth pointing out that regularization plays a central role in modern machine learning, including the training of large language models Andriushchenko et al. (2023). Therefore, including regularization better reflects the training procedure in practice.

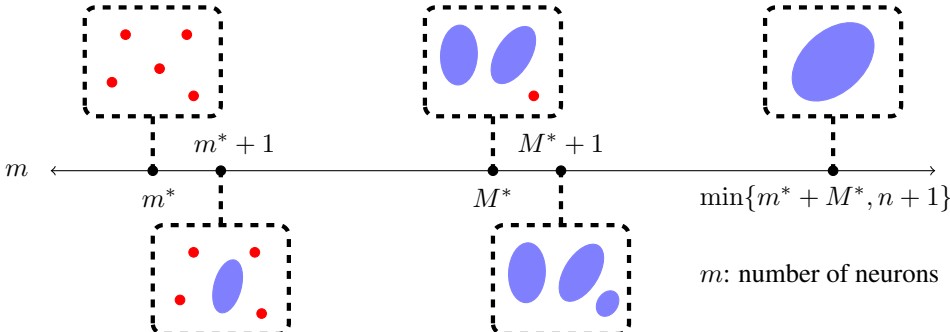

Figure 1: **A schematic that illustrates the staircase of connectivity.** This conceptual figure describes the topological change in solution sets as the number of neurons $m$ changes in a high-level manner. Connected components that are not singletons are shown as blue sets, whereas singletons are depicted as red dots. When $m = m^*$, there are only finitely many red dots. When $m \geq m^* + 1$, there exists a connected component that is not a singleton, i.e. a blue set. When $m = M^*$, there exists a connected component which is a singleton, i.e. a red dot. When $m \geq M^* + 1$, there is no red dot. At last, when $m \geq \min\{m^* + M^*, n + 1\}$, there is a single blue set.

More importantly, adding regularization can change the qualitative behavior of the loss landscape and the global optimum Wang et al. (2021b): for example, there always exist infinitely many optimal solutions for the unregularized problem with ReLU activation due to positive homogeneity. However, regularizing the parameter weights breaks this tie and we may not have infinitely many optimal solutions. It is also possible to design the regularization for the loss landscape to satisfy certain properties such as no spurious local minima Liang et al. (2022), Ge et al. (2017) or unique global optimum Mishkin & Pilanci (2023), Boursier & Flammarion (2023). Understanding the loss landscape of regularized neural networks is not only a more realistic setup but can also give novel theoretical properties that the unregularized problem does not have.

The specific findings we have for regularized neural networks are:

• **The optimal polytope:** We revisit the fact that the regularized neural network's convex reformulation has a polytope as an optimal set Mishkin & Pilanci (2023). We give a connection between the dual optimum and the polytope.

• **The staircase of connectivity:** For two-layer neural networks with scalar output, we give critical widths and phase transitional behavior of the optimal set as the width of the network $m$ changes. See Figure 1 for an abstract depiction of this phenomenon.

• **Nonunique minimum-norm interpolators:** We examine the problem in Boursier & Flammarion (2023) and show that free skip connections (i.e., an unregularized linear neuron), bias in the training problem, and unidimensional data are all necessary to guarantee the uniqueness of the minimum-norm interpolator. We construct explicit examples where the solution is not unique in each case, inspired by the dual problem. In contrast to the previous perspectives Boursier & Flammarion (2023), Joshi et al. (2023), our results imply that free skip connections may change the qualitative behavior of optimal solutions. Moreover, uniqueness does not hold in dimensions greater than one.

• **Generalizations:** We extend our results by providing a general description of solution sets of the cone-constrained group LASSO. The extensions include the existence of fixed first-layer weight directions for parallel deep neural networks, and connectivity of optimal sets for vector-valued neural networks with regularization.

The paper is organized as follows: after discussing related work (Section 1.1) and notations (Section 1.2), we discuss the convex reformulation of neural networks as a preliminary in Section 2. Then we discuss the case of two-layer neural networks with scalar output in Section 3, starting from the optimal polytope characterization (Section 3.1), the staircase of connectivity (Section 3.2), and construction of non-unique minimum-norm interpolators (Section 3.3). The possible generalizations are introduced in Section 4. Finally, we conclude the paper in Section 5. Detailed explanations of the experiments and proofs are deferred to the appendix.

## 1.1 RELATED WORK

**Convex Reformulations of Neural Networks** Starting from Pilanci & Ergen (2020), a series of works have concentrated in reformulating a neural network optimization problem to an equivalent convex problem and training neural networks to global optimality. It has been shown that many different existing neural network architectures with weight decay have such convex formulations, including vector-valued neural networks Sahiner et al. (2020), CNNs Ergen & Pilanci (2020), and parallel three-layer networks Ergen & Pilanci (2021). Furthermore, properties of the original non-convex problem such as the characterization of all Clarke stationary points Wang et al. (2021b), and the polyhedral characterization of optimal set Mishkin & Pilanci (2023) have also been discussed.

**Connectivity of optimal sets of neural networks** Mode connectivity is an empirical phenomenon where the optimal parameters of neural networks are connected by simple curves of almost similar training/test accuracy Garipov et al. (2018). An intriguing phenomenon itself, it has given rise to theoretical analysis of the connectivity of optimal solutions: to name a few, Kuditipudi et al. (2019) introduces the concept of dropout stability to explain such phenomena, Zhao et al. (2023) uses group theory to understand the connected components of deep linear neural networks, and Akhtiamov & Thomson (2023) introduces theory from differential topology to understand mode connectivity. Permutation symmetry in the parameter space also plays an important role in understanding connectivity. Simsek et al. (2021) shows that assuming a unique global minimizer modulo permutations of a certain size, increasing the size of each layer by one connects all global optima. Unfortunately, their assumption does not hold in our case (Appendix G). A similar characterization is also done in Brea et al. (2019), where saddle points with permutation symmetry are connected. Sharma et al. (2024) further discusses different notions of linear connectivity modulo permutations. A different line of work concentrates on the connection between overparametrization and connectivity of solutions: the main insight here is that when the model is as large as the number of data, the solution set becomes connected Nguyen (2021), Nguyen et al. (2021), Nguyen (2019). Cooper (2021) has a similar connection for overparametrized networks, where they characterize the dimension of the manifold of the optimal parameter space.

**Phase transitional behavior of the loss landscape** Here we introduce existing work in the literature that gives a characterization saying "adding one more neuron can change the qualitative behavior of the loss landscape", hence having the notion of critical model sizes. We reiterate Nguyen (2021) and Simsek et al. (2021), where adding one neuron changes the connectivity behavior of the optimal set. Liang et al. (2018) adds an exponential neuron, which is a specifically designed neuron, along with a specific regularization to eliminate all spurious local minimum. Venturi et al. (2019) has the idea of defining upper / lower intrinsic dimensions of the training problem in the unregularized case, and shows that the quantity is related to whether the training problem has no spurious valleys. Li et al. (2022) discusses a critical width $m^*$ where $m \geq m^*$, all suboptimal basins are eliminated for certain activation functions. They also discuss how $m^*$ is related with $n$, the number of data.

**Loss landscapes and optimal sets of regularized networks** Freeman & Bruna (2016) discusses the loss landscape of the population loss along with a certain regularization, and proves the asymptotic connectivity of all sublevel sets as $m$ increases. Bietti et al. (2022) also introduces an asymptotic landscape result for regularized networks. Haeffele & Vidal (2017) deduces the loss landscape of parallel neural networks with the lens of convex equivalent problem, and shows that when the width $m$ is larger than a certain threshold, there are no spurious local minima. Kunin et al. (2019) analyzes regularized linear autoencoders and points out the discrete structure of critical points under some symmetries. Bucarelli et al. (2024) bounds the Betti number of the sublevel set of the loss landscape for Pfaffian activations, discussing topological complexity of sublevel sets for both the unregularized and the regularized case. On the empirical side, Yang et al. (2021) considers certain metrics to consider the mode connectivity and sharpness of the landscape of regularized neural networks, and indeed show that larger models tend to have more connected solutions. A few work design specific regularization to make the loss landscape benign, removing spurious local minima and decreasing paths to infinity Ge et al. (2017), Liang et al. (2022).

**Properties of unidimensional minimum-norm interpolators** Training minimum-norm interpolators for unidimensional data can lead to sparse interpolators Parhi & Nowak (2023). When we do not penalize the bias, Savarese et al. (2019) has an exact characterization of the interpolation problem in function space, and Hanin (2021) completely characterizes the set of optimal interpolators. From the construction of optimal interpolators, it is natural that there exist problems with a continuum of

infinitely many optimal interpolators. A recent work by Nakhleh et al. (2024) extends this setup to vector-valued networks and shows almost-sure uniqueness of a minimum norm interpolator. On the other hand, Boursier & Flammarion (2023) recently showed that when we penalize the bias with free skip connections, we have a unique optimal interpolator. Furthermore, under certain assumptions on the training data, the optimal interpolator is the sparsest. Empirically, it has been believed that having a free skip connection does not affect the behavior of the solution Boursier & Flammarion (2023), Joshi et al. (2023).

## 1.2 PROBLEM SETTING AND NOTATIONS

We are interested in training a neural network with regularization and ReLU activation, namely the optimization problem

$$\min_{\theta \in \mathbb{R}^p} L(f_\theta(X), y) + \beta \mathcal{R}(\theta). \tag{1}$$

Here, $X \in \mathbb{R}^{n \times d}$ is the data matrix, $y \in \mathbb{R}^n$ is the label vector, $\theta \in \mathbb{R}^p$ the concatenation of all parameters of the neural network, $f_\theta$ the parametrization, $\beta > 0$ strength of the regularization, $L : \mathbb{R}^n \times \mathbb{R}^n \to \mathbb{R}$ the convex loss function, and $\mathcal{R} : \mathbb{R}^p \to \mathbb{R}$ the regularization.

We have two different objects of interest in the notion of optimal sets: the optimal solution set in parameter space and the set of optimal functions

$$\Theta^* := \arg\min_{\theta \in \mathbb{R}^p} L(f_\theta(X), y) + \beta \mathcal{R}(\theta) \subseteq \mathbb{R}^p, \quad \mathcal{F}^* := \{f_\theta \mid \theta \in \Theta^*\} \subseteq \mathcal{F}, \tag{2}$$

where $\mathcal{F}$ is the set of functions $f : \mathbb{R}^d \to \mathbb{R}$. The notion of optimal functions will mostly be discussed in Section 3.3, where we discuss minimum-norm interpolators. Note that $\Theta^*$ regards parameters with permutation symmetry as different parameters.

Next, we clarify the notion of connectivity in this paper. We say two points $x, y \in S$ is connected in $S$ if for two points $x, y \in S$, there exists a continuous function $f : [0, 1] \to S$ that satisfies $f(0) = x, f(1) = y$. We say $S$ is connected if for any two points $x, y \in S$, $x$ and $y$ are connected in $S$. Also, an isolated point $x$ in $S$ means a point that has no continuous path from $x$ to $S - \{x\}$.

At last, we clarify the notations. the notation $1(\text{condition}(A))$ is defined for a scalar, vector, or matrix that notes if the entrywise condition is met, the value is 1, and else 0. Note $[m] = \{1, 2, \cdots, m\}$, $\|\cdot\|_2$ as the $l_2$ norm, $\|\cdot\|_F$ as the Frobenious norm, $(\cdot)_+$ as the ReLU function, and $\text{diag}$ the diagonal matrix given a vector. By a hyperplane arrangement, we mean a diagonal matrix $\text{diag}(1(Xh \geq 0))$ for a vector $h \in \mathbb{R}^d$. When we write $D_i$ for $i \in [P]$, we mean all possible hyperplane arrangements generated from the data matrix $X \in \mathbb{R}^{n \times d}$, hence $P$ means the number of all possible arrangement patterns. We also use the notation $\mathcal{K}_i = \{u \mid (2D_i - I)Xu \geq 0\}$ for $i \in [P]$ unless specified differently (in vector-valued networks we will). By $a \oplus b$, we mean the concatenation of two vectors(or matrices) $a$ and $b$: if $a \in \mathbb{R}^m$ and $b \in \mathbb{R}^n$, $a \oplus b \in \mathbb{R}^{m+n}$, $(a_i)_{i=1}^p$ denotes $a_1 \oplus a_2 \oplus \cdots a_p$. For matrices, the notation $A_i$. means the $i$-th row of $A$, $A_{\cdot i}$ means the $i$-th column of $A$, and for vector $v$, $v_{,k}$ denotes the $k$-th entry of $v$. We note the matrix inner product $\langle A, B \rangle_M = tr(A^T B)$.

## 2 CONVEX REFORMULATIONS

Our main proof strategy will be introducing an equivalent convex reformulation of the training problem first introduced in Pilanci & Ergen (2020). In this section, we demonstrate the concept by giving an example for two-layer scalar output networks with weight decay.

Consider the optimization problem in equation 3,

$$p^* := \min_{\{w_j, \alpha_j\}_{j=1}^m} L\left(\sum_{j=1}^m (Xw_j)_+ \alpha_j, y\right) + \frac{\beta}{2} \sum_{j=1}^m \left(\|w_j\|_2^2 + \alpha_j^2\right). \tag{3}$$

The variables $w_j \in \mathbb{R}^d, \alpha_j \in \mathbb{R}$ for $j \in [m]$. When the width $m$ of problem in equation 3 satisfies $m \geq m^*$ for a critical threshold $m^* \leq n$, we have an equivalent convex problem given as a cone-constrained group LASSO,

$$p_{\text{cvx}}^* := \min_{\{u_i, v_i\}_{i=1}^P, \ u_i, v_i \in \mathcal{K}_i} L\left(\sum_{i=1}^P D_i X(u_i - v_i), y\right) + \beta \sum_{i=1}^P (\|u_i\|_2 + \|v_i\|_2). \tag{4}$$

The intuition of convexification is constraining each variable at a certain convex cone so that the model looks linear in that region, and applying an appropriate scaling to deal with regularization.

As an equivalent convex problem, the optimal values $p^*$ and $p^*_{\text{cvx}}$ are equal. Moreover, from a solution $(u_i, v_i)^P_{i=1}$ of equation 4 satisfying $m = \sum^P_{i=1} 1(u_i \neq 0) + 1(v_i \neq 0)$, we can recover the solution of equation 3 with $m$ neurons by a solution mapping $(w_i, \alpha_i) = (u_i/\sqrt{\|u_i\|_2}, \sqrt{\|u_i\|_2})$ for $i \in [a]$, $(w_{i+a}, \alpha_{i+a}) = (v_i/\sqrt{\|v_i\|_2}, -\sqrt{\|v_i\|_2})$ for $i \in [m - a]$, without loss of generality assuming $u_i \neq 0$ for $i \in [a]$ and $v_i \neq 0$ for $i \in [m - a]$.

The problem in equation 3 has a convex dual given as

$$d^* := \max_{|\nu^T(Xu)_+| \leq \beta, \, \forall \|u\|_2 \leq 1} -L^*(\nu), \tag{5}$$

where $L^*$ is the convex conjugate of $L(\cdot, y)$ and $\nu$ denotes the dual variable. Note that strong duality holds and $p^* = p^*_{\text{cvx}} = d^*$ is satisfied when $m \geq m^*$. Furthermore, we will see that the dual optimum $\nu^*$ determines the optimal set of both the convex problem in equation 4 and the original problem in equation 3.

## 3 TWO-LAYER SCALAR OUTPUT NEURAL NETWORKS

### 3.1 THE OPTIMAL POLYTOPE

We first describe the optimal set of the problem in equation 4 where $L$ is strictly convex. Note that the polytope characterization was first done in Mishkin & Pilanci (2023). Here, we emphasize the role of dual optimum in choosing the unique directions. To illustrate the solution set of equation 4, we introduce the notion of an optimal model fit and further characterize it as a singleton.

**Proposition 1.** *Mishkin & Pilanci (2023) Let the optimal solution set of equation 4 as $\Theta^*$. If the loss function L is strictly convex, the optimal model fit is unique, i.e. the set of optimal model fit*

$$\mathcal{C}_y = \left\{ \sum^P_{i=1} D_i X(u^*_i - v^*_i) \mid (u^*_i, v^*_i)^P_{i=1} \in \Theta^* \right\} = \{y^*\} \text{ for some } y^* \in \mathbb{R}^n.$$

The solution set of equation 4 is given as Theorem 1. For a formal statement see Theorem C.1.

**Theorem 1.** *(The Optimal Polytope, informal) Suppose L is a strictly convex loss function. The directions of optimal parameters of the problem in equation 4, noted as $\bar{u}_i, \bar{v}_i$, are uniquely determined from the dual optimum $\nu^*$. Moreover, the solution set of equation 4 is the polytope,*

$$\mathcal{P}^*_{\nu^*} := \left\{ (c_i\bar{u}_i, d_i\bar{v}_i)^P_{i=1} \mid c_i, d_i \geq 0 \quad \forall i \in [P], \quad \sum^P_{i=1} D_i X \bar{u}_i c_i - D_i X \bar{v}_i d_i = y^* \right\} \subseteq \mathbb{R}^{2dP},$$
$$\tag{6}$$

*for the unique optimal model fit $y^*$ defined in Proposition 1.*

Note that $\mathcal{P}^*_{\nu^*}$ is invariant under different choices of $\nu^*$, because they all correspond to the solution set of equation 4. Hence, we use $\mathcal{P}^*$ for simplicity. For a geometric intuition of $\nu^*$, see Appendix G.

Theorem 1 implies that equation 4 has a unique direction for each $u_i, v_i$ where $i \in [P]$, which is determined by solving the dual problem. The intuition for this fact is quite clear: when we assume there exist two different solutions $(u_i, v_i)^P_{i=1}$ and $(u'_i, v'_i)^P_{i=1}$ where $u_i$ and $u'_i$ are not colinear for some $i \in [P]$, $((u_i + u'_i)/2, (v_i + v'_i)/2)^P_{i=1}$ has a strictly smaller objective because $L$ is strictly convex and $\|a\|_2 + \|b\|_2 \geq \|a + b\|_2$ with equality only when $a$ and $b$ are colinear. However, Theorem 1 implies further, that for any conic combination of such vectors $D_i X \bar{u}_i$ and $-D_i X \bar{v}_i$ that sum up to $y^*$, it becomes an optimal solution of equation 4.

Another implication of Theorem 1 is that for all stationary points of equation 3, there exists a finite set of possible first-layer weight directions. For a formal statement see Corollary C.1.

**Corollary 1.** *Denote the set of Clarke stationary points of equation 3 as $\Theta_C$. The set of directions of the stationary point $\bigcup^m_{j=1} \left\{ w_j/\|w_j\|_2 \mid (w_i, \alpha_i)^m_{i=1} \in \Theta_C, \, w_j \neq 0 \right\}$ is finite, and is determined by the dual optimum of subsampled convex problems.*

The result follows from using the fact proven in Ergen & Pilanci (2023), where all stationary points of equation 3 are characterized by the global minimizer of the subsampled convex program that has the same structure with equation 4. The implication shows that not only the global minimum but the stationary points of equation 3 also have a structure that is related to the convex problem.

## 3.2 THE STAIRCASE OF CONNECTIVITY

One significance of this characterization is that when $m \geq m^*$, we can relate the optimal solution set of the nonconvex problem in equation 3 with the subsets of equation 6 with certain cardinality constraints. Specifically, the cardinality-constrained set

$$\mathcal{P}^*(m) := \left\{ (u_i, v_i)_{i=1}^P \mid (u_i, v_i)_{i=1}^P \in \mathcal{P}^*, \sum_{i=1}^P \mathbb{1}(u_i \neq 0) + \mathbb{1}(v_i \neq 0) \leq m \right\} \subseteq \mathbb{R}^{2dP}, \quad (7)$$

will determine the solution set of equation 3 , namely $\Theta^*(m)$, when $m \geq m^*$. We write $\Theta^*(m)$ to emphasize the dependency of $m$, since we illustrate a phase-transitional behavior as $m$ changes. For a formal definition of $\Theta^*(m)$ see Appendix D. The cardinality constraint is the main reason behind the staircase of connectivity: if $m$ were to be unbounded, the optimal set would be a single connected polytope. However, as $m$ becomes smaller, certain regions in the polytope are not reachable, possibly becoming disconnected.

Our proof strategy is first observing phase transitional behaviors in the cardinality-constrained set $\mathcal{P}^*(m)$, and linking the connectivity behavior of $\mathcal{P}^*(m)$ and $\Theta^*(m)$ with appropriate solution mappings (Definition D.7, Definition D.8). Aside from the proof of Theorem 2, the machinery we develop can potentially be applied to extend other topological properties of $\mathcal{P}^*(m)$ to $\Theta^*(m)$.

Theorem 2 states the staircase of connectivity informally. For a formal statement and a precise definition of critical widths see Theorem D.2. Note that we get rid of the trivial case where $\mathcal{P}^* = \{(0,0)_{i=1}^P\}$ by assuming $(w_i, \alpha_i)_{i=1}^m \neq (0,0)_{i=1}^m \in \Theta^*(m)$ exists for some $m$ (Proposition D.1).

**Theorem 2.** *(The staircase of connectivity, informal) Denote the optimal solution set of equation 3 in parameter space as $\Theta^*(m) \subseteq \mathbb{R}^{(d+1)m}$. Suppose $L$ is a strictly convex loss function and there exists $(w_i, \alpha_i)_{i=1}^m \neq (0,0)_{i=1}^m \in \Theta^*(m)$ for some $m$. We have critical widths $m^*, M^*$ that determine the phase transitional behavior of the solution set. Specifically, as $m$ changes, we have that when*

*(i) $m = m^*$, $\Theta^*(m)$ is a finite set. Hence, all solutions are disconnected to each other.*

*(ii) $m \geq m^* + 1$, there exists $A \neq A' \in \Theta^*(m)$ and a path in $\Theta^*(m)$ connecting them.*

*(iii) $m = M^*$, $\Theta^*(m)$ is not a connected set. Moreover, there exists an isolated point in $\Theta^*(m)$.*

*(iv) $m \geq M^* + 1$, permutations of the solution are connected with no isolated points in $\Theta^*(m)$.*

*(v) $m \geq \min\{m^* + M^*, n+1\}$, the set $\Theta^*(m)$ is connected.*

Figure 1 demonstrates Theorem 2 at a conceptual level. When $m = m^*$, the solution set has a discrete structure. One way to see the fact is that when $m = m^*$, the solutions are vertices of the polytope $\mathcal{P}^*$, hence they have a discrete and isolated structure. When $m \geq m^* + 1$, we have a trivial "splitting" operation that connects two solutions with $m^*$ nonzero first-layer weights, which leads to the existence of a "blue set"(a connected component with infinitely many solutions) in Figure 1. When $m = M^*$, the solution having linearly independent first-layer weights with maximum cardinality corresponds to the isolated point in $\Theta^*(M^*)$. When $m \geq M^* + 1$, on the other hand, any solution is connected with permutations of the same solution. The proof follows from first creating a zero slot in the first layer weights and using the zero slot to permute. The idea of the proof is identical to that of Simsek et al. (2021), though the details differ. At last, when $m \geq \min\{m^* + M^*, n+1\}$, the whole set is connected: $m^* + M^*$ is obtained from first transforming the solution to have linearly independent first-layer weights and interpolating the solution with minimum cardinality. $n+1$ follows from the fact that $\mathcal{P}^*(n+1)$ is connected, which needs a more sophisticated argument. For details see the proof in Appendix D. Note that there exists algorithms that can exactly compute these critical widths Remark D.1.

From Haeffele & Vidal (2017), we know that when $m \geq n + 1$ we have that all local minima are global (Theorem 2, Haeffele & Vidal (2017)) and moreover we have a path with non-increasing

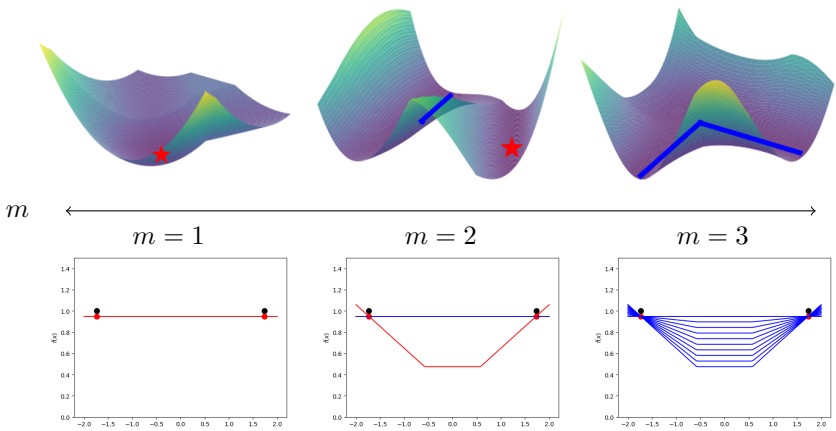

Figure 2: **Staircase of connectivity for a toy example.** The figures above the horizontal line show the toy problem's loss landscape as the width $m$ changes. The red star denotes a single optimal solution while the blue line denotes a continuum of optimal solutions. The figures below the horizontal line show the corresponding optimal functions. The red/blue functions correspond to the functions parametrized by the red/blue sets in the loss landscape. Note that when $m = 3 = \min\{m^* + M^*, n + 1\}$, there exists a continuous deformation from one solution to another.

objective to a global optimum starting from any point Vidal et al. (2022). With Theorem 2-(v) and this fact, we can construct path with non-increasing objective from any point to any global minimum. The path is clear: starting from any point, use the path in Vidal et al. (2022). Then use Theorem 2-(v) to move to any global minimum.

**Corollary 2.** *Consider the problem in equation 3 with $m \geq n + 1$. For any parameter $\theta = (w_i, \alpha_i)_{i=1}^m$, there is a continuous path from $\theta$ to any global minimizer $\theta^*$ with nonincreasing loss.*

Moreover, Corollary 2 implies the connectivity of all sublevel sets when $m \geq n + 1$. This extends the result of Nguyen (2019) to regularized networks. For a more formal statement see Corollary D.2.

**Corollary 3.** *(Informal) Consider the problem in equation 3 with $m \geq n + 1$. All sublevel sets of the loss function is connected.*

Example 1 demonstrates Theorem 2 for a toy optimization problem, which is solving a regularized regression problem with two data points.

**Example 1.** *(Demonstrating the staircase of connectivity for a toy example) Consider the dataset $\{(x_i, y_i)\}_{i=1}^2 = \{(-\sqrt{3}, 1), (\sqrt{3}, 1)\}$ and the regularized regression problem with bias*

$$\min_{\{(\theta_i, a_i, b_i)\}_{i=1}^m} \frac{1}{2} \sum_{j=1}^2 \left( \sum_{i=1}^m (a_i x_j + b_i)_+ \theta_i - y_j \right)^2 + \frac{\beta}{2} \sum_{i=1}^m (\theta_i^2 + a_i^2 + b_i^2).$$

*In Figure 2, we plot the loss landscape and the corresponding optimal functions when $\beta = 0.1$ for $m = 1, 2, 3$.*

*The upper half of Figure 2 illustrates how the loss landscape looks near the global minima, and visualizes the optimal solution set for $m = 1, 2, 3$. The lower half of Figure 2 shows the optimal learned function for $m = 1, 2, 3$. The black dots are the datapoints, the red dots correspond to the optimal model fit $y^*$, and the red/blue functions correspond to the functions parametrized by the red/blue sets in the loss landscape, respectively.*

*When $m = 2$, two different functions are shown. This is because the connected component with infinitely many solutions emerges from the split of a single neuron corresponding to the same optimal function in $\mathcal{F}^*$. When $m = 3$, we have a sequence of functions that continuously deform from one to another with the same cost. For details on the solution set of the training problem, parameterization of optimal functions, and how the loss landscape is visualized, see Appendix A.*

### 3.3 NON-UNIQUE OPTIMAL INTERPOLATORS

In this section, we will see how the dual problem can be used to construct specific problem instances that have non-unique interpolators. There are three different setups of interest. First, the minimum-norm interpolation problem with free skip connection and regularized bias (where we denote as **SB** (Skip connection; Bias)) refers to the problem in equation 8, namely

$$\min_{m,\{a_i,b_i\theta_i\}_{i=0}^m} \sum_{i=1}^m \|a_i\|_2^2 + b_i^2 + \theta_i^2, \quad \text{subject to} \quad Xa_0 + b_0 1 + \sum_{i=1}^m (Xa_i + b_i 1)_+ \theta_i = y. \quad (8)$$

The parameters satisfy $a_i \in \mathbb{R}^d$ and $b_i, \theta_i \in \mathbb{R}$ for $i \in [m] \cup \{0\}$, and $1 \in \mathbb{R}^n$ is a vector of ones. The term "free skip connection" arises, as we have a skip connection, i.e. the linear neuron $a_0$, that is not regularized. Next, we discuss the minimum-norm interpolation problem without free skip connections and regularized bias (**NSB**: No-Skip; Bias), which is the training problem in equation 8 with an additional constraint $a_0 = b_0 = 0$. At last we study the minimum-norm interpolation problem with free skip connections but without bias (**SNB**: Skip; No-Bias), which is the problem in equation 8 with $b_i = 0$ for all $i \in [m] \cup \{0\}$. Also, note that the width $m$ is also optimized.

In Boursier & Flammarion (2023), it was proven that for unidimensional data, i.e. when $d = 1$, the set of all optimal functions $\mathcal{F}^*$ of equation 8 is a singleton. When $d > 1$, it is not the case. This fact implies that to extend Boursier & Flammarion (2023) to higher dimensions, we may need additional structures besides free skip connections.

**Proposition 2.** *When $X \in \mathbb{R}^{n \times 2}, y \in \mathbb{R}^n$, we have a dataset $(X, y)$ that has non-unique minimum-norm interpolator both for the **SB** and **SNB** problem in equation 8.*

When we have no free skip connections, i.e. the case of **NSB**, for $d = 1$ we have a class of data that has infinitely many optimal interpolators. The construction follows from making the dual problem $\max_{\|u\|\leq 1} |\nu^T(Xu)_+|$ have linearly dependent solutions by forcing $n + 1$ optimal solutions. For a rigorous construction of $(X, y)$, see Proposition E.4.

**Proposition 3.** *(A class of training problems with infinitely many optimal interpolators, informal) Consider the **NSB** problem in equation 8 with $d = 1$. For all $n \geq 2$, we can construct infinitely many different datasets $(X, y)$ having infinitely many minimum-norm interpolators.*

In the following example, we give a geometric description of finding the dataset $(X, y)$ and the continuum of optimal interpolators for $n = 5$.

**Example 2.** *(Example of a class of non-unique optimal interpolators) Consider the figure in Figure 3a. Following the arrow, we can find $s_n, s_{n-1}, \cdots s_1$ defined in Proposition E.4, and can find that each $s_i$ has norm 1, $s_n = [0, 1]^T$ and $v_n = [\sqrt{3}/2, 1/2]^T$. For the example in Figure 3a, the data $x$ and $y$ can be chosen as*

$$x = [\tan\frac{\pi}{3}, -\tan\frac{5\pi}{24}, -\tan\frac{7\pi}{24}, -\tan\frac{9\pi}{24}, -\tan\frac{11\pi}{24}]^T, \quad y \simeq [94, 29, 24, 20, 20]^T.$$

*Figure 3b show the continuum of optimal interpolators. We can see that there are infinitely many interpolators with the same cost. A magnification of the range $x \in [-8, 0]$ is given to emphasize that the interpolators are indeed different. We can also see from Figure 3c that gradient descent learns the continuum of optimal interpolators. Here we set $m = 10$. For details on the formula of optimal interpolators, see Appendix B.*

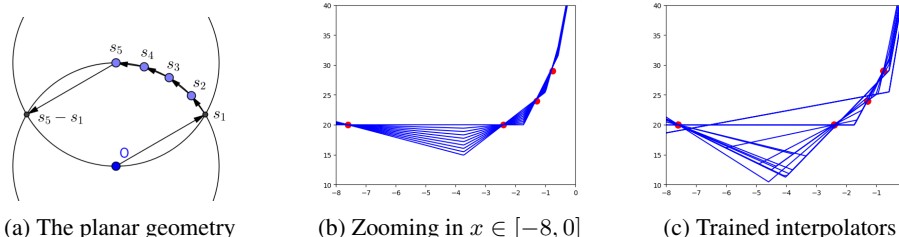

|  |  |  |
|:--:|:--:|:--:|
| (a) The planar geometry | (b) Zooming in $x \in [-8, 0]$ | (c) Trained interpolators |

Figure 3: **A demonstration of non-unique interpolators for n = 5.** Figure 3a shows the geometric construction behind finding $v$ s proposed in Proposition 3. Figure 3b shows the continuum of optimal interpolators, and Figure 3c shows the learned interpolators trained by gradient descent.

Proposition 3 demonstrates that understanding the optimal solution set with dual optimum enables us to enforce non-uniqueness to the solution set. Moreover, these examples are not constructed case-by-case, but from a geometric structure that is motivated by the object $\mathcal{Q}_X = \{(Xu)_+ \mid \|u\|_2 \leq 1\}$, the convex set $\text{Conv}(\mathcal{Q}_X \cup -\mathcal{Q}_X)$, and its supporting hyperplane.

Experimentally, the existence of free skip connections does not seem important in the behavior of the solution Boursier & Flammarion (2023), Joshi et al. (2023). However, note that when there is no skip connection, there exists training problems where the minimum-norm interpolator has infinitely many solutions. Furthermore the interpolators in Example 1 and Example 2 may have $n$ breakpoints even with Assumption 1 in Boursier & Flammarion (2023) - which can never be the sparsest interpolator. Hence, at least theoretically, free skip connection plays a significant role in guaranteeing the uniqueness and sparsity of the interpolator, along with penalizing the bias. Note that these different interpolators may have drastically different behavior for points not in the training set. For example, as $x \to \pm\infty$, the difference between any two different interpolators diverge.

## 4    GENERALIZATIONS

In this section, we will extend our results from Section 3 to a more general training setup. We use the fact that for networks of sufficiently large width, training a neural network can be cast as a cone-constrained group LASSO problem Mishkin & Pilanci (2023). Analogous to Theorem 1, we first derive the optimal set of a general cone-constrained group LASSO in equation 9:

$$\min_{\theta_i \in \mathcal{C}_i \cap \mathcal{V}_i, s_i \in \mathcal{D}_i} L\left(\sum_{i=1}^{P} A_i \theta_i + \sum_{i=1}^{Q} B_i s_i, y\right) + \beta \sum_{i=1}^{P} \mathcal{R}_i(\theta_i). \tag{9}$$

Here, $A_i, B_i \in \mathbb{R}^{n \times d}$, $\theta_i, s_i \in \mathbb{R}^d$, $y \in \mathbb{R}^n$, $\mathcal{C}_i, \mathcal{D}_i$ are proper cones, $\mathcal{R}_i$ is the regularization (which we assume to be a norm defined in a subspace $\mathcal{V}_i \subseteq \mathbb{R}^d$ and satisfy $\mathcal{V}_i \cap \mathcal{C}_i \neq \emptyset$), $\beta > 0$ is the regularization strength, and $L : \mathbb{R}^n \times \mathbb{R}^n \to \mathbb{R}$ is a convex but not necessarily strictly convex loss function. The assumption that $\mathcal{R}_i$ is a norm is natural because it will help the training problem find a simpler solution. Equation (9) enables the analysis of many different training setups, including two-layer networks with free skip connections, interpolation, vector-valued outputs, and parallel deep networks of depth 3, extending the results in Section 3.

### 4.1    DESCRIPTION OF THE MINIMUM-NORM SOLUTION SET

The idea to derive the optimal set of equation 9 is essentially the same as deriving the optimal set of equation 4: we consider the dual problem and use strong duality to obtain the wanted result. The exact description of the optimal set is given as

$$\mathcal{P}_{\text{gen}}^* = \left\{ (c_i \bar{\theta}_i)_{i=1}^{P} \oplus (s_i)_{i=1}^{Q} \mid c_i \geq 0, \sum_{i=1}^{P} c_i A_i \bar{\theta}_i + \sum_{i=1}^{Q} B_i s_i \in \mathcal{C}_y, \bar{\theta}_i \in \bar{\Theta}_i, \langle B_i^T \nu^*, s_i \rangle = 0, s_i \in \mathcal{D}_i \right\}. \tag{10}$$

First, $\mathcal{C}_y$ is the set of optimal model fits which was defined at Proposition 1. We have that $\bar{\theta}_i$ is contained in a certain set, which is an analogy of the optimal polytope in the direction of each variable is fixed. Theorem 1 is a special case where the set of optimal directions is a singleton. Finally, we have a constraint given to variables without regularization, which is also derived from the dual formulation. For a detailed derivation see Theorem F.1

Given the expression in equation 10, we can extend our results in Theorem 1 and Theorem 2 directly to the interpolation problem (Proposition F.1, Proposition F.2) . That is because for the interpolation problem, $\mathcal{C}_y$ is a singleton and the set $\bar{\Theta}_i$ is also a singleton. We can also find the optimal set characterization of the interpolation problem with free skip connections (Proposition F.3). Here, the dual variable has to satisfy $\langle X^T \nu^*, s \rangle = 0$ for all $s \in \mathbb{R}^d$, meaning $X^T \nu^* = 0$ is given as an additional constraint for the dual problem. The additional constraint is the main reason why we have qualitatively different behavior in uniqueness when we have free skip connections.

## 4.2 VECTOR-VALUED NETWORKS

Now we turn to two-layer neural networks with vector-valued outputs, namely the problem

$$\min_{(w_i,z_i)_{i=1}^m} \frac{1}{2}\|\sum_{i=1}^m (Xw_i)_+ z_i^T - Y\|_F^2 + \frac{\beta}{2}\Big(\sum_{i=1}^m \|w_i\|_2^2 + \|z_i\|_2^2\Big), \tag{11}$$

where $w_i \in \mathbb{R}^{d \times 1}$, $z_i \in \mathbb{R}^{c \times 1}$ and $Y \in \mathbb{R}^{n \times c}$. The vector-valued problem is known to have a convex reformulation Sahiner et al. (2020), which is given as

$$\min_{V_i} \frac{1}{2}\|\sum_{i=1}^P D_i X V_i - Y\|_2^2 + \beta \sum_{i=1}^P \|V_i\|_{\mathcal{K}_i,*}.$$

The norm $\|V\|_{\mathcal{K}_i,*}$ is defined as $\|V\|_{\mathcal{K}_i,*} := \min t \ge 0$ s.t. $V \in t\mathcal{K}_i$, for $\mathcal{K}_i = \mathrm{conv}\{ug^T \,|\, (2D_i - I)Xu \ge 0, \|ug^T\|_* \le 1\}$, and $\mathcal{V}_i = \mathrm{span}\{ug^T | (2D_i - I)Xu \ge 0, g \in \mathbb{R}^c\}$.

The problem falls in our category of equation 9, and we can describe the optimal set of the problem completely. With appropriate solution maps, we can describe the optimal set of the nonconvex vector-valued problem in equation 11 (Proposition F.5), and the same idea can be applied to describe a subset of the optimal solution set for deep networks (Theorem F.2). A direct implication extends the loss landscape result in Corollary 2 to vector-valued networks.

**Corollary 4.** *Consider the problem in equation 11 with $m \ge nc + 1$. For any $\theta := (w_i, z_i)_{i=1}^m \in \mathbb{R}^{(d+c)m}$, there exists a continuous path from $\theta$ to any global optimum $\theta^*$ with nonincreasing loss.*

## 4.3 PARALLEL DEEP NEURAL NETWORKS

Finally, we extend the characterization to deeper networks. Convex reformulations of parallel three-layer neural networks have been discussed Ergen & Pilanci (2021), Wang et al. (2021a). The specific training problem we are interested in is

$$\min_{m,\{W_{1i},w_{2i},\alpha_i\}_{i=1}^m} \frac{1}{3}\left(\sum_{i=1}^m \|W_{1i}\|_F^3 + \|w_{2i}\|_2^3 + |\alpha_i|^3\right) \text{ s.t. } \sum_{i=1}^m ((XW_{1i})_+ w_{2i})_+ \alpha_i = y. \tag{12}$$

The size of each weights are $W_{1i} \in \mathbb{R}^{d \times m_1}, w_{2i} \in \mathbb{R}^{m_1}$, and $\alpha_i \in \mathbb{R}$. The dual problem of the convex reformulation can be understood as optimizing a linear function with cone and norm constraints, and we have analogous results of the optimal polytope. Specifically, we have the direction of the columns of first-layer weights as a set of finite vectors (Theorem 3). The result suggests that our results are fairly generic, and could be generalized to other deep parallel architecture with appropriate parametrization. For detailed proof see Appendix F.

**Theorem 3.** *Consider the training problem in equation 12. Then, there are only finitely many possible values of the direction of the columns of $W_{1i}^*$. Moreover, the directions are determined by solving the dual problem $\max_{\|W_1\|_F \le 1, \|w_2\|_2 \le 1} |(\nu^*)^T ((XW_1)_+ w_2)_+|$ when $y \ne 0$.*

## 5 CONCLUSION

In this paper, we present an in-depth exploration of the loss landscape and the solution set of regularized neural networks. We start with a two-layer scalar neural network as the simplest case and demonstrate the properties of the set, including the existence of optimal directions, phase transition in connectivity, and non-uniqueness of minimum-norm interpolators. Then, we give a more general description on the optimal set of cone-constrained group LASSO and extend the previous results to a more general setup.

Our paper may be extended in multiple ways. One interesting problem that is left is, what is the right architecture to ensure the uniqueness of the minimum-norm interpolator for high dimensions, as free skip connection itself does not help. Another interesting problem is showing 'almost sure uniqueness' of problem equation 3, up to permutations: intuitively, from the examples we show, it can be speculated that the solution of the dual problem $\max |\nu^T (Xu)_+|$ subject to $\|u\| \le 1$ is unlikely to have "too many" optimal solutions. Hence it is likely that the dataset that makes the minimum-norm interpolator non-unique is very small. We conjecture that the set will have measure 0 in $\mathbb{R}^{2n}$, and leave it for future work. At last, extending the optimal polytope/connectivity results to tree neural networks Zeger et al. (2024) with arbitrary depth could be a meaningful contribution.

## REPRODUCIBILITY STATEMENT

The only randomness that occurs from our experiments are Figure 5a, Figure 5b, Figure 6a, and Figure 6b, where different initialization may lead to different learned interpolators. We set random seeds properly to make all results reproducible. We used a laptop to do the experiments, and provided the code to generate the figures. Code available at https://github.com/pilancilab/Loss-landscape-convex-duality

## ACKNOWLEDGEMENTS

This work was supported in part by the National Science Foundation (NSF) under CAREER award CCF-2236829, in part by the U.S. Army Research Office Early Career Award W911NF-21-1-0242, and in part by the Office of Naval Research under Grant N00014-24-1-2164.

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

APPENDIX

## A    DETAILS ON THE TOY EXAMPLE IN FIGURE 2

In this section, we give details of the toy example in Figure 2. Specifically, we illustrate how the loss landscape is plotted, how the set of optimal solutions is derived, and present the models that are found by gradient descent.

**One important remark is that "the figure does not directly imply the staircase of connectivity"** - the fact that two optimal solutions are disconnected in the visualization does not mean disconnectedness in the optimal solution, and vice versa. The figures are for the illustration of the phenomenon, not the proof.

The optimization problem that we consider is

$$\min_{\{(\theta_i, a_i, b_i)\}_{i=1}^m} \frac{1}{2} \sum_{j=1}^2 \left( \sum_{i=1}^m (a_i x_j + b_i)_+ \theta_i - y_j \right)^2 + \frac{\beta}{2} \sum_{i=1}^m (\theta_i^2 + a_i^2 + b_i^2),$$

where $\{(x_i, y_i)\}_{i=1}^2 = \{(-\sqrt{3}, 1), (\sqrt{3}, 1)\}$ and $\beta = 0.1$. When we write $X = \begin{bmatrix} -\sqrt{3}, 1 \\ \sqrt{3}, 1 \end{bmatrix} \in \mathbb{R}^{2 \times 2}$, $y = [1, 1]^T$, and the first layer weights as $U \in \mathbb{R}^{2 \times m}$, second layer weights $v = \mathbb{R}^m$, the optimization problem can also be written as

$$\min_{U \in \mathbb{R}^{2 \times m}, v \in \mathbb{R}^m} \frac{1}{2} \|(XU)_+ v - y\|_2^2 + \frac{\beta}{2} (\|U\|_F^2 + \|v\|_2^2).$$

Let the objective be $L(U, v)$. Note that even when $m = 1$, there are three parameters, so it is impossible to plot the loss landscape in a three-dimensional plot. What we do is plot a certain section of the loss landscape, as done in Li et al. (2018), to demonstrate our result.

When $m = 1$, where $r = \sqrt{1 - 0.5\beta}$, we plot

$$F(t, s) = L(\begin{bmatrix} t \\ s \end{bmatrix}, [r]),$$

for $(t, s) \in [-1, 1] \times [-0.5, 2]$. $t = 0, s = r$ is the only optimum.

When $m = 2$, where $r = \sqrt{1 - 0.5\beta}$, we define

$$U_0 = \begin{bmatrix} 0 & 0 \\ r & 0 \end{bmatrix}, \quad U_1 = \begin{bmatrix} \sqrt{3}r/(2\sqrt{2}) & -\sqrt{3}r/(2\sqrt{2}) \\ r/(2\sqrt{2}) & r/(2\sqrt{2}) \end{bmatrix}, \quad U_2 = \begin{bmatrix} 0 & 0 \\ 0 & r \end{bmatrix},$$

$$v_0 = \begin{bmatrix} r \\ 0 \end{bmatrix}, \quad v_1 = \begin{bmatrix} r/\sqrt{2} \\ r/\sqrt{2} \end{bmatrix}, \quad v_2 = \begin{bmatrix} 0 \\ r \end{bmatrix}.$$

Then, we plot

$$F(t, s) = L(\cos(t)U_0 + 2s(U_1 - U_0) + \sin(t)U_2, \cos(t)v_0 + 2s(v_1 - v_0) + \sin(t)v_2).$$

for $(t, s) \in [-0.25, 0.6] \times [-0.5, 0.3]$. The optimal solutions here are $(t, s) = (0, 0.5)$ and the line $s = 0, t \geq 0$.

When $m = 3$, where $r = \sqrt{1 - 0.5\beta}$, we define

$$U_0 = \begin{bmatrix} 0,0,0 \\ r,0,0 \end{bmatrix}, U_1 = \begin{bmatrix} 0 & \sqrt{3}r/(2\sqrt{2}) & -\sqrt{3}r/(2\sqrt{2}) \\ 0 & r/(2\sqrt{2}) & r/(2\sqrt{2}) \end{bmatrix}, U_2 = \begin{bmatrix} 0,0,0 \\ 0,r,0 \end{bmatrix},$$

$$v_0 = \begin{bmatrix} r \\ 0 \\ 0 \end{bmatrix}, \quad v_1 = \begin{bmatrix} 0 \\ r/\sqrt{2} \\ r/\sqrt{2} \end{bmatrix}, \quad v_2 = \begin{bmatrix} 0 \\ r \\ 0 \end{bmatrix}.$$

Then, we plot

$$F(t, s) = L(\cos(t)\cos(s)U_0 + \cos(t)\sin(s)U_1 + \sin(t)U_2, \cos(t)\cos(s)v_0 + \cos(t)\sin(s)v_1 + \sin(t)v_2).$$

for $(t, s) \in [-0.5, 1] \times [-0.5, 1]$. The optimal solutions are $s = 0, t \geq 0$ and $t = 0, s \geq 0$.

The contour plot of the loss landscape can be found in Figure 4. It clearly shows that the connectivity behavior of the optimal solution changes.

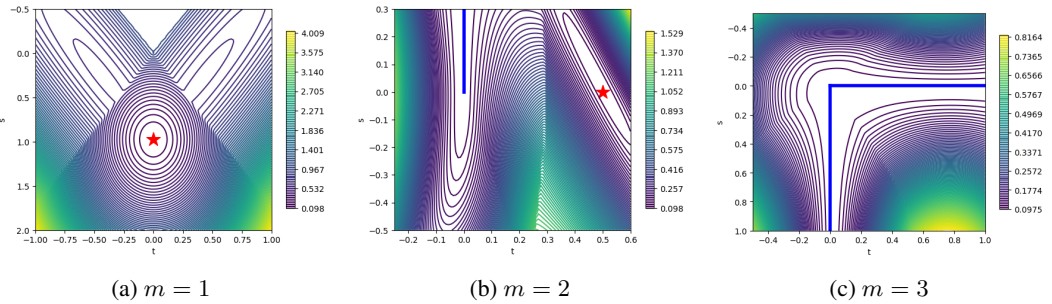

(a) $m = 1$        (b) $m = 2$        (c) $m = 3$

Figure 4: **A contour plot of the loss landscape** The three figures show the contour plot of the loss landscape shown in Figure 2. We can see the staircase of connectivity more clearly.

Figure 5 gives what the gradient descent actually learns for the problem. We can see that gradient descent finds multiple optimal solutions, which verifies our claim that we have a continuum of optimal solutions. We present the case both when $m = 3$ and $m = 5$. When $m$ is increased, the model gets less stuck at local minima.

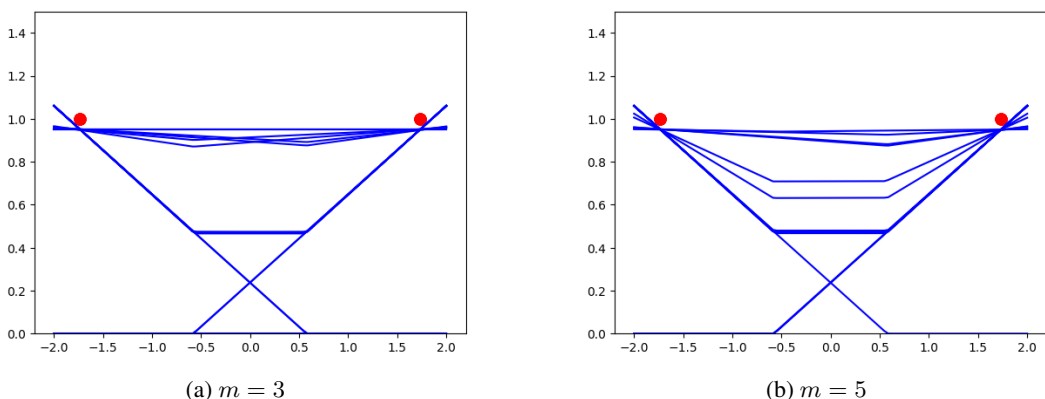

(a) $m = 3$             (b) $m = 5$

Figure 5: **Learned functions found by gradient descent** The two figures show what functions gradient descent learns for the toy problem in Example 1. For both cases in $m = 3$, $m = 5$, either gradient descent gets stuck at a local minimum or finds one of the optimal networks in the continuum of optimal solutions.

At last, we show that all optimal functions can be written as

$$f(x) = \sqrt{\kappa t}(\frac{\sqrt{3\kappa t}}{2}x + \frac{\sqrt{\kappa t}}{2})_+ + \sqrt{\kappa t}(-\frac{\sqrt{3\kappa t}}{2}x + \frac{\sqrt{\kappa t}}{2})_+ + \sqrt{\kappa(1 - 2t)}\left(\sqrt{\kappa(1 - 2t)}\right)_+,$$

where $\kappa = 1 - \beta/2$ and $t \in [0, 1/2]$. For $\nu^T = [1/2, 1/2]$, we know that $\max_{\|u\|_2 \leq 1} |\nu^T(Xu)_+| = 1$. Let's say the optimal model fit

$$y^* = \sum_{i=1}^{m}(Xu_i)_+\alpha_i.$$

Then,

$$\langle \nu, y^* \rangle \leq \sum_{i=1}^{m}|\nu^T(X\frac{u_i}{\|u_i\|_2})|\|u_i\|_2|\alpha_i| \leq \frac{1}{2}\Big(\sum_{i=1}^{m}\|u_i\|_2^2 + |\alpha_i|^2\Big).$$

This means the objective has a lower bound $\frac{1}{2}\|y^* - y\|_2^2 + \beta\langle\nu, y^*\rangle$, and minimum of the lower bound is attained when $y^* = [1 - \beta/2, 1 - \beta/2]^T$. Substitute to see that the lower bound of the objective is $\beta - \beta^2/4$, and when $u_1 = [0, \sqrt{1 - \beta/2}]^T$, $\alpha_1 = \sqrt{1 - \beta/2}$ we have a solution with cost $\beta - \beta^2/4$ hence $y^*$ is indeed optimal. $\nu^* = y - y^*$, and use Theorem 1 to find the complete solution set.

## B  DETAILS ON THE TOY EXAMPLE IN FIGURE 3

The construction of $x$ follows from Proposition 3. For the particular example we distribute the angles identically, hence we obtain the form in Example 2. The six optimal directions are

$$\begin{bmatrix} \sqrt{3}/2 \\ 1/2 \end{bmatrix}, \begin{bmatrix} \sqrt{2}/2 \\ \sqrt{2}/2 \end{bmatrix}, \begin{bmatrix} 1/2 \\ \sqrt{3}/2 \end{bmatrix}, \begin{bmatrix} \sqrt{6} - \sqrt{2}/4 \\ \sqrt{6} + \sqrt{2}/4 \end{bmatrix}, \begin{bmatrix} 0 \\ 1 \end{bmatrix}, \begin{bmatrix} -\sqrt{3}/2 \\ 1/2 \end{bmatrix}.$$

Let's note these optimal directions as $\bar{u}_1, \bar{u}_2, \cdots \bar{u}_6$. We construct $y$ as

$$y = 20((X\bar{u}_1)_+ + (X\bar{u}_3)_+ + (X\bar{u}_5)_+),$$

which is numerically very similar to $[94, 29, 24, 20, 20]$. Note that the class of optimal interpolators are

$$f(x) = (20 - 7.076t)([x, 1] \cdot \bar{u}_1)_+ + (13.1592t)([x, 1] \cdot \bar{u}_2)_+ + (20 - 13.1623t)([x, 1] \cdot \bar{u}_3)_+$$
$$+ (13.159t)([x, 1] \cdot \bar{u}_4)_+ + (20 - 7.081t)([x, 1] \cdot \bar{u}_5)_+ + t([x, 1] \cdot \bar{u}_6)_+,$$

where $t \in [0, 1.5194]$ which all have the same optimal cost 60.

Similar to the experiment in Appendix A, we give an example of the learned functions by gradient descent in Figure 6. We set $\beta = 0.1$ and solve the regularized problem. Here we find multiple functions as optimal: and the important remark is that not all (as some do stuck at local minima), but there exists different interpolators with the same cost.

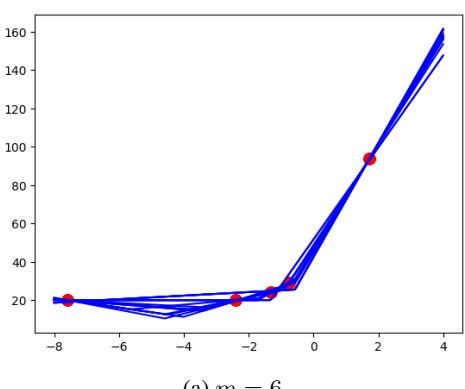
(a) $m = 6$

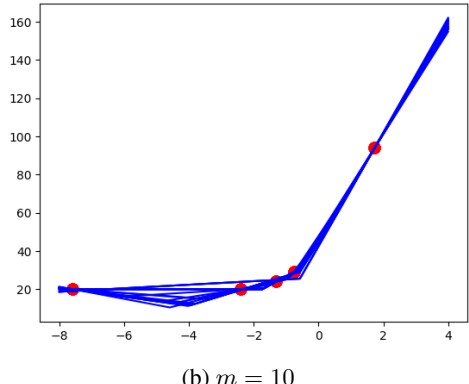
(b) $m = 10$

Figure 6: **Learned interpolators found by gradient descent** The two figures show what functions gradient descent learns for the toy problem in Example 2. We set $\beta = 0.1$ to approximately solve the minimum-norm interpolation problem. For both cases $m = 6$ and $m = 10$, either gradient descent gets stuck at a local minimum or finds one of the optimal networks in the continuum of optimal solutions.

## C  PROOFS IN SECTION 3.1

In this section, we briefly discuss how Theorem 1 is derived and the intuition behind it. Consider the problem introduced in equation 4, and write its optimal solution set as $\Theta^*$. To discuss the solution set, we first define the set of optimal model fits, which was first introduced in Mishkin & Pilanci (2023).

**Definition C.1.** *Mishkin & Pilanci (2023) The set of optimal model fits $\mathcal{C}_y$ is defined as*

$$\mathcal{C}_y = \Big\{ \sum_{i=1}^{P} D_i X(u_i^* - v_i^*) \mid (u_i^*, v_i^*)_{i=1}^{P} \in \Theta^* \Big\}.$$

When $L$ is strictly convex, which is the case for $l_2$ regression for instance, $\mathcal{C}_y$ becomes a singleton Mishkin & Pilanci (2023).

**Proposition C.1.** *(Proposition 1 of the paper) Mishkin & Pilanci (2023) If the loss function $L$ is strictly convex, the optimal model fit is unique, i.e. for the set of optimal model fit*

$$\mathcal{C}_y = \Big\{ \sum_{i=1}^{P} D_i X(u_i^* - v_i^*) \mid (u_i^*, v_i^*)_{i=1}^{P} \in \Theta^* \Big\},$$

$\mathcal{C}_y = \{y^*\}$ *for some* $y^* \in \mathbb{R}^n$.

*Proof.* Assume $y_1, y_2 \in \mathcal{C}_y$ and $y_1 \neq y_2$. Let $\sum_{i=1}^{P} D_i X(u_i - v_i) = y_1$ and $\sum_{i=1}^{P} D_i X(u_i' - v_i') = y_2$ for $(u_i, v_i)_{i=1}^{P}, (u_i', v_i')_{i=1}^{P} \in \Theta^*$. Think of $(\frac{u_i + u_i'}{2}, \frac{v_i + v_i'}{2})_{i=1}^{P} = \theta_{avg}$. The objective value of $\theta_{avg}$ is

$$L(\frac{y_1 + y_2}{2}, y) + \beta \sum_{i=1}^{P} \left( \|\frac{u_i + u_i'}{2}\|_2 + \|\frac{v_i + v_i'}{2}\|_2 \right)$$

which is strictly smaller than

$$\frac{1}{2} \left( L(y_1, y) + \beta \left( \sum_{i=1}^{P} \|u_i\|_2 + \|v_i\|_2 \right) + L(y_2, y) + \beta \left( \sum_{i=1}^{P} \|u_i'\|_2 + \|v_i'\|_2 \right) \right).$$

The strict inequality follows from the fact that $L$ is strictly convex. Contradiction follows, as we have found a parameter that has smaller objective value than the optimal cost.  □

It is not necessary to characterize $\mathcal{C}_y$ as a singleton to derive the solution set itself (see Section 4.1). However, for the notion of the optimal polytope and its application to the staircase of connectivity, we will need that $\mathcal{C}_y$ is a singleton.

Before proving the optimal polytope characterization, we show that the $\bar{u}_i, \bar{v}_i$ introduced in Theorem 1 can be uniquely determined by solving the given optimization problem.

**Proposition C.2.** *Consider the optimization problem*

$$\min_{u \in \mathcal{S}_i} \nu^T D_i X u, \quad \min_{u \in \mathcal{S}_i} -\nu^T D_i X u,$$

*where $\nu \in \mathbb{R}^n$ is an arbitrary vector and $\mathcal{S}_i = \mathcal{K}_i \cap \{u \mid \|u\|_2 \leq 1\}$. If the optimal objective is nonzero, there exists a unique minimizer.*

*Proof.* The problem is equivalent to

$$\min_{u \in \mathcal{S}_i} (w^*)^T u,$$

which is a linear program on a convex set. We write $w^* = \pm X^T D_i \nu^*$ for convenience. Let's say the optimal objective $p^* < 0$ and we have two minimizers $u_1^*, u_2^*$. The first thing to notice is that $\|u_1^*\|_2 = 1$. The reason is that when $\|u_1^*\|_2 < 1$, we can scale it to decrease the objective. Similarly, $\|u_2^*\|_2 = 1$. As they are two different minimizers, we know that

$$(w^*)^T u_1^* = (w^*)^T u_2^* = p^* = (w^*)^T (\frac{u_1^* + u_2^*}{2}),$$

and $\|u_1^* + u_2^*\|_2 < 2$ because $u_1^* \neq u_2^*$. Scale $(u_1^* + u_2^*)/2$ to obtain contradiction that $u_1^*$ is the minimizer.  □

**Theorem C.1.** *(Theorem 1 of the paper) Suppose $L$ is a strictly convex loss function. The directions of optimal parameters of the problem in equation 4, noted as $\bar{u}_i, \bar{v}_i$, are uniquely determined from the dual problem,*

$$\bar{u}_i = \arg\min_{u \in \mathcal{S}_i} \nu^{*T} D_i X u \ \text{ if } \ \min_{u \in \mathcal{S}_i} \nu^{*T} D_i X u = -\beta, \ 0 \ otherwise,$$

$$\bar{v}_i = \arg\min_{v \in \mathcal{S}_i} -\nu^{*T} D_i X v \ \text{ if } \ \min_{v \in \mathcal{S}_i} -\nu^{*T} D_i X v = -\beta, \ 0 \ otherwise.$$

*where $\nu^*$ is any dual optimum that satisfies*

$$\nu^* = \arg\max -L^*(\nu) \ \text{ subject to } \ |\nu^T D_i X u| \leq \beta \|u\|_2 \ \ \forall u \in \mathcal{K}_i, i \in [P].$$

*Here, $D_i s$ are all possible arrangements $diag(1(Xh \geq 0))$ for $i \in [P]$, $\mathcal{S}_i = \mathcal{K}_i \cap \{u \mid \|u\|_2 \leq 1\}$. Moreover, the solution set of equation 4 is given as a polytope,*

$$\mathcal{P}_{\nu^*}^* := \left\{ (c_i \bar{u}_i, d_i \bar{v}_i)_{i=1}^P \mid c_i, d_i \geq 0 \quad \forall i \in [P], \quad \sum_{i=1}^P D_i X \bar{u}_i c_i - D_i X \bar{v}_i d_i = y^* \right\} \subseteq \mathbb{R}^{2dP}, \tag{13}$$

*where $y^*$ is the unique optimal fit satisfying $\mathcal{C}_y = \{y^*\}$.*

*Proof.* Let's note $\Theta^*$ as the solution set of equation 4. Also, fix $\nu^*$ to be any dual optimum. The directions $\bar{u}_i, \bar{v}_i$ are uniquely determined from Proposition C.2. Define $\mathcal{P}^*$ to be the set defined in equation 13: note the dependence of $\mathcal{P}^*$ has with $\nu^*$ (though we will see that for any choice of $\nu^*$, $\mathcal{P}_{\nu^*}^* = \Theta^*$ and the choice of $\nu^*$ does not matter).

We first show that $\Theta^* \subseteq \mathcal{P}_{\nu^*}^*$. Take a point $(u_i^*, v_i^*)_{i=1}^P \in \Theta^*$. We first know that $\sum_{i=1}^P D_i X(u_i^* - v_i^*) = y^*$ from Proposition 1. What we would like to do is showing the existence of $c_i, d_i$ that satisfies

$$c_i \geq 0, \ u_i^* = c_i \bar{u}_i, \ d_i \geq 0, \ v_i^* = d_i \bar{v}_i,$$

where $\bar{u}_i, \bar{v}_i$ are

$$\bar{u}_i = \arg\min_{u \in \mathcal{S}_i} \nu^{*T} D_i X u \ \text{ if } \ \min_{u \in \mathcal{S}_i} \nu^{*T} D_i X u = -\beta, \ 0 \ otherwise,$$

$$\bar{v}_i = \arg\min_{v \in \mathcal{S}_i} -\nu^{*T} D_i X v \ \text{ if } \ \min_{v \in \mathcal{S}_i} -\nu^{*T} D_i X v = -\beta, \ 0 \ otherwise.$$

Consider the Lagrangian

$$\mathcal{L}((u_i, v_i)_{i=1}^P, z, \nu) = L(z, y) - \nu^T z + \sum_{i=1}^P (\beta \|u_i\|_2 + \nu^T D_i X u_i) + \sum_{i=1}^P (\beta \|v_i\|_2 - \nu^T D_i X v_i),$$

where $u_i, v_i \in \mathcal{K}_i$. We can see that

$$\min_{u_i, v_i \in \mathcal{K}_i, z} \max_{\nu} \mathcal{L}((u_i, v_i)_{i=1}^P, z, \nu) = \max_{\nu} \min_{u_i, v_i \in \mathcal{K}_i, z} \mathcal{L}((u_i, v_i)_{i=1}^P, z, \nu),$$

because $\nu$ is the dual variable that is only related to linear constraints. We can prove the fact rigorously by following the reasoning in Boyd & Vandenberghe (2004). We prove the fact for completeness.

First, we define the set

$$\mathcal{A} = \{(w - \sum_{i=1}^P D_i X(u_i - v_i), t) \mid u_i, v_i \in \mathcal{K}_i, L(w, y) + \beta \sum_{i=1}^P \|u_i\|_2 + \|v_i\|_2 \leq t\},$$

where $\mathcal{A} \subseteq \mathbb{R}^n \times \mathbb{R}$. s $\mathcal{A}$ is a convex set. Now, denote the optimal value of problem in equation 4 as $p^*$. When we define

$$\mathcal{B} = \{(0, s) \mid s < p^*\},$$

it is clear that $\mathcal{A} \cap \mathcal{B} = \emptyset$.
By the separating hyperplane theorem, there exists $(\tilde{\nu}, \tilde{\mu}) \in \mathbb{R}^n \times \mathbb{R}$ which is nonzero, $\alpha$ such that

$$(z, t) \in \mathcal{A} \Rightarrow \tilde{\nu}^T z + \tilde{\mu} t \geq \alpha \geq \tilde{\mu} p^*,$$

and we also know $\tilde{\mu} \geq 0$: else $t \to \infty$ and contradiction follows. If $\tilde{\mu} > 0$ we have $\mathcal{L}((u_i, v_i)_{i=1}^P, z, \tilde{\nu}/\tilde{\mu}) \geq p^*$, and strong duality follows. If $\tilde{\mu} = 0$, we conclude that for all $(u_i, v_i)_{i=1}^P, z$ we have that $(\tilde{\nu})^T(z - \sum_{i=1}^P D_i X(u_i - v_i)) \geq 0$, which is simply impossible. Hence, $\tilde{\mu} > 0$ and strong duality holds.

Moreover, the dual problem

$$\max_{\nu} \min_{(u_i, v_i)_{i=1}^P, z} \mathcal{L}((u_i, v_i)_{i=1}^P, z, \nu)$$

writes

$$\text{maximize} - L^*(\nu) \text{ subject to } \beta\|u\|_2 \geq |\nu^T D_i X u| \ \forall u \in \mathcal{K}_i, i \in [P].$$

The reason is the following: suppose for some $\nu' \in \mathbb{R}^n$, there exists $u_i'$ that satisfies $u_i' \in \mathcal{K}_i$ and $\nu'^T D_i X u_i' + \beta\|u_i'\|_2 < 0$. As we can scale $t \to \infty$ for $tu_i'$ to see that for that $\nu'$, $\min_{(u_i, v_i)_{i=1}^P, z} \mathcal{L}((u_i, v_i)_{i=1}^P, z, \nu') = -\infty$. Hence, this $\nu'$ cannot be the dual optimum. This means we only need to see the $\nu$ that satisfies $\nu^T D_i X u + \beta\|u\|_2 \geq 0$ for all $u \in \mathcal{K}_i, i \in [P]$. Similarly, we only need to see $\nu$ that satisfies $-\nu^T D_i X u + \beta\|u\|_2 \geq 0$ for all $u \in \mathcal{K}_i, i \in [P]$. Hence, $\nu^*$ is the maximizer of

$$\max_{\nu} \min_{z} L(z, y) - \nu^T z \quad \text{subject to} \ \beta\|u\|_2 \geq |\nu^T D_i X u| \ \forall u \in \mathcal{K}_i, i \in [P],$$

and the rest follows.

As strong duality holds, for any primal optimum $((u_i^*, v_i^*)_{i=1}^P, y^*)$, the function $\mathcal{L}((u_i, v_i)_{i=1}^P, z, \nu^*)$ attains minimum at $((u_i^*, v_i^*)_{i=1}^P, y^*)$ Boyd & Vandenberghe (2004). Note that $z^*$ is always $y^*$ due to Proposition 1, and replaced by it.

Now, as

$$\mathcal{L}((u_i, v_i)_{i=1}^P, z, \nu^*) = L(z, y) - \nu^{*T} z + \sum_{i=1}^P (\beta\|u_i\|_2 + \nu^{*T} D_i X u_i) + \sum_{i=1}^P (\beta\|v_i\|_2 - \nu^{*T} D_i X v_i),$$

each $u_i^*$ becomes the minimizer of $\beta\|u\|_2 + \nu^{*T} D_i X u$ subject to $u \in \mathcal{K}_i$ and each $v_i^*$ becomes the minimizer of $\beta\|u\|_2 - \nu^{*T} D_i X u$ subject to $u \in \mathcal{K}_i$. Recall that $\nu^*$ is a vector that satisfies $\beta\|u\|_2 \geq |\nu^T D_i X u| \ \forall u \in \mathcal{K}_i, i \in [P]$, and when $u = 0$, both $\beta\|u\|_2 + \nu^{*T} D_i X u$ and $\beta\|u\|_2 - \nu^{*T} D_i X u$ has function value 0. This implies that the minimum of both $\beta\|u\|_2 + \nu^{*T} D_i X u$ and $\beta\|u\|_2 - \nu^{*T} D_i X u$ subject to $u \in \mathcal{K}_i$ is 0 for all $i \in [P]$. As $((u_i^*, v_i^*)_{i=1}^P, y^*)$ minimizes $\mathcal{L}((u_i, v_i)_{i=1}^P, z, \nu^*)$,

$$\beta\|u_i^*\|_2 + \nu^{*T} D_i X u_i^* = 0, \quad \beta\|v_i^*\|_2 - \nu^{*T} D_i X v_i^* = 0.$$

We will find $c_i \geq 0$, and finding $d_i$ will be identical. Let's divide into cases.
i) When $u_i^* = 0$, let $c_i = 0$ to find $c_i \geq 0$ that satisfies $u_i^* = c_i \bar{u}_i$.
ii) When $u_i^* \neq 0$, notice that

$$\min_{u \in \mathcal{S}_i} \nu^{*T} D_i X u = -\beta \neq 0,$$

and the optimum is attained at $u_i^*/\|u_i^*\|_2$. To see this, recall that $(\nu^*)^T D_i X u + \beta\|u\|_2 \geq 0$ and $(\nu^*)^T D_i X u/\|u\|_2 \geq -\beta$ for all nonzero $u \in \mathcal{K}_i$, which implies that $\min_{u \in \mathcal{S}_i}(\nu^*)^T D_i X u = -\beta$. Furthermore, by Proposition C.2, there exists a unique minimizer of the problem $\min_{u \in \mathcal{S}_i}(\nu^*)^T D_i X u$, and $u_i^*/\|u_i^*\|_2 = \bar{u}_i$. Hence choosing $c_i = \|u_i^*\|_2$ gives $c_i \geq 0$ that satisfies $u_i^* = c_i \bar{u}_i$. Hence, we have found $c_i \geq 0, d_i \geq 0$ that satisfies $u_i^* = c_i \bar{u}_i, v_i^* = d_i \bar{v}_i$ and $\sum_{i=1}^P D_i X(u_i^* - v_i^*) = \sum_{i=1}^P D_i X(c_i \bar{u}_i - d_i \bar{v}_i) = y^*$, meaning $(u_i^*, v_i^*)_{i=1}^P \in \mathcal{P}^*$.

Now, we show that $\mathcal{P}_{\nu^*}^* \subseteq \Theta^*$. Take an element $(c_i \bar{u}_i, d_i \bar{v}_i)_{i=1}^P \in \mathcal{P}_{\nu^*}^*$. It is clear that $c_i \bar{u}_i \in \mathcal{C}_i, d_i \bar{v}_i \in \mathcal{D}_i$. If $\bar{u}_i \neq 0$, we know that $(\nu^*)^T D_i X \bar{u}_i = -\beta$. Similarly, if $\bar{v}_i \neq 0$, we know that $-(\nu^*)^T D_i X \bar{v}_i = -\beta$. Also, if $\bar{u}_i, \bar{v}_i \neq 0$, $\|\bar{u}_i\|_2 = 1, \|\bar{v}_i\|_2 = 1$, see the proof of Proposition C.2 why this holds. Now, let's calculate the objective of $(c_i \bar{u}_i, d_i \bar{v}_i)_{i=1}^P$. We know $\sum_{i=1}^P D_i X(c_i \bar{u}_i - d_i \bar{v}_i) = y^*$, hence the objective becomes

$$L(y^*, y) + \beta \sum_{\bar{u}_i \neq 0} c_i + \beta \sum_{\bar{v}_i \neq 0} d_i,$$

using $\|\bar{u}_i\|_2 = 1, \|\bar{v}_i\|_2 = 1$. Now, as $\sum_{i=1}^{P} D_i X(c_i \bar{u}_i - d_i \bar{v}_i) = y^*$, multiplying $(\nu^*)^T$ on both sides gives $\sum_{\bar{u}_i \neq 0} c_i + \sum_{\bar{v}_i \neq 0} d_i = -\langle \nu^*, y^* \rangle / \beta$. Hence, the calculated objective becomes

$$L(y^*, y) - \langle \nu^*, y^* \rangle,$$

hence for all points in $\mathcal{P}_{\nu^*}^*$, the objective becomes constant. We already know that $\Theta^* \subseteq \mathcal{P}_{\nu^*}^*$. This means all points in $\mathcal{P}_{\nu^*}^*$ have the same optimal objective value, and $\mathcal{P}_{\nu^*}^* \subseteq \Theta^*$. This finishes the proof. □

**Corollary C.1.** *(Corollary 1 of the paper) Consider the optimization problem in equation 3. Denote the set of Clarke stationary points of equation 3 as $\Theta_C$. The set*

$$\bigcup_{j=1}^{m} \left\{ \frac{w_j}{\|w_j\|_2} \mid (w_i, \alpha_i)_{i=1}^{m} \in \Theta_C, \ w_j \neq 0 \right\},$$

*is finite, and each direction is determined by the dual optimum of the subsampled convex program.*

*Proof.* From Ergen & Pilanci (2023), we know that all Clarke stationary points of equation 3 have a corresponding subsampled convex problem. More specifically, for any $(w_i, \alpha_i)_{i=1}^{m} \in \Theta_C$, we have a convex program with subsampled arrangement patterns $\tilde{D}_1, \tilde{D}_2, \cdots \tilde{D}_m \in \{D_i\}_{i=1}^{P}$,

$$\min_{u_i, v_i \in \tilde{\mathcal{K}}_i} L\left( \sum_{i=1}^{m} \tilde{D}_i X(u_i - v_i), y \right) + \beta \left( \sum_{i=1}^{m} \|u_i\|_2 + \|v_i\|_2 \right),$$

and a solution mapping

$$(w_i, \alpha_i) = \begin{cases} (u_i / \sqrt{\|u_i\|_2}, \ \sqrt{\|u_i\|_2}) & if \quad u_i \neq 0, \\ (v_i / \sqrt{\|v_i\|_2}, \ -\sqrt{\|v_i\|_2}) & if \quad v_i \neq 0. \end{cases}$$

Hence, the set of first-layer directions of Clarke stationary points is contained in the set of optimal directions of the subsampled convex program. As there are only finitely many subsampled convex programs, and each convex program has a unique set of fixed optimal directions, we know that the set

$$\bigcup_{j=1}^{m} \left\{ \frac{w_j}{\|w_j\|_2} \mid (w_i, \alpha_i)_{i=1}^{m} \in \Theta_C, \ w_j \neq 0 \right\},$$

is a finite set. Furthermore, applying Theorem 1 to the subsampled convex program leads to the wanted result. □

# D PROOFS IN SECTION 3.2

In this section we prove Theorem 2, using the cardinality-constrained optimal polytope $\mathcal{P}^*(m)$ defined in equation 7. One thing to have in mind is that we are not trying to prove that $\mathcal{P}^*(m)$ and $\Theta^*(m)$, the optimal set of the original problem in equation 3 with width $m$, are homeomorphic. Rather, we will argue that certain mappings enable us to link the connectivity behavior between $\mathcal{P}^*(m)$ and $\Theta^*(m)$ to arrive at Theorem 2.

We first start by defining some relevant concepts. As a starting point, we define the cardinality of a solution.

**Definition D.1.** *The cardinality of a solution $(u_i, v_i)_{i=1}^P \in \mathcal{P}^*$ is defined as*

$$\operatorname{card}((u_i, v_i)_{i=1}^P) = \sum_{i=1}^P 1(u_i \neq 0) + 1(v_i \neq 0).$$

We introduce the cardinality-constrained optimal polytope again.

**Definition D.2.** *The cardinality constrained optimal polytope $\mathcal{P}^*(m)$ is defined as the set*

$$\mathcal{P}^*(m) := \left\{ (u_i, v_i)_{i=1}^P \mid (u_i, v_i)_{i=1}^P \in \mathcal{P}^*, \operatorname{card}((u_i, v_i)_{i=1}^P) \leq m \right\} \subseteq \mathbb{R}^{2dP}. \tag{14}$$

One remark is that the largest possible cardinality of $\mathcal{P}^*$ may be huge: in worst case, it could be that the largest cardinality is in a scale of $P$, which is the number of all possible arrangement patterns. In general, the number of arrangement patterns are $O(n^d)$, hence $\mathcal{P}^*(m)$ consists of a very small portion of $\mathcal{P}^*$.

The next concept we introduce is the notion of irreducible solutions. This set can be understood as a set of minimal networks discussed in Mishkin & Pilanci (2023), and is used to define the critical widths of the staircase.

**Definition D.3.** *The irreducible solution set is defined as the set*

$$\mathcal{P}^*_{irr} = \left\{ (u_i, v_i)_{i=1}^P \mid (u_i, v_i)_{i=1}^P \in \mathcal{P}^*, \ \{D_i X u_i\}_{u_i \neq 0} \cup \{D_i X v_i\}_{v_i \neq 0} \text{ linearly independent} \right\}. \tag{15}$$

One intuition of the irreducible solution set is that it is the set of "smallest solutions Mishkin & Pilanci (2023)": if the set $\{D_i X u_i\}_{u_i \neq 0} \cup \{D_i X v_i\}_{v_i \neq 0}$ is linearly dependent we can find a strictly smaller conic combination using the vectors from $\{D_i X u_i\}_{u_i \neq 0} \cup \{-D_i X v_i\}_{v_i \neq 0}$. The set $\mathcal{P}^*_{irr}$ can be understood as a collection of solutions obtained from repeating this "pruning step" - a step that finds smaller solutions using linear dependence. A more rigorous definition of pruning is the following. Note that the existence of $m$ with a nonzero solution in $\Theta^*(m)$ implies $\mathcal{P}^*_{irr} \neq \emptyset$, which is equivalent to having a nonzero element in $\mathcal{P}^*$. We assume the nontrivial case where $\mathcal{P}^*$ has a nonzero element from now on.

**Proposition D.1.** *The following three statements are equivalent:*

*i) There exists $m$ that satisfies $(w_i, \alpha_i)_{i=1}^m \neq 0 \in \Theta^*(m)$*
*ii) There exists $(u_i, v_i)_{i=1}^P \neq 0 \in \mathcal{P}^*$*
*iii) $\mathcal{P}^*_{irr} \neq \emptyset$*

*Proof.* i) $\Rightarrow$ ii): First assume $m \geq 2P$. Consider $\Phi((w_i, \alpha_i)_{i=1}^m) = (u_i, v_i)_{i=1}^P$, where $\Phi$ is defined in Definition D.8. We know that $(u_i, v_i)_{i=1}^P$ is not a solution of zeros, as if it were the case, $\sum_{i=1}^m (Xw_i)_+ \alpha_i = 0$ and $(0, 0)_{i=1}^m$ would have a strictly smaller objective, contradicting $(w_i, \alpha_i)_{i=1}^m \in \Theta^*(m)$. Now when we write the optimal value of the nonconvex objective in Equation (3) with $p_m^*$, and the optimal value of the convex objective in Equation (4) with $p^*$, $p_m^* \leq p^*$. Also, when we write $L$ as the convex objective, we know that $L((u_i, v_i)_{i=1}^P) = p_m^* \geq p^*$. Hence we have found a nonzero $(u_i, v_i)_{i=1}^P$ that has objective value $p^*$, which means there is a point in $\mathcal{P}^*$ which is nonzero.
Now let $m < 2P$. Assume $\mathcal{P}^* = \{(0,0)_{i=1}^P\}$. We know that $p^* \leq p_m^*$, and $p^* = \frac{1}{2}\|y\|_2^2$, hence $\frac{1}{2}\|y\|_2^2 \leq p_m^*$. On the other hand, the value $\frac{1}{2}\|y\|_2^2$ is achievable by setting $(0, 0)_{i=1}^m$ - which means $p_m^* = p^*$. With the same logic, $\Phi((w_i, \alpha_i)_{i=1}^m)$ is not a solution of zeros and its objective value

is same as $p_m^*$ which is $p^*$ - leading to a contradiction that $\mathcal{P}^* = \{(0,0)_{i=1}^P\}$ since $(u_i, v_i)_{i=1}^P$ is nonzero and in $\mathcal{P}^*$.

ii) $\Rightarrow$ iii): We use the pruning step in Definition D.4 to find an element in $\mathcal{P}_{irr}^*$. Note that the pruning step does not end with 0, as $(u_i, v_i)_{i=1}^P$ is not zero and the pruning step should not decrease the objective.

iii) $\Rightarrow$ ii): As $\mathcal{P}_{irr}^* \neq \emptyset$, there is a nonzero solution $nz \in \mathcal{P}_{irr}^*$. As $\mathcal{P}_{irr}^* \subseteq \mathcal{P}^*$, we know the existence of a nonzero solution in $\mathcal{P}^*$.

ii) $\Rightarrow$ i): Set $m = 2P$ and consider $\Psi((u_i, v_i)_{i=1}^P)$, where $\Psi$ is defined in Definition D.7. $\qquad \square$

**Definition D.4.** *(Mishkin & Pilanci (2023)) Pruning a solution $(u_i, v_i)_{i=1}^P \in \mathcal{P}^*$ means repeating:*
*1. Finding a nontrivial linear combination*

$$\sum_{u_i \neq 0} c_i D_i X u_i + \sum_{v_i \neq 0} d_i D_i X v_i = 0,$$

*and without loss of generality assume $d_1 > 0$.*
*2. Constructing a solution with strictly less cardinality $(u_i', v_i')_{i=1}^P = ((1 + c_i t)u_i, (1 - d_i t)v_i)_{i=1}^P$, where $t = \min\{\min_{c_i < 0} -\frac{1}{c_i}, \min_{d_i > 0} \frac{1}{d_i}\}$, and $c_i, d_i$ are defined to be the coefficients defined in 1 when $u_i, v_i \neq 0$ and 0 otherwise.*
*until the set $\{D_i X u_i\}_{u_i \neq 0} \cup \{D_i X v_i\}_{v_i \neq 0}$ is linearly independent.*

The notion of minimality gives a discrete structure in $\mathcal{P}_{irr}^*$, hence the phase transitional behavior follows. The two critical widths of interest are the minimum / maximum cardinality of $\mathcal{P}_{irr}^*$. We denote

$$m^* := \min_{(u_i, v_i)_{i=1}^P \in \mathcal{P}_{irr}^*} \text{card}((u_i, v_i)_{i=1}^P), \quad M^* := \max_{(u_i, v_i)_{i=1}^P \in \mathcal{P}_{irr}^*} \text{card}((u_i, v_i)_{i=1}^P).$$

**Remark D.1.** *These widths can be found computationally by the following scheme: for $t = 1$ to $n$, choose $t$ vectors from the set $\{D_i X \bar{u}_i\}_{i=1}^P \cup \{D_i X \bar{v}_i\}_{i=1}^P$, where $\bar{u}_i, \bar{v}_i$ are optimal directions defined in Theorem 1. Check if they are linearly independent and can express $y^*$ as the conic combination of the $t$ vectors. The first value of $t$ that meets both criteria becomes $m^*$, and $M^*$ will be updated each time $t$ meets both criteria until $t$ becomes $n$.*

The two specific discontinuity results we can achieve are the following:

**Proposition D.2.** $\mathcal{P}^*(m^*)$ *is a finite set.*

*Proof.* Consider two points $(u_i, v_i)_{i=1}^P, (u_i', v_i')_{i=1}^P \in \mathcal{P}^*(m^*)$. Suppose the two points have the same support, i.e. $u_i \neq 0 \Leftrightarrow u_i' \neq 0$ and $v_i \neq 0 \Leftrightarrow v_i' \neq 0$ for $i \in [P]$. We know that as $\mathcal{P}^*(m^*) \subseteq \mathcal{P}^*$,

$$\sum_{i=1}^P D_i X(u_i - v_i) = y^* = \sum_{i=1}^P D_i X(u_i' - v_i').$$

Now, let's write the indices $\{i | u_i \neq 0\} = \{a_1, a_2, \cdots a_t\}$, $\{i | v_i \neq 0\} = \{b_1, b_2, \cdots b_s\}$. We have that $t + s \leq m^*$ as $(u_i, v_i)_{i=1}^P \in \mathcal{P}^*(m^*)$. From Theorem 1, we know the existence of $c_{a_i}, c_{a_i}' \geq 0$ for $i \in [t]$ and $d_{b_i}, d_{b_i}' \geq 0$ for $i \in [s]$ that satisfies

$$u_{a_i} = c_{a_i} \bar{u}_{a_i}, u_{a_i}' = c_{a_i}' \bar{u}_{a_i}, \quad \forall i \in [t],$$

$$v_{b_i} = d_{b_i} \bar{v}_{b_i}, v_{b_i}' = d_{b_i}' \bar{v}_{b_i}, \quad \forall i \in [s].$$

This means that

$$\sum_{i=1}^t c_{a_i} D_{a_i} X \bar{u}_{a_i} - \sum_{i=1}^s d_{b_i} D_{b_i} X \bar{v}_{b_i} = \sum_{i=1}^t c_{a_i}' D_{a_i} X \bar{u}_{a_i} - \sum_{i=1}^s d_{b_i}' D_{b_i} X \bar{v}_{b_i} = y^*,$$

and as $c_{a_i}, d_{b_i}$ s are not all the same, we have that the set

$$\{D_{a_i} X \bar{u}_{a_i}\}_{i=1}^t \cup \{D_{b_i} X \bar{v}_{b_i}\}_{i=1}^s$$

is linearly dependent. Now we apply pruning defined in Definition D.4 to find an irreducible solution with cardinality strictly less than $t + s = m^*$, which is a contradiction to the minimality of $m^*$. This

implies that two different points in $\mathcal{P}^*(m^*)$ cannot have identical support, and the number of points in the set is upper bounded with the number of possible support, which is finite. More specifically,

$$|\mathcal{P}^*(m^*)| \leq \sum_{j=1}^{m^*} \binom{2P}{j}.$$

$\square$

**Proposition D.3.** $\mathcal{P}^*(M^*)$ *has an isolated point, i.e. it has a point* $p \in \mathcal{P}^*(M^*)$ *that has no path in* $\mathcal{P}^*(M^*)$ *from* $p$ *to a different point* $p'$.

*Proof.* Take the maximal-cardinality solution from $(u_i^\circ, v_i^\circ)_{i=1}^P \in \mathcal{P}_{irr}^*$, namely the solution

$$(u_i^\circ, v_i^\circ)_{i=1}^P \in \mathcal{P}^*, \{D_i X u_i^\circ\}_{u_i^\circ \neq 0} \cup \{D_i X v_i^\circ\}_{v_i^\circ \neq 0} \text{ linearly independent,}$$

and $\text{card}((u_i^\circ, v_i^\circ)_{i=1}^P) = M^*$. Assume the existence of a continuous function $f : [0,1] \rightarrow \mathcal{P}^*(M^*)$ satisfying

$$f(0) = (u_i^\circ, v_i^\circ)_{i=1}^P, \; f(1) = (u_i', v_i')_{i=1}^P, \; f(0) \neq f(1).$$

Now, write $f(t) = (u_i(t), v_i(t))_{i=1}^P$ and define

$$c_i(t) = \begin{cases} 0 & if \quad \bar{u}_i = 0 \\ \|u_i(t)\|_2 & otherwise, \end{cases}$$

$$d_i(t) = \begin{cases} 0 & if \quad \bar{v}_i = 0 \\ \|v_i(t)\|_2 & otherwise, \end{cases}$$

For definition of $\bar{u}_i, \bar{v}_i$, see Theorem 1. Some things to notice are:
i) The functions $c_i(t), d_i(t) : [0,1] \rightarrow \mathbb{R}$ are continuous.
ii) $f(t) = (c_i(t)\bar{u}_i, d_i(t)\bar{v}_i)_{i=1}^P$. This holds because if $\bar{u}_i \neq 0$, $\|\bar{u}_i\|_2 = 1$, and same for $\bar{v}_i$.
iii) $\sum_{i=1}^P 1(c_i(t) \neq 0) + 1(d_i(t) \neq 0) \leq M^*$, and $\sum_{i=1}^P 1(c_i(0) \neq 0) + 1(d_i(0) \neq 0) = M^*$. The former holds because $f$ is a path in $\mathcal{P}^*(M^*)$, and the latter holds because $(u_i^\circ, v_i^\circ)_{i=1}^P$ has cardinality $M^*$.
iv) We know that there exists $t' \in [0,1]$ that satisfies $(c_i(t'), d_i(t'))_{i=1}^P \neq (c_i(0), d_i(0))_{i=1}^P$. It is because $f(0) \neq f(1)$.
Based on the observations, let's prove that if there exists such $f$, the set $\{D_i X u_i^\circ\}_{u_i^\circ \neq 0} \cup \{D_i X v_i^\circ\}_{v_i^\circ \neq 0}$ is linearly dependent. Thus we will arrive at a contradiction and will be able to show that there is no such $f$, and $(u_i^\circ, v_i^\circ)_{i=1}^P$ is isolated.
Let's define $t_1$ as

$$t_1 = \inf_{t \geq 0} \left\{ t \mid \sum_{i=1}^P (c_i(0) - c_i(t))^2 + (d_i(0) - d_i(t))^2 > 0 \right\}.$$

In other words, $t_1$ is the instant where $f(0) \neq f(t)$. From observation iv), we know that $t_1 \in [0,1]$. Another fact that we can deduce is that

$$\sum_{i=1}^P (c_i(0) - c_i(t_1))^2 + (d_i(0) - d_i(t_1))^2 = 0. \tag{16}$$

The reason is because if $\sum_{i=1}^P (c_i(0) - c_i(t_1))^2 + (d_i(0) - d_i(t_1))^2 > 0$, we can find some $\epsilon$ that will make $\sum_{i=1}^P (c_i(0) - c_i(t_1 - \epsilon))^2 + (d_i(0) - d_i(t_1 - \epsilon))^2 > 0$ because of continuity, which is a contradiction that $t_1$ is the infremum (because we have found a smaller $t$ that makes $\sum_{i=1}^P (c_i(0) - c_i(t))^2 + (d_i(0) - d_i(t))^2 > 0$). Hence, for $t \in [0, t_1]$,

$$\sum_{i=1}^P (c_i(0) - c_i(t))^2 + (d_i(0) - d_i(t))^2 = 0.$$

At last, we know that for any $\epsilon > 0$, there exists $t_\epsilon \in (t_1, t_1 + \epsilon)$ that satisfies

$$\sum_{i=1}^P (c_i(0) - c_i(t_\epsilon))^2 + (d_i(0) - d_i(t_\epsilon))^2 > 0. \tag{17}$$

If there is no $t \in (t_1, t_1 + \epsilon)$ that satisfies $\sum_{i=1}^{P}(c_i(0) - c_i(t))^2 + (d_i(0) - d_i(t))^2 > 0$ for some $\epsilon > 0$, it means that for all $t \in [0, t_1 + \epsilon_2]$, $\sum_{i=1}^{P}(c_i(0) - c_i(t))^2 + (d_i(0) - d_i(t))^2 = 0$: hence, the infremum should be strictly larger than $t_1$, which is a contradiction.

Now let's prove the claim that if there exists such $f$, the set $\{D_i X u_i^\circ\}_{u_i^\circ \neq 0} \cup \{D_i X v_i^\circ\}_{v_i^\circ \neq 0}$ is linearly dependent. From equation 16, we know that

$$c_i(0) = c_i(t_1), \quad d_i(0) = d_i(t_1) \quad \forall i \in [P].$$

Take $\epsilon_0 > 0$ sufficiently small so that for all $i \in [P]$ that satisfies $c_i(0) > 0$, $c_i(t) > 0$ for all $t \in [t_1 - \epsilon_0, t_1 + \epsilon_0]$, and for all $i \in [P]$ that satisfies $d_i(0) > 0$, $d_i(t) > 0$ for all $t \in [t_1 - \epsilon_0, t_1 + \epsilon_0]$. Such $\epsilon_0$ exists due to the continuity of $c_i, d_i$. Due to the definition, we know that

$$M^* = \sum_{i=1}^{P} 1(c_i(0) \neq 0) + 1(d_i(0) \neq 0) \leq \sum_{i=1}^{P} 1(c_i(t) \neq 0) + 1(d_i(t) \neq 0) \leq M^*,$$

(see observation iii) if any confusion exists), for all $t \in [t_1 - \epsilon_0, t_1 + \epsilon_0]$, and

$$\sum_{i=1}^{P} 1(c_i(t) \neq 0) + 1(d_i(t) \neq 0) = M^*, \quad \forall t \in [t_1 - \epsilon_0, t_1 + \epsilon_0].$$

This means that for all $t \in [t_1 - \epsilon_0, t_1 + \epsilon_0]$, we know that $c_i(0) > 0 \Leftrightarrow c_i(t) > 0$ and $d_i(0) > 0 \Leftrightarrow d_i(t) > 0$. For that $\epsilon_0$, we can find $t_{\epsilon_0}$ that was defined in equation 17. As $t_{\epsilon_0}$ satisfies

$$\sum_{i=1}^{P} (c_i(0) - c_i(t_{\epsilon_0}))^2 + (d_i(0) - d_i(t_{\epsilon_0}))^2 > 0,$$

$(c_i(0), d_i(0))_{i=1}^{P} \neq (c_i(t_{\epsilon_0}), d_i(t_{\epsilon_0}))_{i=1}^{P}$. Also, $c_i(0) > 0 \Leftrightarrow c_i(t_{\epsilon_0}) > 0$ and $d_i(0) > 0 \Leftrightarrow d_i(t_{\epsilon_0}) > 0$. Now we have found two different solutions $(c_i(0)\bar{u}_i, d_i(0)\bar{v}_i)_{i=1}^{P}$, $(c_i(t_{\epsilon_0})\bar{u}_i, d_i(t_{\epsilon_0})\bar{v}_i)_{i=1}^{P} \in \mathcal{P}^*(M^*)$, which means that

$$\sum_{i=1}^{P} c_i(0) D_i X \bar{u}_i - d_i(0) D_i X \bar{v}_i = y^* = \sum_{i=1}^{P} c_i(t_{\epsilon_0}) D_i X \bar{u}_i - d_i(t_{\epsilon_0}) D_i X \bar{v}_i. \tag{18}$$

As $(c_i(0), d_i(0))_{i=1}^{P} \neq (c_i(t_{\epsilon_0}), d_i(t_{\epsilon_0}))_{i=1}^{P}$ and $c_i(0) > 0 \Leftrightarrow c_i(t_{\epsilon_0}) > 0, d_i(0) > 0 \Leftrightarrow d_i(t_{\epsilon_0}) > 0$, we can see that equation 18 is two different linear combinations of the set $\{D_i X u_i^\circ\}_{u_i^\circ \neq 0} \cup \{D_i X v_i^\circ\}_{v_i^\circ \neq 0}$ - hence the set $\{D_i X u_i^\circ\}_{u_i^\circ \neq 0} \cup \{D_i X v_i^\circ\}_{v_i^\circ \neq 0}$ is linearly dependent.

As we claimed, we have arrived at a contradiction assuming a continuous path from $(u_i^\circ, v_i^\circ)_{i=1}^{P}$. Hence the point is isolated. $\qquad \square$

Next, we pay attention to the connectivity results of $\mathcal{P}^*(m)$. Our starting point will be noticing that for any $(u_i, v_i)_{i=1}^{P} \in \mathcal{P}^*$ which is not in $\mathcal{P}_{irr}^*$, the pruning mechanism defined in Definition D.4 gives a continuous path into $\mathcal{P}_{irr}^*$. This means that if we can connect two points in $\mathcal{P}_{irr}^* \cap \mathcal{P}^*(m)$ with a continuous path in $\mathcal{P}^*(m)$, the set $\mathcal{P}^*(m)$ is connected.

**Proposition D.4.** *Consider $(u_i, v_i)_{i=1}^{P} \in \mathcal{P}^* - \mathcal{P}_{irr}^*$. Let $m = \mathrm{card}((u_i, v_i)_{i=1}^{P})$. Then, there exists a continuous path in $\mathcal{P}^*(m)$ that starts with $(u_i, v_i)_{i=1}^{P}$ and ends with a different point $(u_i', v_i')_{i=1}^{P} \in \mathcal{P}^*(m) \cap \mathcal{P}_{irr}^*$.*

*Proof.* We will find such path by pruning the solution as in Definition D.4. For each iteration of the pruning step, the starting solution and the ending solution is connected by a continuous path. As we iterate, we concatenate each continuous path, hence the resulting path should be continuous. The next thing we have to check is that the path is contained in $\mathcal{P}^*(m)$. We can see this due to the fact that in each pruning iteration, the cardinality of the solution does not increase, and the initial cardinality of the solution is $m$. At last, when pruning ends we arrive at a irreducible solution, meaning the final solution we get from pruning is in $\mathcal{P}^*(m) \cap \mathcal{P}_{irr}^*$. $\qquad \square$

We have two different strategies to prove the connectedness of $\mathcal{P}^*(m)$. One is directly interpolating the two solutions in $\mathcal{P}_{irr}^*$, and increasing $m$ to guarantee the validity of such interpolation. From this, we obtain one critical width $m^* + M^*$.

**Proposition D.5.** *The set $\mathcal{P}^*(m^* + M^*)$ is connected.*

*Proof.* We first prove that two points $A, B \in \mathcal{P}^*(m^* + M^*) \cap \mathcal{P}^*_{irr}$ are connected with a continuous path in $\mathcal{P}^*(m^* + M^*)$.
Take two points $A \neq B \in \mathcal{P}^*(m^* + M^*) \cap \mathcal{P}^*_{irr}$. One observation that we can make is that $A$ has a continuous path in $\mathcal{P}^*(m^* + M^*)$ to a certain solution $A_m \in \mathcal{P}^*_{irr}$ that satisfies $\text{card}(A_m) = m^*$. The construction of such a path is simple: interpolate $A$ and $A_m$, i.e. $f(t) = (1 - t)A + tA_m$. As we know

$$\text{card}((1 - t)A + tA_m) \leq \text{card}(A) + \text{card}(A_m) \leq M^* + m^*,$$

the path has cardinality $\leq m^* + M^*$. The last inequality follows from the fact that $A \in \mathcal{P}^*_{irr}$ and $\text{card}(A) \leq M^*$. Due to the polytope characterization, $\mathcal{P}^*$ is convex, and the path is in $\mathcal{P}^*$. Combining these two, we know the existence of a path from $A$ to $A_m$ in $\mathcal{P}^*(m^* + M^*)$. Similarly, we know the existence of a path from $B$ to $B_m$ in $\mathcal{P}^*(m^* + M^*)$. At last, we know the existence of a path from $A_m$ to $B_m$, again by interpolating these two. Concluding, for any two $A, B \in \mathcal{P}^*(m^* + M^*) \cap \mathcal{P}^*_{irr}$, there exists a continuous path in $\mathcal{P}^*(m^* + M^*)$. Now, take any two points $A', B' \in \mathcal{P}^*(m^* + M^*)$. From Proposition D.4, there exists a continuous path in $\mathcal{P}^*(m^* + M^*)$ that starts from $A'$ to a certain $A_{irr} \in \mathcal{P}^*(m^* + M^*) \cap \mathcal{P}^*_{irr}$, and similarly there exists a path from $B'$ to $B_{irr}$. At last, there is a continuous path from $A_{irr}$ to $B_{irr}$ in $\mathcal{P}^*(m^* + M^*)$. Connect all paths to find a continuous path from $A'$ to $B'$ in $\mathcal{P}^*(m^* + M^*)$. □

Another strategy is more involved, which is not directly interpolating two solutions $A, B$ in $\mathcal{P}^*_{irr}$, but repeatedly interpolating $A$ with parts of $B$ until the two are connected with a path. We start with a particular lemma.

**Lemma D.1.** *Suppose we have two linearly independent sets $A = \{a_1, a_2, \cdots a_m\}$, $B = \{b_1, b_2, \cdots b_k\} \subseteq \mathbb{R}^n$ and a given subset $\mathcal{I} = \{a_{i_1}, a_{i_2}, \cdots a_{i_t}\} \subset A$. Also,*

$$\sum_{i=1}^{m} \lambda_i a_i = \sum_{i=1}^{k} \mu_i b_i,$$

*for some $\lambda \in \mathbb{R}^m$ that satisfies $\sum_{j=1}^{t} \lambda_{i_j} > 0$, and $\mu \in \mathbb{R}^k$ that satisfies $\mu > 0$. Then, there exists a vector $\mu^* \in \mathbb{R}^k$ that satisfies the following three properties:*
*1) $\|\mu^*\|_0 \leq n - m + 1$.*
*2) $\mu^* \geq 0$.*
*3) $\sum_{i=1}^{k} \mu_i^* b_i \in span(\{a_1, a_2, \cdots a_m\})$ and when we express*

$$\sum_{i=1}^{k} \mu_i^* b_i = \sum_{i=1}^{m} \delta_i a_i,$$

$$\sum_{j=1}^{t} \delta_{i_j} > 0.$$

*Proof.* If $k \leq n - m + 1$ there is nothing to prove. Assume $k > n - m + 1$. Showing the existence of a vector $\tilde{\mu}$ that satisfies $\|\tilde{\mu}\|_0 < \|\mu\|_0$, $\tilde{\mu} \geq 0$ and

$$\sum_{i=1}^{k} \tilde{\mu}_i b_i \in span(\{a_1, a_2, \cdots a_m\}), \quad \sum_{i=1}^{k} \tilde{\mu}_i b_i = \sum_{i=1}^{m} \delta_i a_i \quad and \quad \sum_{j=1}^{t} \delta_{i_j} > 0,$$

is enough to prove our proposed claim. That is because if the existence of such $\tilde{\mu}$ is proved, we can apply the existence result again to $A = \{a_1, a_2, \cdots, a_m\}$ and $\tilde{B} = \{b_i | i \in [k], \tilde{\mu}_i \neq 0\}$ with the same $\mathcal{I}$. The premises are all satisfied: from the definition of $\tilde{\mu}$ we know that

$$\sum_{i \in [k], \tilde{\mu}_i \neq 0} \tilde{\mu}_i b_i = \sum_{i=1}^{m} \delta_i a_i, \quad \sum_{j=1}^{t} \delta_{i_j} > 0,$$

and $\tilde{\mu}_i > 0$ if $\tilde{\mu}_i \neq 0$. Moreover, $|\tilde{B}| < |B|$. This means if we prove the existence of such $\tilde{\mu}$ and iteratively apply the existence result as stated above, we will arrive at a set $B^\circ \subseteq B$ that satisfies

$$|B^\circ| \leq n - m + 1, \quad \sum_{b_i \in B^\circ} \mu_i^\circ b_i = \sum_{i=1}^m \delta_i^\circ a_i, \quad \sum_{j=1}^t \delta_{i_j}^\circ > 0, \quad \mu_i^\circ > 0 \ if \ b_i \in B^\circ, \ 0 \ otherwise.$$

The reason why $|B^\circ| \leq n - m + 1$ is that if $|B^\circ| > n - m + 1$, we can find a subset of $B^\circ$ with strictly less cardinality with the same property, hence it is not the terminal set. For that set, choosing $\mu^\circ = \mu^*$ gives the wanted vector.

Now we show the existence of such $\tilde{\mu}$ when $k > n - m + 1$. We first extend $\{a_1, a_2, \cdots a_m\}$ to a basis of $\mathbb{R}^n$, and note it as $\{a_1, a_2, \cdots a_n\}$. Express each $b_i$ s as

$$b_i = \sum_{j=1}^n \gamma_{ij} a_j.$$

where $i \in [k]$. Now, write $\Gamma \in \mathbb{R}^{n \times k}$ as $\Gamma_{ij} = \gamma_{ji}$. What we know is the relation

$$\Gamma \mu = \begin{bmatrix} \lambda \\ 0_{n-m} \end{bmatrix},$$

which is simply the coordinate representation of $\sum_{i=1}^k \mu_i b_i$. Now we know the set

$$\{\mu \in \mathbb{R}^k \mid 1^T \Gamma[i_1, i_2, \cdots i_t]\mu = 0, \Gamma[m+1]^T \mu = 0, \cdots \Gamma[n]^T \mu = 0\}$$

has dimension at least $k - n + m - 1$, as each linear constraint decreases the dimension at most 1. Here $\Gamma[p_1, p_2, \cdots p_r] \in \mathbb{R}^{r \times k}$ denotes the concatenation of $r$ rows of $\Gamma$, $\Gamma[p_1]$ to $\Gamma[p_r]$. As $k - n + m - 1 > 0$, there exists a nonzero $\mu'$ that satisfies

$$\Gamma \mu' = \begin{bmatrix} \lambda' \\ 0_{n-m} \end{bmatrix}, \quad \sum_{j=1}^t \lambda'_{i_j} = 0.$$

For that $\mu'$, consider $\mu + \epsilon \mu'$ and either increase or decrease $\epsilon$ until the cardinality of $\mu$ decreases. As $\mu > 0$, we can always find such $\epsilon$ that satisfies $\|\mu + \epsilon \mu'\|_0 < \|\mu\|_0$. As we stop when the cardinality changes, $\mu + \epsilon \mu' \geq 0$ should also hold. At last, we know that

$$\Gamma(\mu + \epsilon \mu') = \begin{bmatrix} \lambda + \epsilon \lambda' \\ 0_{n-m} \end{bmatrix},$$

and as $\sum_{j=1}^t \lambda'_{i_j} = 0$, $\sum_{j=1}^t (\lambda_{i_J} + \epsilon \lambda'_{i_j}) > 0$. As the values of $\Gamma \mu$ directly correspond to the coordinate representation of $\{a_1, a_2, \cdots a_n\}$, we know that $\mu + \epsilon \mu'$ is the $\tilde{\mu}$ that we were looking for. This finishes the proof. $\square$

The necessary width in this case is $n + 1$, and we obtain the following result.

**Theorem D.1.** *The set $\mathcal{P}^*(n+1)$ is connected.*

*Proof.* Similar to the proof of Proposition D.5, we show that for any two $A, B \in \mathcal{P}^*(n+1) \cap \mathcal{P}^*_{irr}$, they are connected with a continuous path in $\mathcal{P}^*(n+1)$. The rest will directly follow.
First, let's write $A = (u_i, v_i)_{i=1}^P, B = (u'_i, v'_i)_{i=1}^P$. Also, let's write

$$\mathcal{A} = \{D_i X \bar{u}_i\}_{u_i \neq 0} \cup \{-D_i X \bar{v}_i\}_{v_i \neq 0}, \quad \mathcal{B} = \{D_i X \bar{u}_i\}_{u'_i \neq 0} \cup \{-D_i X \bar{v}_i\}_{v'_i \neq 0},$$

and note them as $\mathcal{A} = \{a_1, a_2, \cdots a_m\} \subseteq \mathbb{R}^n$, $\mathcal{B} = \{b_1, b_2, \cdots b_k\} \subseteq \mathbb{R}^n$. At last, $\lambda_1, \lambda_2, \cdots \lambda_m$, $\mu_1, \mu_2, \cdots \mu_k$ are unique nonnegative numbers that satisfy

$$\sum_{i=1}^m \lambda_i a_i = \sum_{i=1}^k \mu_i b_i = y^*.$$

The uniqueness follows from the fact that $A, B \in \mathcal{P}^*(n+1) \cap \mathcal{P}^*_{irr}$, and the nonnegativeness follows from the optimal polytope characterization in Theorem 1. Furthermore, note that $\lambda_p > 0$ for

all $p \in [m]$, $\mu_q > 0$ for all $q \in [k]$. For example, if $a_p = D_i X \bar{u}_i$ for some $u_i \neq 0$, $\lambda_p = \|u_i\|_2 > 0$. The rest is similar.

Our main proof strategy will be finding $k + m$ continuous functions $F_1, F_2, \cdots F_m, G_1, G_2, \cdots, G_k : [0,1] \to \mathbb{R}$ that satisfies:

Property 1) $F_i(0) = \lambda_i, F_i(1) = 0, F_i(t) \geq 0 \quad \forall i \in [m], t \in [0.1]$.
Property 2) $G_j(0) = 0, G_j(1) = \mu_j, G_j(t) \geq 0 \quad \forall j \in [k], t \in [0.1]$.
Property 3)

$$\sum_{i=1}^{m} F_i(t) a_i + \sum_{j=1}^{k} G_j(t) b_j = y^* \quad \forall t \in [0,1].$$

Property 4)

$$\sum_{i=1}^{m} \mathbb{1}(F_i(t) > 0) + \sum_{j=1}^{k} \mathbb{1}(G_j(t) > 0) \leq n + 1 \quad \forall t \in [0,1].$$

First, let's see that if we find such continuous functions that satisfy Property 1) to Property 4), we can construct a path from $A$ to $B$ in $\mathcal{P}^*(n+1)$. The specific path we construct is: $(u_i(t), v_i(t))_{i=1}^{P}$ given as

$$u_i(t) = \begin{cases} (F_p(t) + G_q(t))\bar{u}_i & if \quad a_p = b_q = D_i X \bar{u}_i \\ F_p(t)\bar{u}_i & if \quad a_p = D_i X \bar{u}_i, \nexists q \in [k] \; such \; that \; b_q = D_i X \bar{u}_i. \\ G_q(t)\bar{u}_i & if \quad b_q = D_i X \bar{u}_i, \nexists p \in [m] \; such \; that \; a_p = D_i X \bar{u}_i \\ 0 & otherwise. \end{cases}$$

$$v_i(t) = \begin{cases} (F_p(t) + G_q(t))\bar{v}_i & if \quad a_p = b_q = -D_i X \bar{v}_i \\ F_p(t)\bar{v}_i & if \quad a_p = -D_i X \bar{v}_i, \nexists q \in [k] \; such \; that \; b_q = -D_i X \bar{v}_i. \\ G_q(t)\bar{v}_i & if \quad b_q = -D_i X \bar{v}_i, \nexists q \in [m] such \; that \; a_p = -D_i X \bar{v}_i \\ 0 & otherwise. \end{cases}$$

Let's check that $(u_i(t), v_i(t))_{i=1}^{P}$ is a path from $A$ to $B$ in $\mathcal{P}^*(n+1)$. As $F_i(1) = 0$ for all $i \in [m]$, $G_j(0) = 0$ for all $j \in [k]$, we can see that $(u_i(0), v_i(0))_{i=1}^{P} = A$, $(u_i(1), v_i(1))_{i=1}^{P} = B$. Also, we can see that it is a curve in $\mathcal{P}^*$: all $u_i(t), v_i(t)$ are nonnegative multiples of $\bar{u}_i, \bar{v}_i$, and we know that

$$\sum_{i=1}^{P} D_i X(u_i(t) - v_i(t)) = \sum_{i=1}^{m} F_i(t) a_i + \sum_{j=1}^{k} G_j(t) b_j = y^*.$$

Moreover, the cardinality of $(u_i(t), v_i(t))_{i=1}^{P}$ bounded with

$$\mathrm{card}((u_i(t), v_i(t))_{i=1}^{P}) \leq \sum_{i=1}^{m} \mathbb{1}(F_i(t) > 0) + \sum_{j=1}^{k} \mathbb{1}(G_j(t) > 0) \leq n + 1$$

hence the proposed path becomes a continuous path in $\mathcal{P}^*(n+1)$.

Now, we describe how we find such $m + k$ continuous functions. We do:

**Step 0)** Initialize $\mathcal{C} = \mathcal{A}$, $f_i(0) = \lambda_i, g_i(0) = 0$.
**Step 1)** While $T = 0, 1, \cdots$, repeat:

- If $\mathcal{C} \subseteq \mathcal{B}$, break.

- (Facts that hold from the previous iteration) Let's write $\mathcal{C} = \{a_{i_1}, a_{i_2}, \cdots a_{i_r}\} \cup \{b_{j_1}, b_{j_2}, \cdots b_{j_s}\}$. We inductively have:
  1) $\mathcal{C}$ is a linearly independent set.
  2) $f_i(T) \geq 0 \quad \forall i \in [m], g_j(T) \geq 0 \quad \forall j \in [k]$.
  3) $f_i(T) > 0 \Leftrightarrow i \in \{i_1, i_2, \cdots i_r\}, g_j(T) > 0 \Leftrightarrow j \in \{j_1, j_2, \cdots j_s\}$.
  4)
  $$\sum_{w=1}^{r} f_{i_w}(T) a_{i_w} + \sum_{w=1}^{s} g_{j_w}(T) b_{j_w} = y^*.$$

- (Applying Lemma D.1) We also know that $\sum_{w=1}^{k} \mu_w b_w = y^*$ and $\mu_w > 0 \ \forall w \in [k]$. Now we check the conditions to apply Lemma D.1 with $A := \mathcal{C}$, $B := \mathcal{B}$, given subset $\{a_{i_1}, a_{i_2}, \cdots, a_{i_r}\} \subseteq A$. Then we know that $\mathcal{C}, B$ are linearly independent,

$$\sum_{w=1}^{r} f_{i_w}(T) a_{i_w} + \sum_{w=1}^{s} g_{j_w}(T) b_{j_w} = \sum_{w=1}^{k} \mu_w b_w = y^*,$$

and $\sum_{w=1}^{r} f_{i_w}(T) > 0$, $\mu_w > 0$ for all $w \in [k]$. Thus all conditions for Lemma D.1 are met, and we can find $\tilde{\lambda}_{i_w}$ for $w \in [r]$, $\tilde{\mu}_{j_w}$ for $w \in [s]$, $\mu^*$ that satisfies

$$\|\mu^*\|_0 \le n+1-r-s, \ \ \mu^* \ge 0, \ \ \sum_{w=1}^{r} \tilde{\lambda}_{i_w} a_{i_w} + \sum_{w=1}^{s} \tilde{\mu}_{j_w} b_{j_w} = \sum_{w=1}^{k} \mu_w^* b_w, \ \ \sum_{w=1}^{r} \tilde{\lambda}_{i_w} > 0.$$

- (Update 1) Now we update the values of $f_i, g_j$ as the following:

$$f_i(t) = \begin{cases} f_i(T) = 0 & if \quad i \notin \{i_1, i_2, \cdots i_r\} \\ f_i(T) - \alpha \tilde{\lambda}_i(t - T) & if \quad i \in \{i_1, i_2, \cdots i_r\}, \end{cases} \quad t \in [T, T+1/2],$$

$$g_i(t) = \begin{cases} g_i(T) + \alpha \mu_i^*(t - T) = \alpha \mu_i^*(t - T) & if \quad i \notin \{j_1, j_2, \cdots j_s\} \\ g_i(T) + \alpha \mu_i^*(t - T) - \alpha \tilde{\mu}_i(t - T) & if \quad i \in \{j_1, j_2, \cdots j_s\}, \end{cases} \quad t \in [T, T+1/2].$$

  Here, $\alpha = 2 \min\{\min_{\tilde{\lambda}_{i_w} > 0} f_{i_w}(T)/\tilde{\lambda}_{i_w}, \min_{\tilde{\mu}_{j_w} > \mu_{j_w}^*} g_{j_w}(T)/(\tilde{\mu}_{j_w} - \mu_{j_w}^*)\} > 0$.
  Update $\mathcal{C}$ so that $f_i(T + 1/2) > 0 \Leftrightarrow a_i \in \mathcal{C}$, $g_i(T + 1/2) > 0 \Leftrightarrow b_i \in \mathcal{C}$.

- (Update 2: Pruning) After this, we initialize $r = 0$, $s_i(0) = f_i(T + 1/2), i \in [m]$, $z_j(0) = g_j(T + 1/2), j \in [k]$ and repeat:
  (Check) If $\mathcal{C}$ is linearly independent, break
  (Update) Say $\mathcal{C} = \{a_{r1}, a_{r2}, \cdots a_{rx}\} \cup \{b_{s1}, b_{s2}, \cdots b_{sy}\}$. Find a nontrivial linear combination

$$\sum_{w=1}^{x} \eta_w a_{rw} + \sum_{w=1}^{y} \eta_w' b_{sw} = 0,$$

  and without loss of generality suppose $\sum_{w=1}^{x} \eta_w \ge 0$. If $\sum_{w=1}^{x} \eta_w = 0$, choose $\eta_w'$ to find at least one $\eta_w' > 0$ for $w \in [y]$. Then, write

$$s_{rw}(r + t) = s_{rw}(r) - \alpha \eta_w t, \quad z_{sw}(r + t) = z_{sw}(r) - \alpha \eta_w' t$$

  for $t \in [0, 1]$. Here $\alpha = \min\{\min_{\eta_w > 0} s_{rw}(r)/\eta_w, \min_{\eta_w' > 0} z_{rw}(r)/\eta_w'\}$.
  At last, update $\mathcal{C}$ so that $s_i(r + 1) > 0 \Leftrightarrow a_i \in \mathcal{C}$, $z_i(r + 1) > 0 \Leftrightarrow b_i \in \mathcal{C}$. Increase $r$ by 1.

- (Construct $f_i, g_j$ for $t \in [T + 1/2, T + 1]$) Concatente $f_i$ and $s_i$, $g_j$ and $z_j$ for all $i \in [m]$, $j \in [k]$.

**Step 2)** Let the termination time be $T^*$. To obtain $F_i, G_j : [0, 1] \to \mathbb{R}$, simply write $F_i(t) = f_i(T/T^*), G_j(t) = g_j(T/T^*)$.

Let's first verify that the facts that hold from the previous iteration. First, $\mathcal{C}$ is a linearly independent set at the start of each iterate because of step (Update 2: Pruning), and the first fact holds. Also, $f_i, g_j$ are updated only in steps (Update1) and (Update 2: Pruning), and we can see that for all $t \in [0, T^*]$ the function values are nonnegative. In update 1, we chose $\alpha$ sufficiently small and $\mu^* \ge 0$. In update 2, we also chose $\alpha$ sufficiently small. Hence, the second fact holds. At every update, we also update $\mathcal{C}$, and the third fact holds. At last, we add a nontrivial linear combination of $\mathcal{A} \cup \mathcal{B}$ that sums up to 0 at each update, so the sum

$$\sum_{i=1}^{m} f_i(t) a_i + \sum_{j=1}^{k} g_j(t) b_j,$$

is preserved to be $y^*$, which means that the last fact follows.

One important argument to make is that the algorithm actually terminates, i.e. $T^* < \infty$. The iteration in the second update terminates eventually, because at each iteration the cardinality of $\mathcal{C}$ decreases by 1. To see the larger loop terminating, observe that

$$\sum_{i=1}^{m} f_i(t)$$

is a strictly decreasing function for $t \in \mathbb{N}$. This is because in (Update 1), as $\sum_{w=1}^{r} \tilde{\lambda}_{i_w} > 0$, the sum decreases, and in (Update 2), the sum does not increase because we suppose $\sum_{w=1}^{x} \eta_w \geq 0$. This means that at each starting step of the algorithm, identical $\mathcal{C}$ cannot appear twice: as $\mathcal{C}$ is linearly independent there exists a unique expression that gives

$$\sum_{w=1}^{r} f_{i_w}(T) a_{i_w} + \sum_{w=1}^{s} g_{j_w}(T) b_{j_w} = y^*,$$

and if identical $\mathcal{C}$ appeared twice we will have the same value of $\sum_{w=1}^{r} f_{i_w}(T) = \sum_{i=1}^{m} f_i(t)$, which is contradicting the fact that $\sum_{i=1}^{m} f_i(t)$ strictly decreases.

Finally, let's check that we have found the right $F_i, G_j$s. As the algorithm terminated, $\mathcal{C} \subseteq \mathcal{B}$, and we know that $F_i(1) = 0$ for all $i \in [m]$. Also, as

$$\sum_{j=1}^{k} G_j(1) b_j = y^*,$$

and the set $\mathcal{B}$ is linearly independent, we know that $G_j(1) = \mu_j$ for all $j \in [k]$. As previously mentioned, Property 3) is guaranteed as we are adding a nontrivial linear combination that sums up to 0 at each update. To see that Property 4) is true, we see the value of $\sum_{i=1}^{m} 1(F_i(t) > 0) + \sum_{j=1}^{k} 1(G_j(t) > 0)$ for each update. In (Update 1), as $\|\mu^*\| \leq n+1-t-s$, the total cardinality does not exceed $n+1$. In (Update 2), the cardinality always decreases. Hence, $\sum_{i=1}^{m} 1(F_i(t) > 0) + \sum_{j=1}^{k} 1(G_j(t) > 0) \leq n+1$ for all $t \in [0,1]$, and we know that we have actually found the wanted functions. This finishes the proof. □

Recall that in Nguyen (2021), it is proved that the solution set is connected for $m = n + 1$ in the unregularized case. The proof strategy of Nguyen (2021) is first creating a zero entry in the second layer and changing the corresponding first layer weight arbitrarily. If the network is unregularized this is possible because the change in the corresponding first layer weight where the second layer weight is 0 will not change the model fit, hence the optimality. However, when we have regularization, such transformation is not possible as the first and second layer weights are tied together. This is why we need to use the characterization in Theorem 1 and Lemma D.1 to prove Theorem D.1. Overall, our result is a nontrivial extension of Nguyen (2021) to regularized networks.

At last, from Proposition D.6, we know that $\mathcal{P}^*(m)$ is connected when $m \geq \min\{n+1, m^* + M^*\}$.

**Proposition D.6.** *Suppose $\mathcal{P}^*(m')$ is connected and $m' \geq M^*$. Then $\mathcal{P}^*(m)$ is connected for all $m \geq m'$.*

*Proof.* Take two points $A, B$ from $\mathcal{P}^*(m)$. We know that there exists a path from $A$ to $A_{irr}$, $B$ to $B_{irr}$ that satisfies $A_{irr}, B_{irr} \in \mathcal{P}^*(m) \cap \mathcal{P}^*_{irr}$. Notice that as $A_{irr}, B_{irr} \in \mathcal{P}^*_{irr}$, there cardinality is at most $M^*$. Hence they are elements of $\mathcal{P}^*(m')$, which finishes the proof as $\mathcal{P}^*(m')$ is connected. □

Now we connect the connectivity results of $\mathcal{P}^*(m)$ to that of $\Theta^*(m)$, the solution set of the original problem in equation 3. The object $\Theta^*(m)$ we care about is precisely

$$\Theta^*(m) := \left\{ (w_i, \alpha_i)_{i=1}^{m} \mid \min_{(w_i, \alpha_i)_{i=1}^{m}} L\left(\sum_{i=1}^{m} (Xw_i)_+ \alpha_i, y\right) + \frac{\beta}{2} \sum_{i=1}^{m} (\|w_i\|_2^2 + |\alpha_i|^2) \right\} \subseteq \mathbb{R}^{(d+1)m},$$

We first define essential sets and mappings to do this.

**Definition D.5.** *(Minimal Optimal Neural Networks) We say a parameter $(w_j, \alpha_j)_{j=1}^m$ is minimal optimal if $(w_j, \alpha_j)_{j=1}^m \in \Theta^*(m)$ and $\alpha_p \alpha_q > 0$ implies $1(Xw_p \geq 0) \neq 1(Xw_q \geq 0)$ for all $p \neq q \in [m]$. We denote the set of minimal optimal neural networks as $\Theta_{\min}^*(m)$.*

Adapting the proof from Wang et al. (2021b), we can show that for any point $A \in \Theta^*(m)$, there is a continuous path from $A$ to a point in $\Theta_{\min}^*(m)$. The path is essentially merging the neurons with same arrangement patterns and second layer sign.

**Proposition D.7.** *Take any point $A \in \Theta^*(m)$. We have a continuous path from $A$ to some point $A_{\min} \in \Theta_{\min}^*(m)$ in $\Theta^*(m)$.*

*Proof.* Write $A = (w_j, \alpha_j)_{j=1}^m$. Assume we have $(w_1, \alpha_1)$ and $(w_2, \alpha_2)$ that satisfies $\alpha_1 \alpha_2 > 0$ and $1(Xw_1 \geq 0) = 1(Xw_2 \geq 0)$. Let's write $\text{sign}(\alpha_1) = \text{sign}(\alpha_2) = s$. Define the curve

$$C(t) = \left( \frac{w_1 \alpha_1 + t w_2 \alpha_2}{\sqrt{\|w_1 \alpha_1 + t w_2 \alpha_2\|_2}}, \sqrt{\|w_1 \alpha_1 + t w_2 \alpha_2\|_2} s \right)$$

$$\oplus \left( \sqrt{1-t} \frac{w_2 \alpha_2}{\sqrt{\|w_2 \alpha_2\|_2}}, \sqrt{(1-t)\|w_2 \alpha_2\|_2} s \right) \oplus (w_j, \alpha_j)_{j=3}^m,$$

where $t \in [0, 1]$. The intuition of this curve is merging two pair $(w_1, \alpha_1)$ and $(w_2, \alpha_2)$.
Let's check some basic facts to see that this curve indeed merges the two pairs and is in $\Theta^*(m)$.
i) $C(t)$ is well-defined. First, we know $\|w_2 \alpha_2\|_2 \neq 0$, because $\alpha_2 \neq 0$. Also, say $w_1 \alpha_1 + (1-t)w_2 \alpha_2 = 0$ for some $t \in [0, 1]$. Then we have $DX w_1 \alpha_1 = -(1-t)DX w_2 \alpha_2$, where $D = \text{diag}(1(Xw_1 \geq 0))$. As $DX w_1$ and $DX w_2$ is consisted of nonnegative entries and $\alpha_1 \alpha_2 > 0$, $DX w_1 \alpha_1 = 0$ must hold. This means $w_1 = \alpha_1 = 0$ because $A \in \Theta^*(m)$ - which is again contradiction because $\alpha_1 \neq 0$. The well-definedness of $C(t)$ implies that it is continuous, because it is a composition of continuous functions.
ii) $C(0) = A$, $C(1) = \left( \frac{w_1 \alpha_1 + w_2 \alpha_2}{\sqrt{\|w_1 \alpha_1 + w_2 \alpha_2\|_2}}, \sqrt{\|w_1 \alpha_1 + w_2 \alpha_2\|_2} s \right) \oplus (0, 0) \oplus (w_j, \alpha_j)_{j=3}^m$ from direct substitution. Note that the value $\sum_{i=1}^m 1(\alpha_i \neq 0)$ decreased by 1.
iii) $C(t)$ is a curve in $\Theta^*(m)$. This is because the sum $\sum_{i=1}^m (Xw_i)_+ \alpha_i$ is preserved through the curve, and the regularization loss is less than that of $A$ due to triangular inequality. In other words, the loss $L(C(t)) \leq L(C(0))$ for all $t \in [0, 1]$, and as $L(C(0))$ is optimal, $C(t)$ is a curve in $\Theta^*(m)$.

We repeat the merging process until there is no such pair. This process should terminate because each merging decreases $\sum_{i=1}^m 1(\alpha_i \neq 0)$ by 1. After we don't have such pair, concatenate all the curves that we have to find a curve in $\Theta^*(m)$. At the end of the path, we don't have two $(w_i, \alpha_i)$, $(w_j, \alpha_j)$ that satisfy $\alpha_i \alpha_j > 0$ and $1(Xw_i \geq 0) = 1(Xw_j \geq 0)$, hence it is in $\Theta_{\min}^*(m)$. □

Also, we define the notion of a canonical polytope. A canonical polytope is defined to break ties that occur because two cones $\mathcal{K}_i$ and $\mathcal{K}_j$ may have nonempty intersections. For instance, say $D_1 X u_1 = D_2 X u_1$ and $u_2 = 0$ for some $(u_i, v_i)_{i=1}^P \in \mathcal{P}^*$. Then, swapping $(u_1, v_1)$ and $(u_2, v_2)$ will not change the solution's optimality. As we will see later on, we will want to erase such ambiguity, hence we consider a canonical polytope.

**Definition D.6.** *(Canonical Polytope) The canonical polytope is defined as*

$$\mathcal{P}_{\text{can}}^* = \Big\{ (u_i, v_i)_{i=1}^P \mid (u_i, v_i)_{i=1}^P \in \mathcal{P}^*, \text{diag}(1(Xu_i \geq 0)) = D_i \text{ if } u_i \neq 0,$$

$$\text{diag}(1(Xv_i \geq 0)) = D_i \text{ if } v_i \neq 0 \Big\}.$$

**Remark D.2.** $\text{diag}(1(Xu \geq 0)) = D_i$ *implies* $(2D_i - I)Xu \geq 0$*, but not the opposite. The ambiguity happens because $x_j \cdot u$ might be 0 for some rows.*

Given the notion of the minimal optimal neural network and the canonical polytope, we define two natural mappings $\Psi : \mathcal{P}^*(m) \to \Theta^*(m)$ and $\Phi : \Theta^*(m) \to \mathcal{P}^*(m)$. These mappings have been discussed multiple times in the literature Pilanci & Ergen (2020), Wang et al. (2021b), and we introduce it again with slight variations for our needs.

**Definition D.7.** *Suppose $m \geq m^*$. We define $\Psi : \mathcal{P}^*(m) \to \Theta^*(m)$ as*

$$\Psi((u_i, v_i)_{i=1}^P) := \left( \frac{u_i}{\sqrt{\|u_i\|_2}}, \sqrt{\|u_i\|_2} \right)_{u_i \neq 0} \oplus \left( \frac{v_i}{\sqrt{\|v_i\|_2}}, -\sqrt{\|v_i\|_2} \right)_{v_i \neq 0} \oplus (0, 0)^{m - \text{card}((u_i, v_i)_{i=1}^P)},$$

**Definition D.8.** *Suppose $m \geq m^*$. We define $\Phi : \Theta^*(m) \to \mathcal{P}^*(m)$ as*

$$\Phi((w_i, \alpha_i)_{i=1}^m) = (u_i, v_i)_{i=1}^P := \begin{cases} u_p = \sum_{i \in \mathcal{I}} w_i |\alpha_i| \ where \ \mathcal{I} = \{i \mid \alpha_i > 0, D_p = \mathrm{diag}(1(Xw_i \geq 0))\} \\ v_q = \sum_{i \in \mathcal{I}} w_i |\alpha_i| \ where \ \mathcal{I} = \{i \mid \alpha_i < 0, D_q = \mathrm{diag}(1(Xw_i \geq 0))\}. \end{cases}$$

The mappings are indeed well-defined Proposition D.8.

**Proposition D.8.** *Suppose $m \geq m^*$. $\Psi : \mathcal{P}^*(m) \to \Theta^*(m)$ and $\Phi : \Theta^*(m) \to \mathcal{P}^*(m)$ are well defined.*

*Proof.* By well-defined, we want to see that for all $A \in \mathcal{P}^*(m)$, $\Psi(A) \in \Theta^*(m)$ and has a unique value, and similarly for $\Phi$ too.
From Definition D.7 and Definition D.8, it is not hard to see that the function value is uniquely determined for each input. Also, from direct calculation, we can see that

$$L\left(\sum_{j=1}^m (Xw_j)_+ \alpha_j, y\right) + \frac{\beta}{2} \sum_{j=1}^m \left(\|w_j\|_2^2 + \alpha_j^2\right) = L\left(\sum_{i=1}^P D_i X(u_i - v_i), y\right) + \beta \sum_{i=1}^P \left(\|u_i\|_2 + \|v_i\|_2\right),$$

for both $A = (u_i, v_i)_{i=1}^P$ and $\Psi(A) = (w_j, \alpha_j)_{j=1}^m$ and when $A = (w_j, \alpha_j)_{j=1}^m$ and $\Phi(A) = (u_i, v_i)_{i=1}^P$. The former case is rather clear. To see the latter case, first observe that when we apply the merging operation in Proposition D.7, the loss will strictly decrease if for two $w_i, w_j$ with same arrangement pattern weren't parallel. So they are actually parallel, and for all $i \in \mathcal{I}$ such that $D_p = \mathrm{diag}(1(Xw_i \geq 0))$,

$$\|u_p\|_2 = \sum_{i \in \mathcal{I}} \|w_i\|_2 |\alpha_i| = \frac{1}{2} \sum_{i \in \mathcal{I}} (\|w_i\|_2^2 + \alpha_i^2),$$

because all $w_i$ are parallel for $i \in \mathcal{I}$. The last equality follows from $A \in \Theta^*(m)$. As $m \geq m^*$ the optimization problems in equation 3 and equation 4 have same optimal values, which means $\Psi(A) \in \Theta^*(m)$ and $\Phi(A) \in \mathcal{P}^*(m)$. □

Moreover, we can see that the two mappings are similar to inverses of each other.

**Proposition D.9.** *Take any $A \in \mathcal{P}^*_{\mathrm{can}} \cap \mathcal{P}^*(m)$. Then, $\Phi(\Psi(A)) = A$. Also, take any $B = (w_j, \alpha_j)_{j=1}^m \in \Theta^*_{\min}(m)$. Then, $\Psi(\Phi(B)) = (w_{\sigma(j)}, \alpha_{\sigma(j)})_{j=1}^m$ a permutation of $B$.*

*Proof.* We know

$$\Psi((u_i, v_i)_{i=1}^P) := (\frac{u_i}{\sqrt{\|u_i\|_2}}, \sqrt{\|u_i\|_2})_{u_i \neq 0} \oplus (\frac{v_i}{\sqrt{\|v_i\|_2}}, -\sqrt{\|v_i\|_2})_{v_i \neq 0} \oplus (0, 0)^{m - \mathrm{card}((u_i, v_i)_{i=1}^P)}.$$

Write $\Phi(\Psi((u_i, v_i)_{i=1}^P)) = (u_i', v_i')_{i=1}^P$. Let s see that $u_i' = u_i$ for all $i \in [P]$. The case of $v$ will follow similarly.
The first case is when $u_i = 0$. Say there exists $u_j \neq 0$ and $\mathrm{diag}(1(Xu_j \geq 0)) = D_i$. As $(u_i, v_i)_{i=1}^P \in \mathcal{P}^*_{can}$, $\mathrm{diag}(1(Xu_j \geq 0)) = D_j = D_i$, meaning $i = j$. This is a contradiction because $u_i = 0$. This means there is no $u_j \neq 0$ that is $1(Xu_j \geq 0) = D_i$, and there is no $u_j \neq 0$ that is $\mathrm{diag}(1(Xu_j / \sqrt{\|u_j\|_2} \geq 0)) = D_i$, meaning $u_i' = 0$.
The next case is when $u_i \neq 0$. For $u_j \neq 0$ such that $\mathrm{diag}(1(Xu_j \geq 0)) = D_i$, the only possible $j = i$. For that $j$, we know that $\mathrm{diag}(1(Xu_i / \sqrt{\|u_i\|_2} \geq 0)) = D_i$, and $u_i' = u_i / \sqrt{\|u_i\|_2} \times \sqrt{\|u_i\|_2} = u_i$. This means $u_i' = u_i$ for all $i \in [P]$, same for $v$, meaning $\Phi(\Psi((u_i, v_i)_{i=1}^P)) = (u_i, v_i)_{i=1}^P$.
Let's see $\Psi \circ \Phi$. We know

$$\Phi((w_i, \alpha_i)_{i=1}^m) = (u_i, v_i)_{i=1}^P := \begin{cases} u_p = w_i |\alpha_i| \ if \ \alpha_i > 0 \ and \ D_p = \mathrm{diag}(1(Xw_i \geq 0)), \ 0 \ otherwise \\ v_q = w_i |\alpha_i| \ if \ \alpha_i < 0 \ and \ D_q = \mathrm{diag}(1(Xw_i \geq 0)), \ 0 \ otherwise, \end{cases}$$

because $(w_i, \alpha_i)_{i=1}^m$ is minimal. Let's say $\sum_{i=1}^m 1(\alpha_i > 0) = m_p$, $\sum_{i=1}^m 1(\alpha_i = 0) = m_z$, $\sum_{i=1}^m 1(\alpha_i < 0) = m_n$. In $\{u_1, u_2, \cdots u_P\}$, there will be $m_p$ nonzero vectors. Index them as $u_{a_1}, u_{a_2}, \cdots u_{a_{m_p}}$. For $u_{a_i}$, we can find $j_i \in [m]$ that satisfies $u_{a_i} = w_{j_i} |\alpha_{j_i}|$. Furthermore,

$j_{i_1} \neq j_{i_2}$ if $i_1 \neq i_2$ because $j_{i_1} = j_{i_2}$ means $D_{a_{i_1}} = D_{a_{i_2}}$ and $a_{i_1} = a_{i_2}$, $i_1 = i_2$. Similarly, define $v_{b_1}, v_{b_2}, \cdots v_{b_{m_n}}$ and $v_{b_i} = w_{k_i}|\alpha_{k_i}|$. Then,

$$\Psi(\Phi((w_i, \alpha_i)_{i=1}^m)) = \left( \frac{w_{j_i}|\alpha_{j_i}|}{\sqrt{w_{j_i}|\alpha_{j_i}|}}, \sqrt{w_{j_i}|\alpha_{j_i}|} \right)_{i=1}^{m_p} \oplus \left( \frac{w_{k_i}|\alpha_{k_i}|}{\sqrt{w_{k_i}|\alpha_{k_i}|}}, -\sqrt{w_{k_i}|\alpha_{k_i}|} \right)_{i=1}^{m_n} \oplus (0,0)^{m_z}.$$

First, we know that $\|w_j\|_2 = |\alpha_j|$ for all $j \in [m]$. This leads to

$$\Psi(\Phi((w_i, \alpha_i)_{i=1}^m)) = (w_{j_i}, |\alpha_{j_i}|)_{i=1}^{m_p} \oplus (w_{k_i}, -|\alpha_{k_i}|)_{i=1}^{m_n} \oplus (0,0)^{m_z}.$$

As $j_{i_1} \neq j_{i_2}$ if $i_1 \neq i_2$, the result is a permutation of $(w_i, \alpha_i)_{i=1}^m$. $\qquad \square$

At last, for these mappings to be meaningful, we would want them to be continuous. Luckily, $\Phi$ is continuous Proposition D.10.

**Proposition D.10.** *The map* $\Phi : \Theta^*(m) \to \mathcal{P}^*(m)$ *is continuous.*

*Proof.* We consider the sequence $(w_j^k, \alpha_j^k)_{j=1}^m$ in $\Theta^*(m)$ that converges to $(w_j^\infty, \alpha_j^\infty)_{j=1}^m \in \Theta^*(m)$. Let's write

$$\Phi((w_j^k, \alpha_j^k)_{j=1}^m) = (u_i^k, v_i^k)_{i=1}^P, \quad \Phi((w_j^\infty, \alpha_j^\infty)_{j=1}^m) = (u_i^\infty, v_i^\infty)_{i=1}^P.$$

We will show that $u_i^k \to u_i^\infty$. The rest will follow.

As a starting point, we define some necessary constants. We define $M_j$ for $j \in [m]$ that satisfy $w_j^\infty \neq 0$ as the following: if $k \geq M_j$, $\mathbf{1}(Xw_j^\infty \geq 0) = \mathbf{1}(Xw_j^k \geq 0)$ and $\alpha_j^k \alpha_j^\infty > 0$. Such $M_j$ exists due to the following reasoning: we know that for any solution $A \in \Theta^*(m)$, there exists a finite set of possible directions for $w_j$, which are the directions of $\bar{u}_i, \bar{v}_i$ in $\mathcal{P}^*$. As $w_j^\infty \neq 0$ and $w_j^k \to w_j^\infty$, for sufficiently large $k$ so that $\|w_j^k - w_j^\infty\|_2$ is sufficiently small, $w_j^\infty$ has to be a positive scaling of $w_j^k$. Also $w_j^\infty \neq 0$ implies $\alpha_j^\infty \neq 0$, meaning for sufficiently large $k$, $\alpha_j^k \alpha_j^\infty > 0$ holds. For $j \in [m]$ that has $w_j^\infty = 0$, define $N_j(\epsilon)$ to be the number that satisfies $k \geq N_j(\epsilon)$ implies $\|w_j^k \alpha_j^k\|_2 \leq \epsilon$.

Now we prove that for sufficiently large $k$, $\|u_i^k - u_i^\infty\|_2 \leq \epsilon$ for all $i \in [P]$. For a certain $i \in [P]$, suppose there exists $\{j_1, j_2, \cdots j_t\} \subseteq [m]$ that satisfies $D_i = \text{diag}(\mathbf{1}(Xw_{j_1}^\infty \geq 0)) = \cdots = \text{diag}(\mathbf{1}(Xw_{j_t}^\infty \geq 0))$ and $\alpha_{j_1}^\infty, \cdots, \alpha_{j_t}^\infty > 0$ (hence $w_{j_1}^\infty, \cdots, w_{j_t}^\infty \neq 0$). It is clear that $u_i^\infty = \sum_{i=1}^t w_{j_i}^\infty \alpha_{j_i}^\infty$. When $k \geq \max\{\max_{w_j^\infty = 0} N_j(\epsilon/m), \max_{w_j^\infty \neq 0} M_j\}$, we know that $\mathbf{1}(Xw_{j_i}^k \geq 0) = \mathbf{1}(Xw_{j_i}^\infty \geq 0)$ and $\alpha_{j_i}^k > 0$ for $i \in [t]$. Also, for some $j \in [m]$ which is not in $\{j_1, j_2, \cdots, j_t\}$ and $D_i = \text{diag}(\mathbf{1}(Xw_j^k \geq 0))$, $w_j^\infty = 0$. Hence,

$$u_i^k = \sum_{i=1}^t w_{j_i}^k \alpha_{j_i}^k + \sum_{w_j^\infty = 0, D_i = \text{diag}(\mathbf{1}(Xw_j^k \geq 0)), \alpha_j^k > 0} w_j^k \alpha_j^k.$$

$u_i^k \to u_i^\infty$, as $w_{j_i}^k \to w_{j_i}^\infty, \alpha_{j_i}^k \to \alpha_{j_i}^\infty$ for $i \in [t]$ and the rest sum becomes smaller than $\epsilon$, hence converging to 0 as $k \to \infty$.

Finally, let's see the case where there is no $j \in [m]$ that satisfies $D_i = \text{diag}(\mathbf{1}(Xw_j^\infty \geq 0))$ and $\alpha_j^\infty > 0$. Here, $u_i^\infty = 0$. Now take $k \geq \max\{\max_{w_j^\infty = 0} N_j(\epsilon/m), \max_{w_j^\infty \neq 0} M_j\}$. One thing to notice is for this $k$, if $D_i = \text{diag}(\mathbf{1}(Xw_j^k \geq 0))$ and $\alpha_j^k > 0$ for some $j \in [m]$, $w_j^\infty = 0$. Suppose $w_j^\infty \neq 0$. As $k \geq M_j$, we know that $D_i = \text{diag}(\mathbf{1}(Xw_j^\infty \geq 0))$ and $\alpha_j^\infty > 0$, which contradicts the assumption that there is no such $j$. Hence, when we write

$$u_i^k = \sum_{w_j^\infty = 0, D_i = \text{diag}(\mathbf{1}(Xw_j^k \geq 0)), \alpha_j^k > 0} w_j^k \alpha_j^k,$$

as $k \geq N_j(\epsilon/m)$, $\|u_i^k\|_2 \leq \epsilon$. As $u_i^\infty = 0$, we have that $u_i^k \to u_i^\infty$. This finishes the proof. $\qquad \square$

However, $\Psi$ may not be continuous. The intuition is that the solutions in the image of $\Psi$ have zeros at the end, whereas the limit of $\Psi((u_i, v_i)_{i=1}^P)$ as $u_i \to 0$ may have zeros at the middle. Thus we have a slightly weaker notion of continuity for $\Psi$ Proposition D.11.

**Proposition D.11.** *Consider a continuous path $(u_i(t), v_i(t))_{i=1}^P$ in $\mathcal{P}^*(m)$, where $u_i(t), v_i(t)$ : $[0,1] \to \mathbb{R}^d$ is either a zero map or a map can only have zero when $t = 1$. Consider the path $\phi(t) = \Psi((u_i(t), v_i(t))_{i=1}^P)$ in $\Theta^*(m)$. Then, $\phi(t)$ is continuous in $[0,1)$, and $\lim_{t \to 1} \phi(t)$ is a permutation of $\phi(1)$.*

*Proof.* Let's write what $\phi(t)$ looks like. Write $i_1 < i_2 < \cdots i_p$ the indices where $u_i(0) \neq 0$, and write $j_1 < j_2 < \cdots < j_q$ the indices where $v_i(0) \neq 0$. Denote the indices $\mathcal{I}, \mathcal{J}$. For $t \in [0,1)$, $\phi(t)$ is

$$\phi(t) = \left( \frac{u_i(t)}{\sqrt{\|u_i(t)\|_2}}, \sqrt{\|u_i(t)\|_2} \right)_{i \in \mathcal{I}} \oplus \left( \frac{v_i(t)}{\sqrt{\|v_i(t)\|_2}}, -\sqrt{\|v_i(t)\|_2} \right)_{i \in \mathcal{J}} \oplus (0,0)^{m-p-q}.$$

As $\mathcal{I}, \mathcal{J}$ is fixed for $t \in [0,1)$ and $u_i(t) \neq 0$ if $u_i(0) \neq 0$, $v_i(t) \neq 0$ if $v_i(0) \neq 0$, $\phi$ is continuous for [0,1). When $t = 1$, $\lim_{t \to 1} \phi(t)$ may have zeros in the middle, whereas $\phi(1)$ has zeros at the end, and the rest is the same. Hence, $\lim_{t \to 1} \phi(t)$ is a permutation of $\phi(1)$. □

Given the machinery, we are ready to elaborate the results. We start with proving that if $m$ is sufficiently large, all permutations of a point $A \in \Theta^*(m)$ are connected. The proof strategy is analogous to the proof in Simsek et al. (2021), where we create an empty slot to permute the weights.

**Proposition D.12.** *Suppose $m \geq M^* + 1$. Take any $(w_j, \alpha_j)_{j=1}^m \in \Theta^*(m)$. There exists a continuous path from $(w_j, \alpha_j)_{j=1}^m$ to an arbitrary permutation $(w_{\sigma(j)}, \alpha_{\sigma(j)})_{j=1}^m$.*

*Proof.* Our proof will start from showing that for any $A = (w_j, \alpha_j)_{j=1}^m \in \Theta^*(m)$, we can find a continuous path $A' = (w_j', \alpha_j')_{j=1}^m \in \Theta^*(m)$ that satisfies $\sum_{j=1}^m 1(\alpha_j' \neq 0) < m$. First, use Proposition D.7 to find a continuous path from $A$ to some $A_{\min} = (w_j^\circ, \alpha_j^\circ) \in \Theta_{\min}^*(m)$. If $\sum_{j=1}^m 1(\alpha_j^\circ \neq 0) < m$, we have found such path.

If not, let's show that $\{(Xw_j^\circ)_+\}_{j=1}^m$ is linearly dependent. As all $\alpha_j^\circ \neq 0$, all $w_j^\circ \neq 0$. Now think of $\Phi(A_{\min}) = (u_i, v_i)_{i=1}^P$. We can easily see that

$$\{(Xw_j^\circ)_+ \alpha_j^\circ\}_{j=1}^m = \{D_i X u_i\}_{u_i \neq 0} \cup \{-D_i X v_i\}_{v_i \neq 0}.$$

As the latter set has $m > M^*$ elements, it should be linearly dependent. If not, it is a contradiction to the fact that the maximal cardinality of the element in $\mathcal{P}_{\text{irr}}^*$ is $M^*$. Hence, the set $\{(Xw_j^\circ)_+ \alpha_j^\circ\}_{j=1}^m$ is linearly dependent, and as all $\alpha_j^\circ$ is nonzero, the set $\{(Xw_j^\circ)_+\}_{j=1}^m$ is linearly dependent. Now consider a nontrivial linear combination,

$$\sum_{i=1}^m c_i (Xw_i^\circ)_+ = 0.$$

Without loss of generality say $\alpha_1^\circ c_1 < 0$. Define

$$t_m = \min_{\alpha_i^\circ c_i < 0} -\frac{\alpha_i^\circ}{c_i},$$

and for $t \in [0, t_m]$ define

$$\tilde{w}_i(t) = w_i^\circ \sqrt{\frac{|\alpha_i^\circ + tc_i|}{\|w_i^\circ\|_2}}, \quad \tilde{\alpha}_i(t) = \sqrt{\|w_i^\circ\|_2 |\alpha_i^\circ + tc_i|} \, sign(\alpha_i^\circ).$$

From the definition of $t_m$, $sign(\alpha_i^\circ + tc_i) = sign(\alpha_i^\circ)$ for $t \in [0, t_m]$. Also,

$$\sum_{i=1}^m (X\tilde{w}_i(t))_+ \tilde{\alpha}_i(t) = \sum_{i=1}^m (Xw_i^\circ)_+ (\alpha_i^\circ + tc_i) = \sum_{i=1}^m (Xw_i^\circ)_+ \alpha_i^\circ,$$

and

$$\frac{1}{2} \sum_{i=1}^m \|\tilde{w}_i(t)\|_2^2 + |\tilde{\alpha}_i(t)|^2 = \sum_{i=1}^m \|\tilde{w}_i(t)\|_2 |\tilde{\alpha}_i(t)| = \sum_{i=1}^m \|w_i^\circ\|_2 |\alpha_i^\circ| + \|w_i^\circ\|_2 tc_i sign(\alpha_i^\circ).$$

At last, we know that for the dual optimum $\nu^*$ defined in Theorem 1,

$$(\nu^*)^T (Xw_j^\circ)_+ = -\beta \|w_j^\circ\|_2 sign(\alpha_j^\circ),$$

for all $j \in [m]$. This is obtained by using the fact that $\Phi(A_{\min}) \in \mathcal{P}^*$. Hence, if $\sum_{i=1}^m c_i (Xw_i^\circ)_+ = 0$, multiplying $(\nu^*)^T$ on both sides leads

$$\sum_{i=1}^m \|w_i^\circ\|_2 t c_i sign(\alpha_i^\circ) = 0,$$

and

$$\frac{1}{2} \sum_{i=1}^m \|\tilde{w}_i(t)\|_2^2 + |\tilde{\alpha}_i(t)|^2 = \sum_{i=1}^m \|w_i^\circ\|_2 |\alpha_i^\circ| = \frac{1}{2} \sum_{i=1}^m \|w_i^\circ\|_2^2 + |\alpha_i^\circ|^2.$$

Hence the objective is preserved throughout the curve, meaning the curve is in $\Theta^*(m)$. The cardinality decreased by at least 1 at the end due to the definition of $t_m$.

Now that we can find a continuous path from $A$ to $A' = (w_j', \alpha_j')_{j=1}^m \in \Theta^*(m)$ where $\sum_{j=1}^m 1(\alpha_j' \neq 0) < m$, we will find a path from $A'$ to any permutation of $A'$, namely $(w_{\sigma(j)}', \alpha_{\sigma(j)}')_{j=1}^m$ for some permutation $\sigma : [m] \to [m]$. A simple path construction is as follows: we know that at least one $\alpha_i' = 0$. Let that $i = m$ without loss of generality. Starting from $i_0 = 1$, we do the following: if $w_{i_0}' = w_{\sigma(i_0)}'$, we do nothing. If $w_{i_0}' \neq w_{\sigma(i_0)}'$, we first write

$$w_{i_0}'(t) = w_{i_0}'\sqrt{1-t}, \alpha_{i_0}(t) = \alpha_{i_0}'\sqrt{1-t}, w_m'(t) = w_{i_0}'\sqrt{t}, \alpha_m'(t) = \alpha_{i_0}'\sqrt{t},$$

for $t \in [0,1]$, which intuitively 'moves' $w_{i_0}'$ to the empty space $w_m$ and making $w_{i_0}' = 0$. Next we move $w_{\sigma(i_0)}'$ to $w_i'$ with

$$w_{i_0}'(t) = w_{\sigma(i_0)}'\sqrt{t}, \alpha_{i_0}(t) = \alpha_{\sigma(i_0)}'\sqrt{t}, w_{\sigma(i_0)}'(t) = w_{\sigma(i_0)}'\sqrt{1-t}, \alpha_{\sigma(i_0)}'(t) = \alpha_{\sigma(i_0)}'\sqrt{1-t},$$

for $t \in [0,1]$, which intuitively 'moves' $w_{\sigma(i_0)}'$ to the empty space $w_{i_0}$ and making $w_{\sigma(i_0)} = 0$. At last, we make $w_m$ empty by using

$$w_{\sigma(i_0)}'(t) = w_{i_0}'\sqrt{t}, \alpha_{\sigma(i_0)}(t) = \alpha_{i_0}'\sqrt{t}, w_m'(t) = w_{i_0}'\sqrt{1-t}, \alpha_m'(t) = \alpha_{i_0}'\sqrt{1-t}.$$

To wrap up, we may swap the element in $(w_i, \alpha_i)$ and $(w_{\sigma(i)}, \alpha_{\sigma(i)})$ by first moving $w_i$ to $w_m$, then moving $w_\sigma(i)$ to $w_i$, and at last moving $w_m$ to $w_{\sigma(i)}$.

Until here we connected $A = (w_j, \alpha_j)_{j=1}^m$ with $A'$, and then $A'$ with a permutation of $A'$. To connect $A$ with $(w_{\sigma(j)}, \alpha_{\sigma(j)})$, simply run the path $A \to A'$ backwards to obtain $(w_{\sigma(j)}, \alpha_{\sigma(j)})_{j=1}^m$. □

Proposition D.12 enables us to connect two different permutations. Even though $\Psi$ is not essentially continuous, the fact that two permutations are connected will allow us to construct paths in $\Theta^*(m)$ from paths in $\mathcal{P}^*(m)$.

**Proposition D.13.** *Suppose $m \geq M^* + 1$. If any two points $A, B \in \mathcal{P}^*(m)$ are connected with a path with finite cardinality changes, $\Theta^*(m)$ is connected.*

*Proof.* Take two points $A, B \in \Theta^*(m)$. First use Proposition D.7 to find path from $A$ to $A_{\min} \in \Theta_{\min}^*(m)$ and $B$ to $B_{\min} \in \Theta_{\min}^*(m)$. Our main goal will be connecting $A_{\min}$ and $B_{\min}$ by using the path from $\Phi(A_{\min})$ to $\Phi(B_{\min})$.

Consider the continuous path from $\Phi(A_{\min})$ to $\Phi(B_{\min})$, namely $f : [0,1] \to \mathcal{P}^*(m)$ satisfying $f(0) = \Phi(A_{\min})$, $f(1) = \Phi(B_{\min})$. Write $f(t) = (u_i(t), v_i(t))_{i=1}^P$. Divide $[0,1]$ to times $(t_0 = 0, t_1), (t_1, t_2) \cdots (t_{k-1}, t_k = 1)$, where in each time interval either each $u_i, v_i$ are either always zero or always nonzero. We have finitely many such $t_i$s because we assume that the cardinality change is finite. From Proposition D.11, we can see that the path $\Psi \circ f(t)$ is continuous at each interval. However, as we saw in Proposition D.11, $\Psi \circ f(t_i)$, $\lim_{t \to t_i^-} \Psi \circ f(t)$, $\lim_{t \to t_i^+} \Psi \circ f(t)$ are all permutations of each other.

We construct a path from $\Psi \circ f(0)$ to $\Psi \circ f(1)$ as following: First, for each $p = 0, 1, \cdots, k-1$, construct a path from $\lim_{t \to t_p^+} \Psi \circ f(t)$ to $\lim_{t \to t_{p+1}^-} \Psi \circ f(t)$ by defining

$$g(t) = \begin{cases} \lim_{t \to t_p^+} \Psi \circ f(t) & if \quad t = t_p \\ \Psi \circ f(t) & if \quad t \in (t_p, t_{p+1}) \\ \lim_{t \to t_{p+1}^-} \Psi \circ f(t) & if \quad t = t_{p+1}, \end{cases}$$

for $t \in [t_p, t_{p+1}]$. It is clear that $g$ is continuous. Moreover, we can connect each $\Psi \circ f(t_p)$ with $\lim_{t \to t_p^+} \Psi \circ f(t)$ and $\lim_{t \to t_p^-} \Psi \circ f(t)$, because from Proposition D.12, we know that when $m \geq M^* + 1$, two permutations are connected in $\Theta^*(m)$, and from Proposition D.11, from this construction, a one-sided limit is a permutation of the image. Hence, a connection from $\Psi \circ f(0)$ to $\Psi \circ f(1)$ is possible, by connecting permutations at boundaries of each interval $t_0, t_1, \cdots, t_k$, and moving with $\Psi \circ f$ inside the intervals.

Hence, $\Psi \circ \Phi(A_{\min})$ and $\Psi \circ \Phi(B_{\min})$ are connected. From Proposition D.9, we know that $\Psi \circ \Phi(A_{\min})$ is a permutation of $A_{\min}$, and as $m \geq M^* + 1$ we know that they are connected. Same holds for $\Psi \circ \Phi(B_{\min})$. This means that we have found a continuous path from $A_{\min}$ to $B_{\min}$. At the beginning we connected $A$ with $A_{\min}$, $B$ with $B_{\min}$, which finishes the proof. $\qquad \square$

For discontinuity, we use the property of isolated points $A \in \mathcal{P}_{\text{can}}^* \cap \mathcal{P}^*(m)$ with cardinality $m$.

**Proposition D.14.** *Suppose $m \geq m^*$. If $A \in \mathcal{P}^*(m) \cap \mathcal{P}_{\text{can}}^*$ is an isolated point in $\mathcal{P}^*(m)$ and has $\text{card}(A) = m$, $\Psi(A)$ is an isolated point in $\Theta^*(m)$.*

*Proof.* Assume the existence of a continuous function $f : [0, 1] \to \Theta^*(m)$ that satisfies $\Psi(A) = f(0)$, $\Phi \circ f(1) \neq A$. Consider the path $\Phi \circ f(t)$ in $\mathcal{P}^*(m)$. As $A \in \mathcal{P}^*(m) \cap \mathcal{P}_{\text{can}}^*$, $\Phi(\Psi(A)) = A \neq \Phi \circ f(1)$, which is a contradiction that $A$ is an isolated point in $\mathcal{P}^*(m)$. Hence, $\Psi(A)$ does not have a path into $\Theta^*(m) - \Phi^{-1}(A)$.

To finish the proof, we show that $\Phi^{-1}(A)$ is finite. If $\Phi^{-1}(A)$ is finite, we will not be able to move from $\Psi(A)$ to a point in $\Phi^{-1}(A) - \{\Psi(A)\}$ only using points in $\Phi^{-1}(A) - \{\Psi(A)\}$, meaning we do not have a path from $\Psi(A)$ to $\Theta^*(m) - \{\Psi(A)\}$, proving our claim.

Suppose $\Phi((w_i, \alpha_i)_{i=1}^m) = A$. If $\text{diag}(1(Xw_i \geq 0)) = \text{diag}(1(Xw_j \geq 0)) = D_p$ for some $\alpha_i \alpha_j > 0$, the two indices $i$ and $j$ will either correspond to the same $u_p$ or $v_p$, and $\text{card}(\Phi((w_i, \alpha_i)_{i=1}^m)) < m$, which is a contradiction. Hence, $(w_i, \alpha_i)_{i=1}^m \in \Theta_{\min}^*(m)$. From Proposition D.9, we know that $\Psi(\Phi((w_i, \alpha_i)_{i=1}^m)) = \Psi(A)$ is a permutation of $(w_i, \alpha_i)_{i=1}^m$. This means $\Phi^{-1}(A)$ is contained in a set of permutation of $\Psi(A)$, which is finite. $\qquad \square$

Proposition D.13 and Proposition D.14 enables us to discuss connectivity of $\Theta^*(m)$ with the connectivity of $\mathcal{P}^*(m)$. With the results that we obtained for $\mathcal{P}^*(m)$, more specifically Proposition D.2, Proposition D.3, Proposition D.5, Theorem D.1, and applying Proposition D.13 and Proposition D.14 appropriately, we arrive at the staircase of connectivity defined in Theorem 2.

**Theorem D.2.** *(Theorem 2 of the paper) (The staircase of connectivity) Denote the optimal solution set of equation 3 in parameter space as $\Theta^*(m) \subseteq \mathbb{R}^{(d+1)m}$. Suppose $L$ is a strictly convex loss function and there exists $(w_i, \alpha_i)_{i=1}^m \neq (0, 0)_{i=1}^m \in \Theta^*(m)$ for some $m$. Let $m^*, M^*$ be two critical values defined in Theorem 2.*
*As $m$ changes, we have that when*

  (i) *$m = m^*$, $\Theta^*(m)$ is a finite set. Hence, for any two optimal points $A \neq A' \in \Theta^*(m)$, there is no path from $A$ to $A'$ inside $\Theta^*(m)$.*

  (ii) *$m \geq m^* + 1$, there exists optimal points $A, A' \in \Theta^*(m)$ and a path in $\Theta^*(m)$ connecting them.*

  (iii) *$m = M^*$, $\Theta^*(m)$ is not a connected set. Moreover, there exists $A \in \Theta^*(m)$ which is an isolated point, i.e. there is no path in $\Theta^*(m)$ that connects $A$ with $A' \neq A \in \Theta^*(m)$.*

  (iv) *$m \geq M^* + 1$, permutations of the solution are connected. Hence, for all $A \in \Theta^*(m)$, there exists $A' \neq A$ in $\Theta^*(m)$ and a path in $\Theta^*(m)$ that connects $A$ and $A'$.*

  (v) *$m \geq \min\{m^* + M^*, n + 1\}$, the set $\Theta^*(m)$ is connected, i.e. for any two optimal points $A \neq A' \in \Theta^*(m)$, there exists a continuous path from $A$ to $A'$.*

*Proof.* Proof of i) starts by observing that $A \in \Theta_{\min}^*(m^*)$ if $A \in \Theta^*(m^*)$. If not, we can find a solution $A' \in \Theta_{\min}^*(m^* - 1)$ using Proposition D.7, and its image $\Phi(A') \in \mathcal{P}^*(m^* - 1)$. At last, $\Phi(A')$ is connected with a point in $\mathcal{P}^*(m^* - 1) \cap \mathcal{P}_{\text{irr}}^*$ using Proposition D.4, which contradicts the minimality of $m^*$. Now, for any $A \in \Theta^*(m^*)$, $\Phi(A) = B \in \mathcal{P}^*(m^*)$ satisfies that $\Psi(B)$ is a permutation of $A$. Hence, $\Theta^*(m^*)$ is contained in the set of permutations of $\Psi(\mathcal{P}^*(m^*))$, and as

$\mathcal{P}^*(m^*)$ is finite from Proposition D.2, we know that $\Theta^*(m^*)$ is also finite.

Proof of ii) is rather simple: we know that the solution in $\Theta^*(m^*)$ will have a zero slot in $\Theta^*(m^*+1)$, and by using the moving operation in the proof of Proposition D.12, we can show that we can move a neuron to the zero slot.

Proof of iii) follows from Proposition D.3 and Proposition D.14. We can find an isolated point with cardinality $M^*$ and is also canonical: the isolated point in Proposition D.3 has cardinality $M^*$ and is in $\mathcal{P}^*_{\text{irr}}$. Say the isolated point is $(u_i, v_i)_{i=1}^P$. If $\text{diag}(1(Xu_i \geq 0)) = \text{diag}(1(Xu_j \geq 0)) = D_p$ for some $i \neq j$ and $u_i \neq 0, u_j \neq 0$, $D_p X u_i$ and $D_p X u_j$ are colinear, which leads to a contradiction. Hence all patterns are different for $u, v$, and by appropriate rearrangement, we can find a canonical solution with cardinality $M^*$. Say that solution is $P_{\text{can}}$ Now we apply Proposition D.14 to see that $\Psi(P_{\text{can}})$ is isolated in $\Theta^*(M^*)$.

Proof of iv) almost directly follows from Proposition D.12. Note that when $w_1 = w_2 = \cdots w_m$, we can prune the solution to connect it with a different solution, hence there is no isolated point.

Proof of v) directly follows from Proposition D.5, Theorem D.1 and Proposition D.13. Note that the paths constructed in Proposition D.5 and Theorem D.1 has only finitely many cardinality changes, hence we can apply Proposition D.13. The solution set is not a singleton because $m \geq m^* + 1$. $\quad\square$

**Corollary D.1.** *(Corollary 2 of the paper) Consider the optimization problem in equation 3. Suppose $m \geq n+1$ and denote the optimal set of equation 3 as $\Theta^*(m)$. For any $\theta := (w_i, \alpha_i)_{i=1}^m \in \mathbb{R}^{(d+1)m}$, there exists a continuous path from $\theta$ to any point $\theta^* \in \Theta^*(m)$ with nonincreasing loss.*

*Proof.* The proof almost directly follows from Haeffele & Vidal (2017) and Theorem D.1. First, we know the existence of a continuous path with nonincreasing loss from any point to any local minimum. We know that the local minimum is actually global from Haeffele & Vidal (2017). Now, we know that the set $\Theta^*(m)$ is connected: hence, we can construct a continuous path from that global minimum to any global minimum we want. Note that we can apply the result of Haeffele and Vidal (Grohs & Kutyniok (2022), chapter4) because $\phi(X, w, \alpha) = (Xw)_+\alpha$ and $\theta(w, \alpha) = \frac{\|w\|_2^2 + |\alpha|^2}{2}$ are nondegenerate pairs. $\quad\square$

**Corollary D.2.** *(Corollary 3 of the paper) Consider the optimization problem in equation 3. Suppose $m \geq n+1$ and denote the objective in equation 3 as $\mathcal{L}(\theta)$, where $\theta := (w_i, \alpha_i)_{i=1}^m \in \mathbb{R}^{(d+1)m}$. Let the optimal value of equation 3 as $p^*$. For any $\lambda$ greater than or equal to $p^*$, we have that the sublevel set $\{\theta \mid \mathcal{L}(\theta) \leq \lambda\}$ is connected.*

*Proof.* Take two points $\theta_1, \theta_2$ that satisfies $\mathcal{L}(\theta_1), \mathcal{L}(\theta_2) \leq \lambda$. Fix an arbitrary $\theta^*$ from the optimal set $\Theta^*$. From Corollary 2, we know the existence of a path with nonincreasing loss from $\theta_1$ to $\theta^*$, and $\theta_2$ to $\theta^*$. Hence we found a path inside the sublevel set $\{\theta \mid \mathcal{L}(\theta) \leq \lambda\}$ that connects $\theta_1$ and $\theta_2$. This means that the sublevel set is connected. $\quad\square$

# E   PROOFS IN SECTION 3.3

In this section, we give the examples constructed in Section 3.3 and their rigorous proof.

The specific examples we present are the following:

**Proposition E.1.** *Suppose $n = 3$, input data is given as*

$$\{(x_{1i}, x_{2i}, y_i)\}_{i=1}^3 = \{(1, 0, 1/6), (-1/2, \sqrt{3}/2, 2/3), (-1/2, -\sqrt{3}/2, 1/6)\},$$

$X = [x_1 x_2] \in \mathbb{R}^{3 \times 2}$. *Then, the minimization problem in equation 8 with free skip connections and without bias, namely the SNB problem, has at least two different solutions in $\mathcal{F}^*$.*

*Proof.* Let's consider the set $\mathcal{Q}_X = \{(Xu)_+ \mid \|u\|_2 \leq 1\}$. The six possible hyperplane arrangement patterns $\mathbb{1}(Xu \geq 0)$ are $(001), (010), (100), (011), (101), (110)$, and when we draw the set $\mathcal{Q}_X$ we get the following shape in Figure 7.

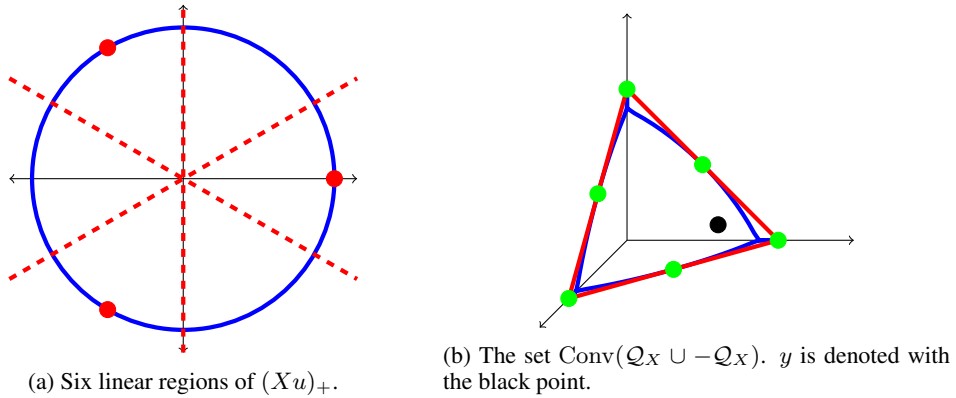

(a) Six linear regions of $(Xu)_+$.

(b) The set $\text{Conv}(\mathcal{Q}_X \cup -\mathcal{Q}_X)$. $y$ is denoted with the black point.

Figure 7: The shape of $\mathcal{Q}_X$ for a certain 2d data. The input space is split into six regions, and each region becomes either a 1d or a 2d object in a 3-dimensional space. Observe that $\mathcal{Q}_X$ meets with $x + y + z = 1$ with six points.

With some scaling trick, we can see that the problem is further equivalent to

$$\text{minimize}_{m, w_0, \{w_i, \alpha_i\}_{i=1}^m} \quad \sum_{i=1}^m |\alpha_i|, \quad \text{subject to} \quad Xw_0 + \sum_{i=1}^m (Xw_i)_+ \alpha_i = y, \ \|w_i\|_2 \leq 1 \quad \forall i \in [m].$$

Now, choose $\nu = [1, 1, 1]^T$. For any optimal $w_0, w_1, w_2, \alpha_0, \alpha_1$, we know that

$$\nu^T X w_0 + \sum_{i=1}^m \nu^T (Xw_i)_+ \alpha_i = \sum_{i=1}^m \nu^T (Xw_i)_+ \alpha_i = \langle \nu, y \rangle,$$

and

$$\langle \nu, y \rangle \leq \sum_{i=1}^m |\nu^T (Xw_i)_+| |\alpha_i| \leq \sum_{i=1}^m |\alpha_i|,$$

which means that the objective value is lower bounded by $\langle \nu, y \rangle = 1$. At last, we have two different models that have different breaklines and have objective value 1. The two models are:

$$w_0 = -\frac{1}{3} \begin{bmatrix} 1 \\ 0 \end{bmatrix}, w_1 = \frac{1}{\sqrt{2}} \begin{bmatrix} 1 \\ 0 \end{bmatrix}, w_2 = \frac{1}{\sqrt{2}} \begin{bmatrix} -1/2 \\ \sqrt{3}/2 \end{bmatrix}, \alpha_1 = \frac{1}{\sqrt{2}}, \alpha_2 = \frac{1}{\sqrt{2}},$$

and

$$w_0 = -\frac{1}{3} \begin{bmatrix} -1/2 \\ -\sqrt{3}/2 \end{bmatrix}, w_1 = \frac{1}{\sqrt{2}} \begin{bmatrix} -1/2 \\ -\sqrt{3}/2 \end{bmatrix}, w_2 = \frac{1}{\sqrt{2}} \begin{bmatrix} -1/2 \\ \sqrt{3}/2 \end{bmatrix}, \alpha_1 = \frac{1}{\sqrt{2}}, \alpha_2 = \frac{1}{\sqrt{2}}.$$

With direct substitution, we can see that they are both valid interpolators. A direct calculation shows that both have objective value of 1. At last, the breaklines differ, as the breaklines directly correspond to the weight vectors $w_1, w_2$: this means that the two optimal functions are different. $\qquad \square$

**Proposition E.2.** *Suppose $n = 4$, input data is given as*

$$\{(x_{1i}, x_{2i}, y_i)\}_{i=1}^4 = \{(1, 0, 1), (0, 1, -1), (-1, 0, 1), (0, -1, -1)\},$$

$X = [x_1 x_2] \in \mathbb{R}^{n \times 2}$. *Then, the minimization problem in equation 8, namely the SB problem, has at least two different solutions in $\mathcal{F}^*$.*

*Proof.* We give a similar proof strategy as that in Proposition E.1, writing $\bar{X} = [X \mid 1] \in \mathbb{R}^{n \times 3}$ and bounding $|\nu^T (\bar{X} u)_+|$ for $\nu = [1, -1, 1, -1]^T$ and $\|u\| \leq 1$. Note that $\bar{X}^T \nu = 0$. It is not easy to visualize the shape of $\{(\bar{X} u)_+ \mid \|u\|_2 \leq 1\}$ as in Figure 7, but we can solve the optimization problem by splitting the input domain into 14 regions where the function $(\bar{X} u)_+$ is linear. There are 14 such regions due to the classical result of Cover (1965). A simple convex optimization for these 14 liner regions yields that

$$|\nu^T (\bar{X} u)_+| \leq 1 \quad \forall \|u\|_2 \leq 1.$$

Another way to see this is writing $u = [a, b, c]$ and $\nu^T (\bar{X} u)_+$ as

$$\frac{1}{2}(|a + c| + |a - c| - |b + c| - |b - c|),$$

and with the constraint $a^2 + b^2 + c^2 \leq 1$, we have that the above formula is bounded between $-1$ and $1$. Using the same scaling trick, we see that the lower bound of the objective is $\langle \nu, y \rangle$ as in Proposition E.1, which is 4. At last, choose two models as

$$w_0 = \begin{bmatrix} 0 \\ 0 \\ 1 \end{bmatrix}, w_1 = \begin{bmatrix} 0 \\ \sqrt{2} \\ 0 \end{bmatrix}, w_2 = \begin{bmatrix} 0 \\ -\sqrt{2} \\ 0 \end{bmatrix}, \alpha_1 = -\sqrt{2}, \alpha_2 = -\sqrt{2},$$

and

$$w_0 = \begin{bmatrix} 0 \\ 0 \\ -1 \end{bmatrix}, w_1 = \begin{bmatrix} \sqrt{2} \\ 0 \\ 0 \end{bmatrix}, w_2 = \begin{bmatrix} -\sqrt{2} \\ 0 \\ 0 \end{bmatrix}, \alpha_1 = \sqrt{2}, \alpha_2 = \sqrt{2}.$$

Both solutions have cost 4 and interpolates the data. With some simplification, we can see that the first solution gives $f(x, y) = 1 - 2(y)_+ - 2(-y)_+$, whereas the second solution gives $f(x, y) = -1 + 2(x)_+ + 2(-x)_+$. Apparently we have two different minimum-norm interpolators. A visualization (and the symmetry behind it) can be found in Figure 8.

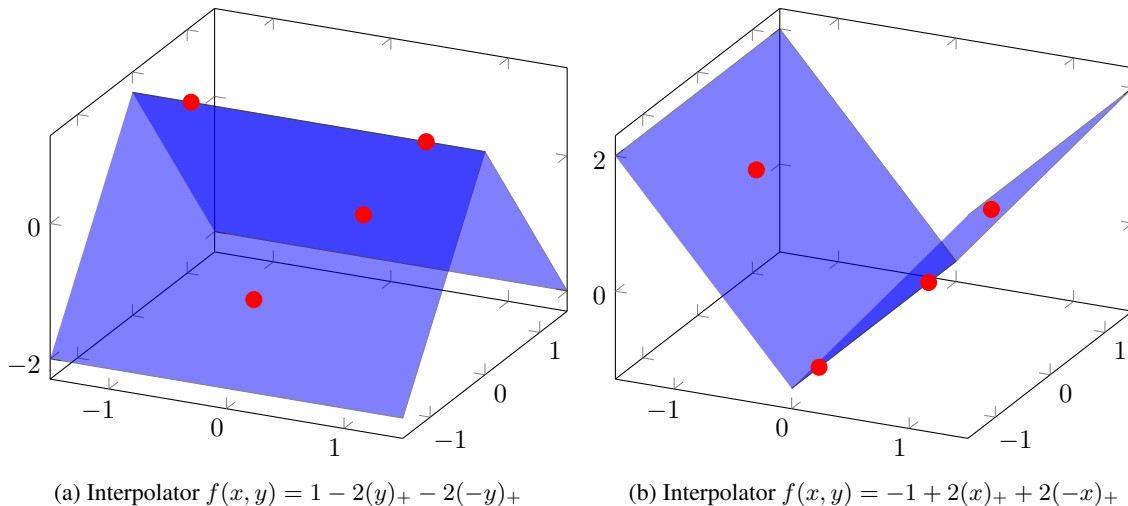

(a) Interpolator $f(x, y) = 1 - 2(y)_+ - 2(-y)_+$   (b) Interpolator $f(x, y) = -1 + 2(x)_+ + 2(-x)_+$

Figure 8: Two different minimum-norm interpolators. We can see that the V shape is the minimum-norm interpolator, and one is the rotation of the other.

□

The two propositions Proposition E.1 and Proposition E.2 gives Proposition 2.

**Proposition E.3.** *(Proposition 2 of the paper) Consider the minimum-norm interpolation problem*

$$\min_{m,u,\{w_i,\alpha_i\}_{i=1}^m} \frac{1}{2}\sum_{i=1}^m \left(\|w_i\|_2^2 + \alpha_i^2\right), \quad \text{subject to} \quad Xu + \sum_{i=1}^m (Xw_i)_+\alpha_i = y,$$

*where for input $X_{in}$, $X = X_{in}$ in the unbiased case and $X = [X_{in}|1]$ in the biased case. Note that $m$ is also an optimization variable that is not fixed. When the input is 2-dimensional, we have a non-unique minimum-norm interpolator regardless of bias.*

*Proof.* This is a direct consequence of Proposition E.1 and Proposition E.2. $\square$

**Proposition E.4.** *(Proposition 3 of the paper) Consider the minimum-norm interpolation problem without free skip connections and regularized bias, namely the NSB problem. Take $n$ vectors in $\mathbb{R}^2$ that satisfy*

$$v_n = [\frac{\sqrt{3}}{2}, \frac{1}{2}]^T, \ \|s_k\|_2 = 1, s_k > 0 \ \forall k \in [n-1], \ s_n = [0,1]^T, \ v_{i,2} > 0 \ \forall i \in [n].$$

*where we write $\sum_{i=1}^k v_{n-i+1} = s_k$. Now, choose $x_i = v_{i,1}/v_{i,2}$ and choose $y$ as any conic combination of $n+1$ vectors*

$$(s_{i,1}x + s_{i,2}1)_+ \quad \forall i \in [n], \quad ((s_n - v_n)_1 x + (s_n - v_n)_2 1)_+,$$

*with positive weights. Then, there exist infinitely many minimum-norm interpolators.*

*Proof.* Let's choose $x$ as proposed in Proposition 3. Also, let's choose $\nu \in \mathbb{R}^n$ as $\nu_i = v_{i,2}$. Write $\bar{X} = [x|1] \in \mathbb{R}^{n\times 2}$. Then the NSB problem is written as

$$\min_{m,\{w_i,\alpha_i\}_{i=1}^m} \sum_{i=1}^m \|w_i\|_2^2 + |\alpha_i|^2 \quad \text{subject to} \quad \sum_{i=1}^m (\bar{X}w_i)_+\alpha_i = y.$$

We first show that

$$\max_{\|u\|_2 \leq 1} |\nu^T(\bar{X}u)_+| = 1, \tag{19}$$

and the solutions are $s_1, s_2, \cdots s_n, s_n - v_n$.

The first thing to observe is that $x_1 < x_2 < \cdots < x_n$. To prove this, let's see that for $i = 2, 3, \cdots n-1$,

$$x_{i-1} < -\frac{s_{n-i+1,2}}{s_{n-i+1,1}} < x_i.$$

The reason is the following: for $i = 2, 3, \cdots n-1$, we know $\|s_{n-i+1} + v_{i-1}\|_2 = 1$, and as $\|s_{n-i+1}\|_2 = 1$ we know $s_{n-i+1} \cdot v_{i-1} = -1/2 \cdot \|v_{i-1}\|_2^2 < 0$. Hence, $s_{n-i+1,1}v_{i-1,1} + s_{n-i+1,2}v_{i-1,2} < 0$, and as $s_{n-i+1,1}, s_{n-i+1,2}, v_{i-1,2} > 0$, we have $s_{n-i+1,2}/s_{n-i+1,1} < -v_{i-1,1}/v_{i-1,2} = -x_{i-1}$. Similarly, $\|s_{n-i+1} - v_i\|_2 = 1$, and as $s_{n-i+1} \cdot v_i > 0$, we have $s_{n-i+1,2}/s_{n-i+1,1} > -v_{i,1}/v_{i,2} = -x_i$. This means for $i = 2, 3, \cdots n-1$, $x_{i-1} < x_i$, and $x_1 < x_2 < \cdots < x_{n-1}$. At last, we have $v_{n-1} \cdot v_n < 0$ because $\|v_n\|_2 = \|v_n + v_{n-1}\|_2 = 1$, meaning $x_{n-1} < 0$, whereas $x_n = \sqrt{3} > 0$, meaning $x_1 < x_2 < \cdots < x_n$.

Now we consider the possible arrangement patterns $\text{diag}(1(\bar{X}u \geq 0))$. We can see that the possible patterns are $\text{diag}([0,0,\cdots,0])$, $\text{diag}([0,0,\cdots,0,1])$, $\text{diag}([0,0,\cdots,1,1])$, $\cdots \text{diag}([0,1,\cdots,1,1])$, $\text{diag}([1,1,\cdots,1,1])$, $\cdots \text{diag}([1,0,\cdots,0,0])$. In other words, starting from the $n$-th entry, 0 turns to 1 in reverse order, then we have all ones, then 1s become 0s at starting from the $n$-th entry. Let's denote the diagonal matrices $D_1, D_2, \cdots D_{2n}$.

Solving equation 19 is equivalent to solving

$$\max_{(2D_i-I)\bar{X}u\geq 0, \ \|u\|_2\leq 1} \nu^T D_i \bar{X} u.$$

The absolute value function is erased as $\nu > 0$.

For $D_1$, the objective is 0. For $D_2$ to $D_{n+2}$, we first know that $\|\nu^T D_i \bar{X}\|_2 = 1$. To see this, observe that $\|\nu^T D_2 \bar{X}\|_2 = \|\nu_n[x_n,1]\|_2 = \|v_n\|_2 = 1$, $\|\nu^T D_3 \bar{X}\|_2 = \|\nu_n[x_n,1] + \nu_{n-1}[x_{n-1},1]\|_2 =$

$\|v_n + v_{n-1}\|_2 = 1, \cdots, \|\nu^T D_{n+1} \bar{X}\|_2 = \|\sum_{i=1}^{n} v_i\|_2 = 1, \|\nu^T D_{n+2} \bar{X}\|_2 = \|\sum_{i=1}^{n-1} v_i\|_2 = \|[-\frac{\sqrt{3}}{2}, \frac{1}{2}]\|_2 = 1$. For $D_{n+3}$ to $D_{2n}$, we can also see that $\|\nu^T D_i \bar{X}\|_2 < 1$. That is because $\nu^T D_{n+k} \bar{X} = s_n - s_{k-1}$ for $k \geq 2$. $\|s_n - s_{k-1}\|_2 = 2 - 2s_n \cdot s_{k-1}$, and as we know $1/2 = s_n \cdot s_1 < s_n \cdot s_2 < \cdots s_n \cdot s_{n-1}$ (which is the sum of $v_{i,2}$, and as $v_{i,2} > 0$ we have the property), $\|s_n - s_{k-1}\|_2 < 1$ for $k = 3, 4, \cdots n$.

We can then see that $\max_{(2D_i - I)\bar{X}u \geq 0, \|u\|_2 \leq 1} \nu^T D_i \bar{X} u \leq \max_{i \in [2n]} \|\nu^T D_i \bar{X}\|_2 = 1$. The last thing to check is that $s_1, s_2, \cdots s_n, s_n - v_n$ are actual solutions. For $i = 2, 3, \cdots n - 1$, we know that

$$x_1 < x_2 < \cdots < x_{i-1} < -\frac{s_{n-i+1,2}}{s_{n-i+1,1}} < x_i < x_{i+1} < \cdots < x_n,$$

and when we write $k_i = [x_i, 1]^T$, we have that $k_n \cdot s_{n-i+1} > 0, \cdots k_i \cdot s_{n-i+1} > 0, k_{i-1} \cdot s_{n-i+1} < 0, \cdots k_1 \cdots s_{n-i+1} < 0$ for all $i = 2, 3, \cdots n - 1$. Hence, for $s_2, s_3, \cdots s_{n-1}, (2D_{i+1} - I)\bar{X} s_i \geq 0$. As $s_i = \bar{X}^T D_{i+1} \nu$, for these $s_i$s, $\nu^T (\bar{X} s_i)_+ = 1$.

The three cases we have to check are when $i = 1, n$, and $s_n - v_n$. When $i = n$, $s_n = [0, 1]^T$ and as all $k_i$s have positive y values, $s_n \cdot k_i > 0$ for all $i \in [n]$ and indeed, $s_n$ becomes a solution. Also, we know that $\|s_1 + v_{n-1}\|_2 = 1$, meaning $x_1 < x_2 < \cdots < x_{n-1} < -s_{1,2}/s_{1,1}$. Hence, $k_{n-1} \cdot s_1 < 0, \cdots k_1 \cdot s_1 < 0$. As $s_1$ and $k_n$ are parallel, we know $k_n \cdot s_1 > 0$. Same with other cases, as $\|s_1\|_2 = 1$, $s_1$ is a solution. At last, let's check that $(2D_{n+2} - I)\bar{X}(s_n - v_n) \geq 0$. As $v_n = [\sqrt{3}/2, 1/2]^T$, $s_n - v_n = [-\sqrt{3}/2, 1/2]^T$, $v_n \cdot (s_n - v_n) < 0$ and $k_n \cdot (s_n - v_n) < 0$. For $i \in [n - 1]$, we know $x_i < 0$. Hence, $-\sqrt{3}x_i + 1 > 0$. This means $(2D_{n+2} - I)\bar{X}(s_n - v_n) > 0$ and as $\|s_n - v_n\|_2 = 1$, we have $s_n - v_n$ as a solution too.

Now that we have found $n + 1$ different solutions to problem in equation 19, let's note them $w_1, w_2, \cdots w_{n+1}$. $y$ is chosen as any conic combination

$$y = \sum_{i=1}^{n+1} c_i (\bar{X} w_i)_+, \tag{20}$$

where $c_i > 0$. We know that the interpolation problem is equivalent to

$$\min t \quad \text{subject to} \quad y \in t \text{Conv}(\mathcal{Q}_{\bar{X}} \cup -\mathcal{Q}_{\bar{X}}),$$

where $\mathcal{Q}_{\bar{X}} = \{(\bar{X}u)_+ \mid \|u\|_2 \leq 1\}$ Pilanci & Ergen (2020). In other words, the minimum-norm interpolation problem without free skip connections and regularized bias is equivalent to

$$\min_{m, (z_i, d_i)_{i=1}^m} \sum_{i=1}^m |d_i|, \quad y = \sum_{i=1}^m d_i (\bar{X} z_i)_+,$$

for some $\|z_i\|_2 \leq 1, i \in [m]$. For any $d_i, z_i$ that satisfies $\|z_i\|_2 \leq 1$ and

$$y = \sum_{i=1}^m d_i (\bar{X} z_i)_+,$$

we have that

$$\langle \nu, y \rangle = \sum_{i=1}^{n+1} c_i = \sum_{i=1}^m d_i \nu^T (\bar{X} z_i)_+ \leq \sum_{i=1}^m |d_i \nu^T (\bar{X} z_i)_+| \leq \sum_{i=1}^m |d_i|,$$

meaning $\langle \nu, y \rangle$ is the optimal value, and any conic combination of $\{(\bar{X} w_i)_+\}_{i=1}^{n+1}$ yields a solution.

Let's write $w_i = [a_i, b_i]^T$. The optimal interpolator then becomes

$$f(\bar{X}) = \sum_{i=1}^{n+1} c_i (a_i x + b_i)_+,$$

for $c_i$ s satisfying equation 20. The last thing to check is that there are infinitely many such interpolators. This is slightly different from $y$ having infinitely many different conic combination expressions of $\{(\bar{X} w_i)_+\}_{i=1}^{n+1}$, because different $c_i$ may correspond to the same function.

As a final step, we show that we indeed have infinitely many different interpolators. Recall that $\{s_1, s_2, \cdots s_n, s_n - v_n\} = \{w_1, w_2, \cdots w_{n+1}\}$. Note that $s_{1,1}, s_{2,1}, \cdots s_{n,1} \geq 0$ and $s_{n,1} - v_{n,1} < 0$. Without loss of generality let $s_i = w_i$ for $i \in [n]$ and $s_n - v_n = w_{n+1}$. As $x \to -\infty$, the slope will be $c_{n+1} a_{n+1}$. Showing the different conic representations of $y$ have different $c_{n+1}$ values is enough. The interesting observation is that the vectors $\{(\bar{X} w_i)_+\}_{i=1}^n$ is actually linearly independent, because each vector has $D_2, D_3, \cdots D_{n+1}$ as arrangement patterns and for each $s_1, s_2, \cdots s_n$, strict inequality holds. Hence, for each different conic combination of $y$, $c_{n+1}$ should be different. This means the slope at $x \to -\infty$ is different, and we indeed have infinitely many optimal interpolators. $\qquad\square$

# F PROOFS IN SECTION 4

In this section, we prove how our results can be generalized to different architectures and setups. We begin by describing a general solution set.

**Theorem F.1.** *Consider the cone-constrained group LASSO with regularization $\mathcal{R}_i$,*

$$\min_{\theta_i \in \mathcal{C}_i \cap \mathcal{V}_i, s_i \in \mathcal{D}_i} L(\sum_{i=1}^{P} A_i \theta_i + \sum_{i=1}^{Q} B_i s_i, y) + \beta \sum_{i=1}^{P} \mathcal{R}_i(\theta_i). \tag{21}$$

*Assume $\theta_i, s_j \in \mathbb{R}^d$, $A_i, B_j \in \mathbb{R}^{n \times d}$, $\mathcal{R}_i : \mathcal{V}_i \to \mathbb{R}$ are norms, $\mathcal{C}_i, \mathcal{D}_j \subseteq \mathbb{R}^d$ are proper cones for $i \in [P], j \in [Q]$, $\mathcal{C}_i \cap \mathcal{V}_i \neq \emptyset$ for $i \in [P]$ and $\beta > 0$. The optimal set $\mathcal{P}^*_{\text{gen}}$ is given as*

$$\mathcal{P}^*_{\text{gen}} = \Big\{ (c_i \bar{\theta}_i)_{i=1}^{P} \oplus (s_i)_{i=1}^{Q} \mid c_i \geq 0, \sum_{i=1}^{P} c_i A_i \bar{\theta}_i + \sum_{i=1}^{Q} B_i s_i \in \mathcal{C}_y,$$
$$\bar{\theta}_i \in Zer(F(\mathcal{S}_i, A_i^T \nu^*, -\beta, \langle, \rangle)), \langle B_i^T \nu^*, s_i \rangle = 0, s_i \in \mathcal{D}_i \Big\}, \tag{22}$$

*where $\nu^*$ is any vector that minimizes $f^*(\nu)$ subject to the constraint*

$$\min_{u \in \mathcal{C}_i \cap \mathcal{V}_i} \langle A_i^T \nu, u \rangle + \beta \mathcal{R}_i(u) = 0, \quad \min_{s \in \mathcal{D}_j} \langle B_j^T \nu, s \rangle = 0,$$

*for all $i \in [P]$, $j \in [Q]$, $F(\mathcal{S}, v, -\beta, \langle, \rangle) = \{u \mid u \in \mathcal{S}, \langle v, u \rangle = -\beta\}$, $\mathcal{S}_i = \mathcal{C}_i \cap \{u \mid \mathcal{R}_i(u) \leq 1\}$ and $Zer(\mathcal{S}) = \{0\}$ if $\mathcal{S} = \emptyset$, $\mathcal{S}$ otherwise. Here, $f(\cdot) = L(\cdot, y)$ and $f^*$ denotes the Fenchel conjugate of $f$.*

*Proof.* Suppose the optimal set of the problem in equation 21 is $\Theta^*_{\text{gen}}$. We show that $\Theta^*_{\text{gen}} \subseteq \mathcal{P}^*_{\text{gen}}$ and vice versa. Suppose $(\theta^*, s^*) = (\theta_i^*)_{i=1}^{P} \oplus (s_i^*)_{i=1}^{Q} \in \Theta^*_{\text{gen}}$. We know that $\sum_{i=1}^{P} A_i \theta_i^* + \sum_{i=1}^{Q} B_i s_i^* \in \mathcal{C}_y$, hence it satisfies the second condition for $w^* = \sum_{i=1}^{P} A_i \theta_i^* + \sum_{i=1}^{Q} B_i s_i^*$. Also, consider the convex optimization problem

$$\min_{w, \theta_i \in \mathcal{C}_i \cap \mathcal{V}_i, s_i \in \mathcal{D}_i} L(w, y) + \beta \sum_{i=1}^{P} \mathcal{R}(\theta_i) \text{ subject to } \sum_{i=1}^{P} A_i \theta_i + \sum_{i=1}^{Q} B_i s_i = w,$$

and its Lagrangian

$$\mathcal{L}(w, \theta, s, \nu) = L(w, y) - \nu^T w + \sum_{i=1}^{P} (\langle A_i^T \nu, \theta_i \rangle + \beta \mathcal{R}_i(\theta_i)) + \sum_{i=1}^{Q} \langle B_i^T \nu, s_i \rangle. \tag{23}$$

The strong duality argument is essentially the same as that with the proof in Theorem 1. Moreover, for the dual problem

$$\max_{\nu} \min_{w, \theta_i \in \mathcal{C}_i \cap \mathcal{V}_i, s_i \in \mathcal{D}_i} \mathcal{L}(w, \theta, s, \nu),$$

if $\min_{u \in \mathcal{C}_i \cap \mathcal{V}_i} \langle A_i^T \nu, u \rangle + \beta \mathcal{R}_i(u) < 0$, we can scale $u$ infinitely large to attain the minimum $-\infty$. Same holds when $\min_{u \in \mathcal{D}_i} \langle B_i^T \nu, u \rangle < 0$. Hence, these cases cannot maximize the dual objective, and the dual problem can be written as

$$\max_{\substack{\min_{u \in \mathcal{C}_i \cap \mathcal{V}_i} \langle A_i^T \nu, u \rangle + \beta \mathcal{R}_i(u) = 0 \\ \min_{u \in \mathcal{D}_i} \langle B_i^T \nu, u \rangle = 0}} \min_w L(w, y) - \nu^T w = \max_{\substack{\min_{u \in \mathcal{C}_i \cap \mathcal{V}_i} \langle A_i^T \nu, u \rangle + \beta \mathcal{R}_i(u) = 0 \\ \min_{u \in \mathcal{D}_i} \langle B_i^T \nu, u \rangle = 0}} -f^*(\nu),$$

meaning $\nu^*$ is the dual optimal point. When strong duality holds, for any primal optimal point $(w^*, \theta^*, s^*)$ and the dual optimal point $\nu^*$, the Lagrangian $\mathcal{L}(w, \theta, s, \nu^*)$ attains minimum at $(w^*, \theta^*, s^*)$. Substitute $\nu^*$ in equation 23 to see that each $\theta_i^*$ is a minimizer of the problem

$$\min \langle A_i^T \nu^*, u \rangle + \beta \mathcal{R}(u) \text{ subject to } u \in \mathcal{C}_i \cap \mathcal{V}_i.$$

One thing to notice is that the value $\langle A_i^T \nu^*, \theta_i^* \rangle + \beta \mathcal{R}_i(\theta_i^*) = 0$, because if it is strictly smaller than 0 we can strictly decrease the objective $\langle A_i^T \nu^*, u \rangle + \beta \mathcal{R}_i(u)$ with $u = 2\theta_i^*$.

If $\theta_i^* = 0$, we can choose $c_i = 0$ to find $c_i, \bar{\theta}_i \in Zer(F(\mathcal{S}_i, A_i^T \nu^*, -\beta))$. If $\theta_i^* \neq 0$, we know that $\mathcal{R}_i(\theta_i^*) \neq 0$, and the vector $\theta_i^* / \mathcal{R}_i(\theta_i^*)$ satisfies $\theta_i^* / \mathcal{R}_i(\theta_i^*) \in \mathcal{S}_i$ and $\langle A_i^T \nu^*, \theta_i^* / \mathcal{R}_i(\theta_i^*) \rangle = -\beta$. Choose $c_i = \mathcal{R}_i(\theta_i^*)$, $\bar{\theta}_i = \theta_i^* / \mathcal{R}_i(\theta_i^*)$ to find $c_i, \bar{\theta}_i \in Zer(F(\mathcal{S}_i, A_i^T \nu^*, -\beta))$.

For $s_i^*$ s, we know that each $s_i^*$ s are the minimizer of the problem

$$\min \langle B_i^T \nu^*, u \rangle \text{ subject to } u \in \mathcal{D}_i,$$

hence it should be in $\mathcal{D}_i$ and the value $\langle B_i^T \nu^*, s_i^* \rangle = 0$.

Concluding, for any $(\theta^*, s^*)$, clearly $\sum_{i=1}^P A_i \theta_i^* + \sum_{j=1}^Q B_j s_j^* \in \mathcal{C}_y$ and $s_i^* \in \mathcal{D}_i$, $\langle B_i^T \nu^*, s_i^* \rangle = 0$, choose $c_i = 0$ when $\theta_i^* = 0$, $c_i = \mathcal{R}(\theta_i^*)$, $\bar{\theta}_i = \theta_i^* / \mathcal{R}(\theta_i^*)$ otherwise to see that $(\theta^*, s^*) \in \mathcal{P}_{\text{gen}}^*$, and

$$\Theta_{\text{gen}}^* \subseteq \mathcal{P}_{\text{gen}}^*.$$

Now, let's take an element $(\theta, s) \in \mathcal{P}_{\text{gen}}^*$. We know that $\theta \in \mathcal{C}_i \cap \mathcal{V}_i$ and $s \in \mathcal{D}_i$. If $\bar{\theta}_i \neq 0$, we know that $\langle \nu^*, A_i \bar{\theta}_i \rangle = -\beta$ as $\bar{\theta}_i \in F(\mathcal{S}_i, A_i^T \nu^*, -\beta)$. Moreover, $\bar{\theta}_i$ is the solution to

$$\min_{u \in \mathcal{C}_i \cap \mathcal{V}_i, \mathcal{R}_i(u) \leq 1} \langle A_i^T \nu^*, u \rangle,$$

because for all $u \in \mathcal{S}_i$, $\langle A_i^T \nu^*, u \rangle \geq -\beta \mathcal{R}_i(u) \geq -\beta$ holds. This means $\mathcal{R}_i(\bar{\theta}_i) = 1$ for all $\bar{\theta}_i \neq 0$, as the minimum will be attained at a nonzero point, hence the boundary where $\mathcal{R}_i(u) = 1$. Using $\langle \nu^*, A_i \bar{\theta}_i \rangle = -\beta$ and $\langle B_i^T \nu^*, s_i \rangle = 0$, we get

$$\langle \nu^*, w' \rangle = \langle \nu^*, \sum_{i=1}^P c_i A_i \bar{\theta}_i + \sum_{i=1}^Q B_i s_i \rangle = -\beta \sum_{\bar{\theta}_i \neq 0} c_i,$$

for some $w' \in \mathcal{C}_y$. On the other hand, from $\mathcal{R}_i(\bar{\theta}_i) = 1$ for all $i \in [P]$, we know that

$$\sum_{i=1}^P \mathcal{R}_i(c_i \bar{\theta}_i) = \sum_{\bar{\theta}_i \neq 0} c_i.$$

This leads to the fact that for $(\theta, s)$,

$$L(\sum_{i=1}^P A_i \theta_i + \sum_{i=1}^Q B_i s_i, y) + \beta \sum_{i=1}^P \mathcal{R}_i(\theta_i) = L(w', y) + \beta \sum_{\bar{\theta}_i \neq 0} c_i = L(w', y) - \langle \nu^*, w' \rangle.$$

At last, we show that for all $w' \in \mathcal{C}_y$,

$$L(w', y) - \langle \nu^*, w' \rangle = \min_{\theta_i \in \mathcal{C}_i \cap \mathcal{V}_i, s_i \in \mathcal{D}_i} L(\sum_{i=1}^P A_i \theta_i + \sum_{i=1}^Q B_i s_i, y) + \beta \sum_{i=1}^P \mathcal{R}_i(\theta_i).$$

The fact follows when we use the fact that for $(\theta', s') \in \Theta_{\text{gen}}^*$ that satisfies $w' = \sum_{i=1}^P A_i \theta_i' + \sum_{i=1}^Q B_i s_i'$, the point $(w', \theta', s')$ becomes a minimizer of $\mathcal{L}(w, \theta, s, \nu^*)$. Hence, each minimizer $\theta_i'$ is a minimizer of the problem

$$\min \langle A_i^T \nu^*, u \rangle + \beta \mathcal{R}_i(u) \text{ subject to } u \in \mathcal{C}_i \cap \mathcal{V}_i,$$

which means that $\beta \mathcal{R}_i(\theta_i') = -\langle \nu^*, A_i \theta_i' \rangle$ for all $i \in [P]$, as $\nu^*$ satisfies

$$\min_{u \in \mathcal{C}_i \cap \mathcal{V}_i} \langle A_i^T \nu^*, u \rangle + \beta \mathcal{R}_i(u) = 0.$$

Also, $\langle \nu^*, B_i s_i' \rangle = 0$ as $s_i'$ minimizes $\langle B_i^T \nu^*, s \rangle$ subject to $s \in \mathcal{D}_i$, and we see that

$$\beta \sum_{i=1}^P \mathcal{R}_i(\theta_i') = -\langle \nu^*, w' \rangle,$$

and

$$\min_{\theta_i \in \mathcal{C}_i \cap \mathcal{V}_i, s_i \in \mathcal{D}_i} L(\sum_{i=1}^P A_i \theta_i + \sum_{i=1}^Q B_i s_i, y) + \beta \sum_{i=1}^P \mathcal{R}(\theta_i) = L(\sum_{i=1}^P A_i \theta_i' + \sum_{i=1}^Q B_i s_i', y) + \beta \sum_{i=1}^P \mathcal{R}(\theta_i')$$
$$= L(w', y) - \langle \nu^*, w' \rangle,$$

meaning $(\theta, s) \in \Theta^*_{\text{gen}}$ because

$$L(\sum_{i=1}^{P} A_i \theta_i + \sum_{i=1}^{Q} B_i s_i, y) + \beta \sum_{i=1}^{P} \mathcal{R}(\theta_i) = L(w', y) - \langle \nu^*, w' \rangle.$$

This means

$$\mathcal{P}^*_{\text{gen}} \subseteq \Theta^*_{\text{gen}},$$

and finishes the proof. $\qquad\square$

One application of the theorem is characterizing the optimal set of the interpolation problem. This leads to the staircase of connectivity for interpolation problems.

**Proposition F.1.** *The solution set of the optimization problem*

$$\min_{u_i, v_i \in \mathcal{K}_i} \quad \sum_{i=1}^{P} \|u_i\|_2 + \|v_i\|_2$$

*subject to*

$$\sum_{i=1}^{P} D_i X (u_i - v_i) = y,$$

*is given as*

$$\mathcal{P}^* := \left\{ (c_i \bar{u}_i, d_i \bar{v}_i)_{i=1}^{P} \mid c_i, d_i \geq 0 \quad \forall i \in [P], \quad \sum_{i=1}^{P} D_i X \bar{u}_i c_i - D_i X \bar{v}_i d_i = y \right\} \subseteq \mathbb{R}^{2dP},$$

*where $\bar{u}_i, \bar{v}_i$ are fixed directions found by solving the optimization problem*

$$\bar{u}_i = \arg\min_{u \in \mathcal{S}_i} \nu^{*T} D_i X u \text{ if } \min_{u \in \mathcal{S}_i} \nu^{*T} D_i X u = -1, \; 0 \text{ otherwise,}$$

$$\bar{v}_i = \arg\min_{v \in \mathcal{S}_i} -\nu^{*T} D_i X v \text{ if } \min_{v \in \mathcal{S}_i} -\nu^{*T} D_i X v = -1, \; 0 \text{ otherwise.}$$

*and $\nu^*$ is any dual optimum that satisfies*

$$\nu^* = \arg\min \langle \nu, y \rangle \text{ subject to } |\nu^T D_i X u| \leq \|u\|_2 \; \forall u \in \mathcal{K}_i, i \in [P].$$

*Here, $\mathcal{S}_i = \mathcal{K}_i \cap \{u \mid \|u\|_2 \leq 1\}$.*

*Proof.* Let's apply Theorem F.1 to the problem. Note that we can set $\beta = 1$. In fact, $\beta$ can be arbitrary, and scaling $\nu^*$ to make $\beta = 1$ will lead to the same result.

When we apply Theorem F.1, we have that

$$\mathcal{P}^*_{\text{gen}} = \Big\{ (c_i \bar{u}_i, d_i \bar{v}_i)_{i=1}^{m} \mid c_i, d_i \geq 0, \sum_{i=1}^{P} D_i X \bar{u}_i c_i - D_i X \bar{v}_i d_i = y,$$

$$\bar{u}_i \in Zer(F(\mathcal{S}_i, X^T D_i \nu^*, -1)), \bar{v}_i \in Zer(F(\mathcal{S}_i, -X^T D_i \nu^*, -1)) \Big\},$$

where $\nu^*$ is the dual optimum that minimizes $L^*(\cdot, y)$ subject to the constraint

$$\min_{u \in \mathcal{K}_i} \langle X^T D_i \nu, u \rangle + \|u\|_2 = 0, \quad \min_{u \in \mathcal{K}_i} \langle -X^T D_i \nu, u \rangle + \|u\|_2 = 0, \tag{24}$$

for all $i \in [P]$. We know that $L^*(\nu) = \langle \nu, y \rangle$, and equation 24 can be rewritten to $|\nu^T D_i X u| \leq \|u\|_2$.

Also, $F(\mathcal{S}_i, X^T D_i \nu^*, -1) = \{0\}$ if there is no $u$ such that $\nu^{*T} D_i X u = -1$, and is exactly that vector if exists. Note that as $\min_{u \in \mathcal{K}_i} \langle X^T D_i \nu, u \rangle + \|u\|_2 = 0$, we have a unique minimum for the optimal direction Proposition C.2. Same holds for $\bar{v}_i$. $\qquad\square$

**Proposition F.2.** *(The staircase of connectivity for minimum-norm interpolation problem) Write the solution set of the optimization problem*

$$\min_{(w_j, \alpha_j)_{j=1}^m} \quad \frac{1}{2} \sum_{i=1}^m \|w_i\|_2^2 + |\alpha_i|^2,$$

*subject to*

$$\sum_{i=1}^m (Xw_i)_+ \alpha_i = y,$$

*as $\Theta^*(m)$. Suppose $y \neq 0$. As $m$ changes, we have that*

(i) *$m = m^*$, $\Theta^*(m)$ is a finite set. Hence, for any two optimal points $A \neq A' \in \Theta^*(m)$, there is no path from $A$ to $A'$ inside $\Theta^*(m)$.*

(ii) *$m \geq m^* + 1$, there exists optimal points $A, A' \in \Theta^*(m)$ and a path in $\Theta^*(m)$ connecting them.*

(iii) *$m = M^*$, $\Theta^*(m)$ is not a connected set. Moreover, there exists $A \in \Theta^*(m)$ which is an isolated point, i.e. there is no path in $\Theta^*(m)$ that connects $A$ with $A' \neq A \in \Theta^*(m)$.*

(iv) *$m \geq M^* + 1$, permutations of the solution are connected. Hence, for all $A \in \Theta^*(m)$, there exists $A' \neq A$ in $\Theta^*(m)$ and a path in $\Theta^*(m)$ that connects $A$ and $A'$.*

(v) *$m \geq \min\{m^* + M^*, n + 1\}$, the set $\Theta^*(m)$ is connected, i.e. for any two optimal points $A \neq A' \in \Theta^*(m)$, there exists a continuous path from $A$ to $A'$.*

*Proof.* The proof follows from observing that Proposition D.2, Proposition D.3, Proposition D.4, Proposition D.5, Theorem D.1, Proposition D.7, Proposition D.12, Proposition D.13, Proposition D.14 holds for interpolation problems too. We can apply the same proof strategy for Proposition D.2, Proposition D.3, Proposition D.4, Proposition D.5, Theorem D.1, because the description of the optimal polytope is identical except for which directions the solutions are fixed at. The same solution map can be applied because it preserves both the fit and the regularization. The continuity is preserved, and we have Proposition D.12, Proposition D.13, Proposition D.14. The mapping in Proposition D.7 can also be applied here. □

Another implication of the theorem is that for free skip connections, the dual variable has to satisfy $X^T \nu = 0$. The existence of free skip connections constrain freedom on $\nu$, which brings qualitative difference to the uniqueness of the solution set.

**Proposition F.3.** *The solution set of the optimization problem*

$$\min_{u_i, v_i \in \mathcal{K}_i} \quad \sum_{i=1}^P \|u_i\|_2 + \|v_i\|_2$$

*subject to*

$$Xu_0 + \sum_{i=1}^P D_i X(u_i - v_i) = y,$$

*is given as*

$$\mathcal{P}^* := \left\{ u_0 \oplus (c_i \bar{u}_i, d_i \bar{v}_i)_{i=1}^P \mid c_i, d_i \geq 0 \; \forall i \in [P], \; Xu_0 + \sum_{i=1}^P D_i X \bar{u}_i c_i - D_i X \bar{v}_i d_i = y \right\} \subseteq \mathbb{R}^{2dP},$$

*where $\bar{u}_i, \bar{v}_i$ are fixed directions found by solving the optimization problem*

$$\bar{u}_i = \arg\min_{u \in \mathcal{S}_i} \nu^{*T} D_i X u \; \text{ if } \; \min_{u \in \mathcal{S}_i} \nu^{*T} D_i X u = -1, \; 0 \; \text{ otherwise,}$$

$$\bar{v}_i = \arg\min_{v \in \mathcal{S}_i} -\nu^{*T} D_i X v \; \text{ if } \; \min_{v \in \mathcal{S}_i} -\nu^{*T} D_i X v = -1, \; 0 \; \text{ otherwise.}$$

*where $\nu^*$ is the dual optimum that satisfies*

$$\nu^* = \arg\min \langle \nu, y \rangle \; \text{subject to} \; |\nu^T D_i X u| \leq \|u\|_2 \; \forall u \in \mathcal{K}_i, i \in [P], \quad X^T \nu = 0.$$

*Here, $\mathcal{S}_i = \mathcal{K}_i \cap \{u \mid \|u\|_2 \leq 1\}$.*

*Proof.* Note that if we apply Theorem F.1 to the given problem, we have almost an identical characterization from Proposition F.1, except for the free skip connection. For the skip connection $u_0 \in \mathbb{R}^d$, we know that $\min_{u_0 \in \mathbb{R}^d} \langle X^T \nu^*, u_0 \rangle = 0$ for all $u_0 \in \mathbb{R}^d$ because $u_0$ is unconstrained. This means $X^T \nu^* = 0$. $\qquad\square$

Next we give applications of Theorem F.1 to different architectures. We start by characterizing the optimal set of a vector-valued neural network with weight decay.

**Proposition F.4.** *The solution set of the convex reformulation of the vector-valued problem given as*

$$\min_{V_i} \frac{1}{2} \| \sum_{i=1}^{P} D_i X V_i - Y \|_2^2 + \beta \sum_{i=1}^{P} \| V_i \|_{\mathcal{K}_i, *}, \tag{25}$$

*where the norm $\| V \|_{\mathcal{K}_i, *}$ is defined as*

$$\min t \geq 0 \ \text{ such that } \ V \in t\mathcal{K}_i,$$

*for $\mathcal{K}_i = \text{conv}\{ug^T \,|\, (2D_i - I)Xu \geq 0, \|ug^T\|_* \leq 1\}$ defined in $\mathcal{V}_i = \text{span}\{ug^T \,|\, (2D_i - I)Xu \geq 0, \ g \in \mathbb{R}^c\}$. The optimal solution set of equation 25 is given as*

$$\mathcal{P}^*_{\text{vec}} = \left\{ (c_i \bar{V}_i)_{i=1}^P \,|\, c_i \geq 0, \sum_{i=1}^{P} c_i D_i X \bar{V}_i = Y^*, \bar{V}_i \in Zer(\mathrm{F}(\mathcal{K}_i, X^T D_i N^*, -\beta, \langle, \rangle_M)) \right\},$$

*where $N^* = Y^* - Y$ is the dual optimum that minimizes $\|N + Y\|_F^2$ subject to*

$$\langle N, D_i X A \rangle + \beta \|A\|_{\mathcal{K}_i, *} \geq 0 \quad \forall A \in \mathcal{V}_i, \quad i \in [P].$$

*Proof.* Let's define $A_i$ as

$$A_i = \begin{bmatrix} D_i X & & \\ & \ddots & \\ & & D_i X \end{bmatrix},$$

which is a block matrix in $A_i \in \mathbb{R}^{nc \times dc}$. Also, define the flattening operation $Fl_{dc} : \mathbb{R}^{d \times c} \to \mathbb{R}^{dc}$ and $Fl_{nc} : \mathbb{R}^{n \times c} \to \mathbb{R}^{nc}$. For optimization variables $\theta_i \in \mathbb{R}^{dc}$, we have the equivalent problem

$$\min_{\theta_i} \frac{1}{2} \| \sum_{i=1}^{P} A_i \theta_i - Fl_{nc}(Y) \|_2^2 + \beta \sum_{i=1}^{P} \| \theta_i \|_{Fl_{dc}(\mathcal{K}_i), *}.$$

Here, we are merely flattening each $V_i$ s to make it into a vector-input optimization problem. When we apply Theorem F.1 to the flattened problem, we have the optimal set

$$\mathcal{P}^*_{\text{flat}} = \left\{ (c_i \bar{\theta}_i)_{i=1}^P \,|\, c_i \geq 0, \sum_{i=1}^{P} c_i A_i \bar{\theta}_i = Fl_{nc}(Y^*), \bar{\theta}_i \in Zer(\mathrm{F}(S_i, A_i^T \nu^*, -\beta, \langle, \rangle)) \right\},$$

where

$$S_i = \{u | \ \|u\|_{Fl_{dc}(\mathcal{K}_i), *} \leq 1\} = Fl_{dc}(\mathcal{K}_i),$$

$\nu^*$ being the minimizer of $f^*(\nu)$ where $f = L_F(\ \cdot\ , Fl_{nc}(Y))$, subject to $\langle A_i^T \nu^*, s \rangle + \beta \|s\|_{Fl_{dc}(\mathcal{K}_i), *} \geq 0$. We know that $\nu^* = Fl_{nc}(Y^*) - Fl_{nc}(Y)$ for the optimal model fit $Y^*$. Write $N^* = Fl_{nc}^{-1}(\nu^*)$.

Now we use $Fl_{dc}^{-1}, Fl_{nc}^{-1}$ to go back to the original solution space and recover $\mathcal{P}^*_{\text{vec}}$. First, we know that $A_i \bar{\theta}_i = Fl_{nc}(D_i X Fl_{dc}^{-1}(\bar{\theta}_i))$. Hence, the constraint $\sum_{i=1}^P c_i A_i \bar{\theta}_i = Fl_{nc}(Y^*)$ is equivalent to

$$\sum_{i=1}^{P} c_i D_i X Fl_{dc}^{-1}(\bar{\theta}_i) = Y^*. \tag{26}$$

Also, consider the set

$$\mathrm{F}(Fl_{dc}(\mathcal{K}_i), A_i^T \nu^*, -\beta, \langle, \rangle) = Fl_{dc}(\mathcal{K}_i) \cap \{u \,|\, \langle A_i^T \nu^*, u \rangle = -\beta\}.$$

When we use block notation $(\nu^*)^T = [(\nu_1^*)^T, (\nu_2^*)^T, \cdots (\nu_c^*)^T]$, $u^T = [(u_1)^T, (u_2)^T, \cdots, (u_c)^T]$ for $\nu^* \in \mathbb{R}^{nc}$, $u \in \mathbb{R}^{dc}$, we can see that

$$(\nu^*)^T A_i u = \sum_{j=1}^{c} (\nu_j^*)^T D_i X u_j = \langle Fl_{nc}^{-1}(\nu^*), D_i X Fl_{dc}^{-1}(u) \rangle_M,$$

using notations for matrix inner product. Hence, we can see that

$$Fl_{dc}^{-1}(\mathrm{F}(Fl_{dc}(\mathcal{K}_i), A_i^T \nu^*, -\beta, \langle, \rangle_M)) = \mathcal{K}_i \cap \{U \mid \langle Fl_{nc}^{-1}(\nu^*), D_i X U \rangle = -\beta\}$$
$$= \mathrm{F}(\mathcal{K}_i, X^T D_i N^*, -\beta, \langle, \rangle_M),$$

and for $(\theta_i)_{i=1}^{P} \in \mathcal{P}_{\text{flat}}^*$, $Fl_{dc}^{-1}(\theta_i)$ satisfies equation 26 and we also have the fact that $Fl_{dc}^{-1}(\bar{\theta}_i) \in \mathrm{F}(\mathcal{K}_i, X^T D_i N^*, -\beta, \langle, \rangle_M)$. Hence we arrive at the desired result. $\qquad\square$

**Proposition F.5.** *Assume $m \geq m^*$ so that the nonconvex problem in equation 11 and its convex reformulation in equation 25 are equivalent. The solution set of the vector-valued problem*

$$\min_{\{w_i, z_i\}_{i=1}^{m}} \frac{1}{2}\|\sum_{i=1}^{m}(Xw_i)_+ z_i^T - Y\|_2^2 + \frac{\beta}{2}\sum_{i=1}^{m}\|w_i\|_2^2 + \|z_i\|_2^2,$$

*where $w_i \in \mathbb{R}^{d \times 1}, z_i \in \mathbb{R}^{c \times 1}$ is given as*

$$\mathcal{S} = \Big\{(w_i, z_i)_{i=1}^{m} \mid \phi((w_i, z_i)_{i=1}^{m}) \in \mathcal{P}_{\text{vec}}^*, \mathcal{R}((w_i, z_i)_{i=1}^{m}) = \|\phi((w_i, z_i)_{i=1}^{m})\|_{\mathcal{K}_i, *},$$
$$\|w_i\|_2 = \|z_i\|_2, \ \forall i \in [m]\Big\},$$

*where*

$$\phi((w_i, z_i)_{i=1}^{m}) = (V_i)_{i=1}^{P} := V_p = \begin{cases} 0 \ if \ \nexists \ w_i \ s.t. \ D_p = \mathrm{diag}(1(Xw_i \geq 0)) \\ \sum_{j=1}^{t_p} w_{a_j} z_{a_j}^T \ if \ D_p = \mathrm{diag}(1(Xw_{a_j} \geq 0)) \ for \ j \in [t_p], \end{cases}$$

$$\mathcal{R}((w_i, z_i)_{i=1}^{m}) = (R_i)_{i=1}^{P} := R_p = \begin{cases} 0 \ if \ \nexists \ w_i \ s.t. \ D_p = \mathrm{diag}(1(Xw_i \geq 0)) \\ \sum_{j=1}^{t_p} \|w_{a_j}\|_2 \|z_{a_j}\|_2 \ if \ D_p = \mathrm{diag}(1(Xw_{a_j} \geq 0)) \ for \ j \in [t_p], \end{cases}$$

*and $\mathcal{P}_{\text{vec}}^*$ is defined in Proposition F.4.*

*Proof.* Let's note the solution set of equation 11 as $\Theta^*$. We will prove that $\Theta^* = \mathcal{S}$. First, find a point $(w_i^*, z_i^*)_{i=1}^{m}$ in $\Theta^*$. When $\phi((w_i^*, z_i^*)_{i=1}^{m}) = (V_i^*)_{i=1}^{P}$, we know that

$$\sum_{i=1}^{m}(Xw_i^*)_+(z_i^*)^T = \sum_{i=1}^{P} D_i X V_i^*,$$

hence the $l_2$ error is the same for both parameters. Also, we have that $\sum_{i=1}^{P}\|V_i^*\|_{\mathcal{K}_i, *} \leq \sum_{i=1}^{m}\|w_i^*\|_2 \|z_i^*\|_2 = \frac{1}{2}\sum_{i=1}^{m}\|w_i^*\|_2^2 + \|z_i^*\|_2^2$. Thus, when we note $L_{\text{noncvx}}$ as the loss function of equation 11 and note $L_{\text{cvx}}$ as the loss function of equation 25, we have that

$$L_{\text{noncvx}}((w_i^*, z_i^*)_{i=1}^{m}) \geq L_{\text{cvx}}(\phi((w_i^*, z_i^*)_{i=1}^{m})), \tag{27}$$

holds in general. As the minimal value of $L_{\text{noncvx}}$ and $L_{\text{cvx}}$ is the same, we have that $\phi((w_i^*, z_i^*)_{i=1}^{m}) \in \mathcal{P}_{\text{vec}}^*$. Also, the inequality in equation 27 is actually an equality, and we have $\mathcal{R}((w_i, z_i)_{i=1}^{m}) = (\|V_i\|_{\mathcal{K}_i, *})_{i=1}^{P}$.

Now we take a point $(w_i, z_i)_{i=1}^{m}$ in $\mathcal{S}$. We know that $L_{\text{cvx}}(\phi((w_i, z_i)_{i=1}^{m}))$ is the optimal value. Also, we know that $L_{\text{noncvx}}((w_i, z_i)_{i=1}^{m}) = L_{\text{cvx}}(\phi((w_i, z_i)_{i=1}^{m}))$ because $\mathcal{R}((w_i, z_i)_{i=1}^{m}) = (\|\phi((w_i, z_i)_{i=1}^{m})\|_{\mathcal{K}_i, *})_{i=1}^{P}$ and $\|w_i\|_2 = \|z_i\|_2 \forall i \in [m]$. At last, the fact that as $m \geq m^*$ and the two optimal values are the same implies that $(w_i, z_i)_{i=1}^{m} \in \Theta^*$. $\qquad\square$

**Theorem F.2.** *Consider a $L$ - layer neural network*

$$f_\theta(X) = ((((XW_1)_+ W_2)_+ \cdots)W_{L-1})_+ W_L$$

*where $W_i \in \mathbb{R}^{d_{i-1} \times d_i}$, $d_0 = d$ and $\theta = (W_i)_{i=1}^L$. Consider the training problem*

$$\min_\theta L(f_\theta(X), y) + \frac{\beta}{2} \sum_{i=1}^L \|W_i\|_F^2,$$

*and denote its optimal set as $\Theta^*$. We can characterize a subset of $\Theta^*$, namely the set*

$$\Theta_{k-1,k}^*(Y', W_1', W_2', \cdots, W_{k-2}', W_{k+1}', \cdots W_L')$$
$$:= \Big\{ \theta = (W_i')_{i=1}^{k-2} \oplus (W_{k-1}, W_k) \oplus (W_i')_{i=k+1}^L \mid \theta \in \Theta^*, \ (\tilde{X} W_{k-1})_+ W_k = Y' \Big\}.$$

*Here, $\tilde{X} = (((( X W_1')_+ W_2')_+) \cdots W_{k-2}')_+$.*

*The expression of $\Theta_{k-1,k}^*(Y', W_1', W_2', \cdots, W_{k-2}', W_{k+1}', \cdots W_L')$ is given as*

$$\Big\{ \theta = (W_i')_{i=1}^{k-2} \oplus (W_{k-1}, W_k) \oplus (W_i')_{i=k+1}^L \mid \theta \in \Theta^*, \ \phi_{d_k}(W_{k-1}, W_k) \in \mathcal{P}_{\text{vec,intp}}^*,$$
$$\mathcal{R}_{d_k}(W_{k-1}, W_k) = \|\phi_{d_k}(W_{k-1}, W_k)\|_{\mathcal{K}_i,*}, \|(W_{k-1})_{\cdot,i}\|_2 = \|(W_k)_{i,\cdot}\|_2 \ \forall i \in [d_k] \Big\},$$

*where $\phi_m(A, B) = \phi((A_{\cdot,i}, B_{i,\cdot})_{i=1}^m)$, $\mathcal{R}_m(A, B) = \mathcal{R}((A_{\cdot,i}, B_{i,\cdot})_{i=1}^m)$ for $\phi$ defined in Proposition F.5, and $\mathcal{P}_{\text{vec,intp}}^*$ is defined as*

$$\mathcal{P}_{\text{vec,intp}}^* = \Big\{ (c_i \bar{V}_i)_{i=1}^P \mid c_i \geq 0, \sum_{i=1}^P c_i D_i X \bar{V}_i = Y', \bar{V}_i \in Zer(\mathrm{F}(\mathcal{K}_i, X^T D_i N^*, -1, \langle,\rangle_M)) \Big\},$$

*for the dual optimum $N^* \in \mathbb{R}^{n \times c}$ that minimizes $\langle N, Y \rangle_M$ subject to*

$$\langle N, D_i X A \rangle + \beta \|A\|_{\mathcal{K}_i,*} \geq 0 \quad \forall A \in \mathbb{R}^{d \times c}, \quad i \in [P].$$

*Here $\mathcal{K}_i = \mathrm{conv}\{ug^T \mid (2D_i - I)Xu \geq 0, \|ug^T\|_* \leq 1\}$, where $D_i$ denotes all possible arrangements $\mathrm{diag}(1(Xh \geq 0))$.*

*Proof.* The result is an application of Theorem F.1 to the vector-valued interpolation problem

$$\sum_{i=1}^{d_k} \|u_i\|_2^2 + \|v_i\|_2^2,$$

subject to

$$\sum_{i=1}^{d_k} (Xu_i)_+ v_i^T = Y',$$

where $u_i \in \mathbb{R}^{d_{k-1} \times 1}$, $v_i \in \mathbb{R}^{d_{k+1} \times 1}$, and then applying Proposition F.5. $\qquad \square$

The characterization enables us to extend the connectivity result to vector-valued networks.

**Theorem F.3.** *Consider the optimization problem*

$$\min_{\{w_i, z_i\}_{i=1}^m} \frac{1}{2} \Big\| \sum_{i=1}^m (Xw_i)_+ z_i^T - Y \Big\|_2^2 + \frac{\beta}{2} \sum_{i=1}^m \|w_i\|_2^2 + \|z_i\|_2^2, \qquad (28)$$

*where $w_i \in \mathbb{R}^d$, $z_i \in \mathbb{R}^c$ for $i \in [m]$, and $Y \in \mathbb{R}^{n \times c}$. If $m \geq nc + 1$, the solution set in parameter space $\Theta^* \subseteq \mathbb{R}^{m(d+c)}$ is connected.*

*Proof.* Let's take two solutions $(w_i, z_i)_{i=1}^m, (w_i', z_i')_{i=1}^m \in \Theta^*$. We write $\bar{w}$ as the direction of $w$, i.e. $w/\|w\|_2$ for $w \neq 0$.

The first claim we prove is that for given $\{(X\bar{w}_{a_i})_+ \bar{z}_{a_i}^T\}_{i=1}^{m_1}$ and $\{(X\bar{w}'_{b_i})_+ \bar{z}'_{b_i}^T\}_{i=1}^{m_2}$, consider the conic combination that satisfies

$$\sum_{i=1}^{m_1} c_i (X\bar{w}_{a_i})_+ \bar{z}_{a_i}^T + \sum_{i=1}^{m_2} d_i (X\bar{w}'_{b_i})_+ \bar{z}'_{b_i}^T = Y^*,$$

for the optimal model fit $Y^*$. Then $(\sqrt{c_i}\bar{w}_{a_i}, \sqrt{c_i}\bar{z}_{a_i})_{i=1}^{m_1} \oplus (\sqrt{d_i}\bar{w}'_{b_i}, \sqrt{d_i}\bar{z}'_{b_i})_{i=1}^{m_2} \oplus (0,0)^{m-m_1-m_2}$ is an optimal solution of equation 28 when $m_1 + m_2 \leq m$, given that $w_{a_i}, w'_{b_i} \neq 0$.

To see this, we first see that for the dual variable $N^*$, $\langle N^*, (X\bar{w}_i)_+\bar{z}_i^T\rangle = -\beta$ for all $w_i \neq 0$. Suppose $D_p = \text{diag}(1(Xw_{a_i} \geq 0))$ for $i \in [t_p]$, the same notation as in the statement of Proposition F.5, and without loss of generality assume $a_1 = i$. As $(w_i, z_i)_{i=1}^m \in \mathcal{S}$, we know $\mathcal{R}((w_i, z_i)_{i=1}^m) = (\|V_i\|_{\mathcal{K}_{i,*}})_{i=1}^P$. Hence, when we write $V_p = c_p\bar{V}_p$ for some $\bar{V}_p \in F(\mathcal{K}_p, X^T D_p N^*, -\beta, \langle,\rangle_M)$, we first know that $V_p = \sum_{j=1}^{t_p} \|w_{a_j}\|_2\|z_{a_j}\|_2\bar{w}_{a_j}\bar{z}_{a_j}^T$. We can find such $V_p$ because if $V_p = 0$, we could set all $w_{a_i} = 0$ and it will strictly decrease the objective. Note that $\|\bar{V}_p\|_{\mathcal{K}_{p,*}} = 1$, yielding $c_p = \|V_p\|_{\mathcal{K}_{p,*}} = \sum_{j=1}^{t_p} \|w_{a_j}\|_2\|z_{a_j}\|_2$, and $\bar{V}_p$ is a convex combination of $\bar{w}_{a_j}\bar{z}_{a_j}^T$. Now, let

$$\bar{V}_p = \sum_{j=1}^{t_p} \lambda_j \bar{w}_{a_j}\bar{z}_{a_j}^T,$$

where $\lambda_j$ s sum up to 1. We know that $\langle N^*, D_p X\bar{V}_p\rangle = -\beta$ and $N^*$ satisfy

$$\min_{A \in \mathcal{K}_p} \langle N^*, D_p X A\rangle \geq -\beta.$$

Hence, for all $\bar{w}_{a_j}\bar{z}_{a_j}^T$, we have that

$$\langle N^*, D_p X\bar{w}_{a_j}\bar{z}_{a_j}^T\rangle = -\beta,$$

for $j \in [t_p]$. This implies for all $i \in [m]$, we have that when $w_i \neq 0$,

$$\langle N^*, (X\bar{w}_i)_+\bar{z}_i^T\rangle = -\beta,$$

and same for $w'_i \neq 0$. Now we are ready to prove the claim. We first know that the regression error is the same, as we have the same model fit $Y^*$. The regularization error is given as

$$\beta\Big(\sum_{i=1}^{m_1} c_i + \sum_{i=1}^{m_2} d_i\Big) = -\langle N^*, Y^*\rangle.$$

Hence, the cost of the problem is the same for any choice of the conic combination, and $(\sqrt{c_i}\bar{w}_{a_i}, \sqrt{c_i}\bar{z}_{a_i})_{i=1}^{m_1} \oplus (\sqrt{d_i}\bar{w}'_{b_i}, \sqrt{d_i}\bar{z}'_{b_i})_{i=1}^{m_2}$ is optimal when $m_1 + m_2 \leq m$.

At last, suppose $m \geq nc + 1$. Note that the vectors $\{(Xw_i)_+z_i^T\}_{i=1}^m$ and $\{(Xw'_i)_+z_i'^T\}_{i=1}^m$ are matrices in $nc$-dimensional subspace. As any conic combination that sums up to $Y^*$ makes a solution, we can first prune both solutions to make them linearly independent, and then connect the two using the same idea introduced in Theorem D.1. $\square$

**Corollary F.1.** *(Corollary 4 of the paper) Consider the optimization problem in equation 11. Suppose $m \geq nc + 1$ and denote the optimal set of equation 11 as $\Theta^*(m)$. For any $\theta := (w_i, z_i)_{i=1}^m \in \mathbb{R}^{(d+c)m}$, there exists a continuous path from $\theta$ to any point $\theta^* \in \Theta^*(m)$ with nonincreasing loss.*

*Proof.* The proof is identical to that of Corollary 2. From Haeffele & Vidal (2017), we know that when $m \geq nc + 1$, the vector-valued training problem in equation 11 has no strict local minimum, i.e. all local minima are global. Now from any $\theta$, move to a local minimum using a path with nonincreasing loss, then the local minimum is global. As $\Theta^*(m)$ is connected, we know that we can arrive at any global minimum using a path with nonincreasing loss. $\square$

Finally, we extend our theory to parallel neural networks with depth 3. We have an optimal polytope characterization that states the first-layer weights have a finite set of fixed possible directions.

**Theorem F.4.** *(Theorem 3 of the paper) Consider the training problem*

$$\min_{m,\{W_{1i},w_{2i},\alpha_i\}_{i=1}^m} \frac{1}{3}\left(\sum_{i=1}^m \|W_{1i}\|_F^3 + \|w_{2i}\|_2^3 + |\alpha_i|^3\right)$$

*subject to*

$$\sum_{i=1}^m ((XW_{1i})_+w_{2i})_+\alpha_i = y.$$

Here, $W_{1i} \in \mathbb{R}^{d \times m_1}, w_{2i} \in \mathbb{R}^{m_1}$ and $\alpha_i \in \mathbb{R}$ for $i \in [m]$. Then, there are only finitely many possible values of the direction of the columns of $W_{1i}^*$. Moreover, when $y \neq 0$, the directions are determined by solving the dual problem

$$\max_{\|W_1\|_F \leq 1, \|w_2\|_2 \leq 1} |(\nu^*)^T((XW_1)_+ w_2)_+|$$

*Proof.* We can see the problem is equivalent to

$$\min_{m, \{\|W_{1i}\|_2 \leq 1, \|w_{2i}\|_2 \leq 1, \alpha_i\}_{i=1}^m} \sum_{i=1}^m |\alpha_i|,$$

subject to

$$\sum_{i=1}^m ((XW_{1i})_+ w_{2i})_+ \alpha_i = y,$$

from Wang et al. (2021a). Furthermore, when we write

$$\mathcal{Q}_X = \left\{ ((XW_1)_+ w_2)_+ \mid W_1 \in \mathbb{R}^{d \times m_1}, w_2 \in \mathbb{R}^{m_1}, \|W_1\|_F \leq 1, \|w_2\|_2 \leq 1 \right\},$$

the problem is equivalent to

$$\min \ t \geq 0 \quad \text{subject to} \quad y \in t \operatorname{Conv}(\mathcal{Q}_X \cup -\mathcal{Q}_X).$$

Now we find a cone-constrained linear expression of $((XW_1)_+ w_2)_+$. Let's denote $\mathcal{D} = \{D_i\}_{i=1}^{P_1}$ as the set of all possible arrangement patterns $\operatorname{diag}(1(Xh \geq 0))$ and $\mathcal{D}(m_1)$ denote all possible $\binom{P_1}{m_1}$ size $m_1$ tuples of elements in $\mathcal{D}$. Let's note $D_i(m_1) = (D_{i1}, D_{i2}, \cdots D_{im_1})$, where $D_{i1}, D_{i2}, \cdots D_{im_1} \in \mathcal{D}$. $i$ runs from 1 to $\binom{P_1}{m_1}$. Also, let's fix $s \in \{1, -1\}^{m_1}$.

Given $D_i(m_1), s$, we define the set $\mathcal{D}' = \{D_j'\}_{j=1}^{P_2(i)}$ as the set of all possible arrangements of $\operatorname{diag}(1(\tilde{X}h \geq 0))$, where $\tilde{X} = [D_{i1}X|D_{i2}X|\cdots|D_{im_1}X] \in \mathbb{R}^{n \times m_1 d}$.

When $D_i(m_1), s, D_j'$ are fixed, and $(W_1)_{\cdot,i}$ (which denote the $i$-th column of $W_1$), $w_{2i}$ are fixed in sets:

$$(2D_{ik} - I)X(W_1)_{\cdot,k} \geq 0, \quad s_k w_{2k} \geq 0 \quad \forall k \in [m_1],$$

$$(2D_j' - I)(\sum_{k=1}^{m_1} D_{ik}X(W_1)_{\cdot,k} w_{2k}) \geq 0,$$

the ReLU expression becomes

$$((XW_1)_+ W_2)_+ = \sum_{k=1}^{m_1} D_j' D_{ik} X(W_1)_{\cdot,k} w_{2k}.$$

In other words, when we denote $\mathcal{K}(D_i(m_1), s, D_j')$ as

$$\mathcal{K}(D_i(m_1), s, D_j') = \Big\{ (W_1, w_2) \mid (2D_{ik} - I)X(W_1)_{\cdot,k} \geq 0, \ s_k w_{2k} \geq 0 \quad \forall k \in [m_1],$$

$$(2D_j' - I)(\sum_{k=1}^{m_1} D_{ik}X(W_1)_{\cdot,k} w_{2k}) \geq 0 \Big\},$$

$$\mathcal{Q}_X = \bigcup_{i=1}^{\binom{P_1}{m_1}} \bigcup_{s \in \{-1,1\}^{m_1}} \bigcup_{j=1}^{P_2(i)} \Big\{ \sum_{k=1}^{m_1} D_j' D_{ik} X(W_1)_{\cdot,k} w_{2k} \mid (W_1, w_2) \in \mathcal{K}(D_i(m_1), s, D_j'),$$

$$\|W_1\|_F \leq 1, \|w_2\| \leq 1 \Big\}.$$

Now we consider the change of variables, where we write $(W_1)_{\cdot,k} w_{2k} = v_k \in \mathbb{R}^d$. The norm constraint becomes $\sum_{k=1}^{m_1} \|v_k\|_2 \leq 1$. To show this, we show that

$$\{((W_1)_{\cdot,k} w_{2k})_{k=1}^{m_1} \mid \|W_1\|_F \leq 1, \|w_2\|_2 \leq 1\} = \{(v_k)_{k=1}^{m_1} \mid \sum_{k=1}^{m_1} \|v_k\|_2 \leq 1\}.$$

First, for $\|W_1\|_F \leq 1, \|w_2\|_2 \leq 1$, assume the column weights are $a_1, a_2, \cdots a_{m_1}$. Then we have $a_1^2 + \cdots a_{m_1}^2 \leq 1$, $w_{21}^2 + w_{22}^2 + \cdots + w_{2m_1}^2 \leq 1$, and use Cauchy-Schwartz to see that $\sum_{k=1}^{m_1} a_k |w_{2k}| \leq 1$. To prove the latter, choose $W_1 = [v_1/\sqrt{\|v_1\|_2} | \cdots |v_{m_1}/\sqrt{\|v_{m_1}\|_2}], w_2 = [\sqrt{\|v_1\|_2}, \cdots, \sqrt{\|v_{m_1}\|_2}]^T$.

With change of variables, the cones are written as

$$\mathcal{K}_v(D_i(m_1), s, D_j') = \Big\{ (v_k)_{k=1}^{m_1} \mid (2D_{ik} - I)s_k X v_k \geq 0, \ \forall k \in [m_1],$$

$$(2D_j' - I) \sum_{k=1}^{m_1} D_{ik} X v_k \geq 0 \Big\},$$

which is a fixed cone in $\mathbb{R}^d$. Hence, $\mathcal{Q}_X$ can be rewritten as

$$\mathcal{Q}_X = \bigcup_{i=1}^{\binom{P_1}{m_1}} \bigcup_{s \in \{-1,1\}^{m_1}} \bigcup_{j=1}^{P_2(i)} \Big\{ \sum_{k=1}^{m_1} D_j' D_{ik} X v_k \mid (v_k)_{k=1}^{m_1} \in \mathcal{K}_v(D_i(m_1), s, D_j'), \sum_{k=1}^{m_1} \|v_k\|_2 \leq 1 \Big\}.$$

As a result, we have found a piecewise linear expression of $\mathcal{Q}_X$. When $y = 0$, we know that the optimal weights are all zeros. If not, we know that the problem

$$\text{minimize } t \geq 0 \quad \text{subject to} \quad y \in tC$$

has a dual variable $\nu^*$ that satisfies: if $y = t^*(\sum_{i=1}^m \lambda_i c_i)$ for some $c_i \in C$, all $c_i$ s are minimizers of $(\nu^*)^T c$ subject to $c \in C$. To see this fact, consider the supporting hyperplane on $y$. We can find a vector that satisfies $\langle \nu^*, y \rangle \leq \langle \nu^*, t^* c \rangle$ for all $c \in C$ and $\langle \nu^*, y \rangle / t^* \leq \langle \nu^*, c \rangle$ for all $c \in C$. Write $y = t^*(\sum_{i=1}^m \lambda_i c_i)$ and apply inner product with $\nu^*$ to see the wanted result. More specifically, we have that $\lambda_i \langle \nu^*, y \rangle \leq \lambda_i \langle \nu^*, t^* c_i \rangle$ for all $i \in [m]$, and add them to see that the inequalities are actually an equality, and $\langle \nu^*, y \rangle = \langle \nu^*, t^* c_i \rangle$ for all $i \in [m]$.

Hence, noting $C$ as $\text{Conv}(Q_X \cup -Q_X)$, there exists a dual variable $\nu^*$ where the optimal $(W_1, w_2)$ must lie in the set $\arg\max_{\|W_1\|_F \leq 1, \|w_2\|_2 \leq 1} |(\nu^*)^T((XW_1)_+ w_2)_+|$. For each constraint set

$$(v_k)_{k=1}^{m_1} \in \mathcal{K}_v(D_i(m_1), s, D_j'), \quad \sum_{k=1}^{m_1} \|v_k\|_2 \leq 1,$$

we are optimizing a linear function over this set (as the ReLU expression is a linear function of $(v_k)_{k=1}^{m_1}$). If there exists two different maximizers of the problem $(v_k)_{k=1}^{m_1}, (v_k')_{k=1}^{m_1}$, the average of the two will still be in the cone and satisfy the norm constraint strictly. Say $(v_k'')_{k=1}^{m_1}$ is the average of the two solutions - the cost function (which is either $(\nu^*)^T \sum_{k=1}^{m_1} D_j' D_{i1} X v_k$ or its negation) value will be the same, but $\sum_{k=1}^{m_1} \|v_k''\|_2 < 1$. Multiplying $1/\sum_{k=1}^{m_1} \|v_k''\|_2$ leads to a contradiction in the optimality. Hence, for fixed cone $\mathcal{K}_v(D_i(m_1), s, D_j')$, the optimal $(v_k)_{k=1}^{m_1}$ are fixed. As $v_k = (W_1)_{\cdot,k} w_{2k}$, the direction of the columns of $W_1$ are fixed to a finite set of values. $\square$

## G  ADDITIONAL DISCUSSIONS

In this section, we discuss the geometrical intuition of the dual optimum, non-unique solutions, and also explain why the assumption in Simsek et al. (2021) might not hold in our case.

The specific problem of interest is interpolating the dataset $\{(-\sqrt{3}, 1), (\sqrt{3}, 1)\}$ with a two-layer neural network with bias. We want to find a minimum-norm interpolator, where the cost function also includes regularizing the bias. We can write the problem as

$$\min_{m,\{w_i,\alpha_i\}_{i=1}^m} \frac{1}{2}\Big(\sum_{i=1}^m \|w_i\|_2^2 + |\alpha_i|^2\Big)$$

subject to

$$\sum_{i=1}^m (\bar{X}w_i)_+ \alpha_i = y.$$

Here, $\bar{X} = \begin{bmatrix} -\sqrt{3} & 1 \\ \sqrt{3} & 1 \end{bmatrix}$ and $y = [1, 1]^T$. The last column of $\bar{X}$ denotes the bias term.

The problem is equivalent to

$$\min_{m,\{w_i,\alpha_i\}_{i=1}^m} \sum_{i=1}^m |\alpha_i|,$$

subject to

$$\sum_{i=1}^m (\bar{X}w_i)_+ \alpha_i = y, \quad \|w_i\|_2 \leq 1.$$

See Pilanci & Ergen (2020) for a similar "scaling trick".

In other words, when we denote $\mathcal{Q}_{\bar{X}} = \{(\bar{X}u)_+ \mid \|u\|_2 \leq 1\}$, the problem becomes Pilanci & Ergen (2020)

$$\min t \geq 0 \quad \text{subject to} \quad y \in t\text{Conv}(\mathcal{Q}_{\bar{X}} \cup -\mathcal{Q}_{\bar{X}}).$$

Figure 9 shows the shape of $\mathcal{Q}_{\bar{X}}$ and its convex hull.

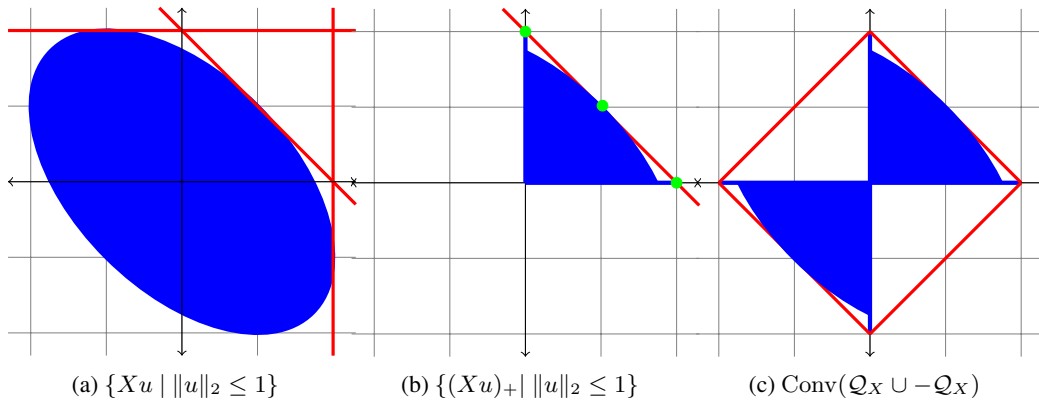

(a) $\{Xu \mid \|u\|_2 \leq 1\}$      (b) $\{(Xu)_+ \mid \|u\|_2 \leq 1\}$      (c) $\text{Conv}(\mathcal{Q}_X \cup -\mathcal{Q}_X)$

Figure 9: The shape of $\text{Conv}(\mathcal{Q}_X \cup -\mathcal{Q}_X)$. We can see that the line $x + y = 2$ is tangent to the set $\{Xu \mid \|u\|_2 \leq 1\}$, and meets with two points $(2, 0)$, $(0, 2)$ on the set $\mathcal{Q}_X$. Hence, $\text{Conv}(\mathcal{Q}_X \cup -\mathcal{Q}_X)$ is exactly the diamond $|x| + |y| \leq 2$.

One thing to notice is that in Figure 9b, the line $x + y = 2$ meets with $\mathcal{Q}_{\bar{X}}$ with three points, and the convex hull $\text{Conv}(\mathcal{Q}_{\bar{X}} \cup -\mathcal{Q}_{\bar{X}})$ is a diamond. The intuition of the dual variable is that it is the normal vector of a face where the optimal fit exists. In our case, $y = [1, 1]^T$ lies on the exact line $x + y = 2$. Hence the dual optimum is $\nu^* = [1, 1]^T$. We can also construct different minimum-norm interpolators by linear combinations of the three green points in Figure 9b: we can express $y$ by only using the middle point $(1, 1)$ - here, the interpolator becomes $y = 1$. We can use two points $(2, 0)$

and $(0, 2)$ to express $(1, 1)$ - here, we have another interpolator that has two breakpoints. We can use three points - where there will be infinitely many ways to express $(1, 1)$, that leads to a continuum of interpolators.

The assumption in Simsek et al. (2021) that there exists a unique model with zero loss and minimal width does not work here. We can adapt it to the regularized case, and assume that there exists a unique interpolator with minimal width and a solution to

$$\min_{m, \{w_i, \alpha_i\}_{i=1}^m} \frac{1}{2} \Big( \sum_{i=1}^m \|w_i\|_2^2 + |\alpha_i|^2 \Big)$$

subject to

$$\sum_{i=1}^m (\bar{X} w_i)_+ \alpha_i = y.$$

Here, $\bar{X} = [x \mid 1] \in \mathbb{R}^{n \times 2}$. Now choose $x = [-\sqrt{3}, \sqrt{3}]$ as before, but choose $y = [1/2, 3/2]$. Then, there exist two ways to express $y$ as a conic combination of $(2, 0)$, $(1, 1)$, and $(0, 2)$ with two points. As $y$ is not parallel to $[2, 0], [1, 1], [0, 2]$, we can see that $m^* = 2$ is minimal. Hence we don't have uniqueness of the smallest model in this case, and the results in Simsek et al. (2021) will not apply in general.

