# OpenReview forum: "Exploring The Loss Landscape Of Regularized Neural Networks Via Convex Duality"
_ICLR.cc/2025/Conference — ICLR 2025 Oral_

### Official Review · Reviewer_ki4t · 2024-10-29

**Soundness:** 3
**Presentation:** 3
**Contribution:** 3
**Rating:** 8
**Confidence:** 4

**Summary:**

This manuscript proposes a characterization of the topology of the global optima in the loss landscape, focusing on regularized two-layers neural networks with free skip-connections with a partial extension to deep neural networks. The authors provide a characterization of the optimal set in terms of the width of the hidden layer, which determines a so-called "staircase of connectivity" when such a width occurs in critical values and phase transitions. The authors study the uniqueness of the minimum-norm interpolator, highlighting necessary guarantees (such as the free ski connections, bias in the training problem and unidmiensional data). An experimental study integrates the theoretical findings.

**Strengths:**

The paper has original content, and bridges together several concepts proposed in the literature on the topic. The method of analysis is rigorous and it gives a solid contribution to the field.

**Weaknesses:**

The manuscript presents some unclear sentences (e.g. line 198-199). There is a clear math error at line 232 (the triangle inequality holds with the reverse inequality; to be candid, I am quite sure it is a typo) and some symbols are not defined at all, not even in the Appendix (e.g. the symbol P, that occurs very often through the entire manuscript).
There are many references to results listed in the Appendix; if relevant, I think it might be better to put them in the main manuscript.
An important reference to the characterization of loss landscapes over neural networks with regularization terms and/or skip connections is missing, also because it gives a theoretical hint on the low importance of skip connections [1].

[1] Bucarelli, M. S., D’Inverno, G. A., Bianchini, M., Scarselli, F., & Silvestri, F. (2024). A topological description of loss surfaces based on Betti Numbers. Neural Networks, 106465.

**Questions:**

Can the authors proofread again the manuscript to eliminate typos, unclear sentences and missing notation?
Can they make the presentation of the result more readable, including relevant results from the Appendix?
Can they integrate the relevant existing literature in the Related Work section?

---

> ### Author Response · Authors · 2024-11-21
>
> Thank you for your helpful comments on the paper. We have addressed them as the following:
>
> **1. Clarifications: Unclear sentences, math error (triangular inequality reversed), some undefined variables (e.g. P), results referring to the appendix, missing reference**
>
> Please see the general response. Some fixes that were made were:
>
>  1) Clearer statement on $\mathcal{P}^{\ast}_{\nu^{\ast}}$ and the intuition of optimal polytope
>  2) More accessible statement of main results
>  3) Bottom of Figure 2 explained
>  4) Clarification of the three interpolation problems solved (Section 3.3)
>
> We have fixed the math error and added notations to the main paper, defining all undefined variables there. Also, we tried to move most results from the appendix (as much as the page limit allows). At last, thank you for pointing out an important reference that was missing. We added the reference in the related works section.

---

> ### Comment · Reviewer_ki4t · 2024-11-24
>
> The authors have addressed every concern raised in my review, therefore I suggest this paper to be accepted at ICLR25.

---

### Official Review · Reviewer_eG7f · 2024-11-02

**Soundness:** 3
**Presentation:** 2
**Contribution:** 3
**Rating:** 8
**Confidence:** 3

**Summary:**

This paper studies the loss landscape of ReLU networks with $L_2$-regularization. The authors first study the canonical case of a two-layer  network with scalar output, and characterize the connectivity of the solution for different number of neurons. Then, the authors extend the results to a more general class of problems, including: minimal norm interpolation, vector-valued ReLU network, and parallel deep neural network.

**Strengths:**

1. This paper develops a general framework to characterize the global optimum of the regularized ReLU network via convex duality. From my understanding, the key contribution of this convex duality framework in Theorem 1 is that it allows one to characterize the "direction" of the weights separately in the regularized case, which is then useful for characterizing the global optimum. I believe this contribution is novel and solid.

2. I think the framework of characterizing the global optimal is quite general even though it is restricted to ReLU network. In particular, it do not require large over-parameterization, special scaling, or special data distributions. Thus, I believe the results can be applied to other more specific settings and is potentially useful for characterizing other properties besides the connectivity of the solutions.

**Weaknesses:**

1.Although I believe this paper has a solid contribution, I found there's a few part I don't understand the significance:
- I think I understand the contributions in section 3.1 and 3.2, however, I'm not sure about  In section 3.3, the authors showed that for a class of data set with dimension $=1$ that satisfies certain conditions, if the network do not have skip connection, then there are infinitely many minimal norm interpolators (which is a connected set ). I'm not sure the significance of these results, since (1) it is for a special construction of dataset. (2) it might be that those infinitely many minimal norm interpolators behave qualitatively almost the same, for example, the radius of the solution set is small. Could you discuss more on the significance of the results?

- In section 4, I understand the contribution of generalizing it to a vector-valued function. However, I'm not sure the significance of the results in Theorem 4. Since anyway you fixed all the other layers but only keep two consecutive layers, and technically I didn't see any difference from a two-layer network.  Could you discuss more on the significance of the results?

2. One main issue of the paper is the writing, especially the main part of the paper. I check the appendices, and it is much more readable. So I suggest the authors consider rearranging the content. To name a few issues\typos that confuse me when reading the main part:

    - Line 215: and 216, what is the definition of $\mathcal{S}_i$?
    - Line 223: what the definition of "optimal model fit", what is $u_i^*, v_i^*$, and why it is unique?
    - Line 232: the triangle inequality is reversed. Also could you be more specific about the discussion between Liine 229-232?
    - The statement of Proposition 1:  First, you use $v_{i,1}$ to denote the first entry of a vector, could you specify this?  Also, you define $s_k = \sum_{i=1}^k v_{n-i+1},$ but also require $||s_k|| =1, s_n = [0,1]^\top$, could you discuss the existence of such construction?
    - In equation (7), could you specify the dimension of the variables?

**Questions:**

1.  A general question is about the scope of the techniques in this paper. It seems that the techniques only apply to  two-layer ReLU networks, since the problem can be equivalently written as a convex problem with regularization. It is not applicable to other activation functions and seems hard to generalize to multi-layer cases. Thus, could you elaborate more on the universality of the techniques?

2. The results in this paper require the number of neurons $m \geq m_*$. As far as  I understand, $m_*$ is the minimal number of neurons needed to achieve the optimal model. I'm wondering what would happen if $m<m_*$? Also, in general, what is the scaling of $m_*$ depending on $n,d$?

3. The results in Theorem 2 consider the connectivity of the optimal solution set, which is equivalent to the connectivity of a path with $0$ perturbation. What about the case that allow $\epsilon$-pertubation along the path? Is the techniques still applicable?

---

> ### Author Response · Authors · 2024-11-21
>
> Thank you for your helpful comments on the paper. We have addressed them as the following:
>
> **1. Significance of the result in 3.3**
>
> There are two consequences in the result of Theorem 3.3, both are of theoretical interest. The consequences are:
> 1) We show that free skip connections may affect the qualitative behavior of the optimal solutions(e.g. uniqueness and sparsity)
> 2) The nonunique min-norm interpolators imply that we have multiple interpolators with the same norm but with different generalization performances, and their behavior may diverge significantly outside the training set.
>
> The first consequence is that in the literature, free skip connections have been imposed in the problem for a clearer derivation of optimal solutions ([1], [2], [3]). It has been believed in [2], [3] that not having free skip connections does not affect the optimal solution that much in practice. For example, in pg 2 of [3], it is stated that
>
> "This skip connection avoids some complications and allows better characterizing ˆfS but does not meaningfully change the behavior [3]".
>
> Our example shows that this statement is false, and not having free skip connections can make the problem have multiple min-norm interpolators and make the solution not sparse (i.e. the optimal interpolator may not be the sparsest). Thus, free skip connection may change the qualitative behavior of the solution (uniqueness & sparsity).
>
> Another significance is that these different minimum-norm interpolators, though having the same norm, have different generalization performances (because they behave differently outside the training set). Generalization bounds are generally associated with parameter norms, but here we see that interpolators with the same norm have are drastically different: as $x \rightarrow \infty$, the two function differences diverge, and have population loss (which is a qualitative difference).
>
> Moreover, the example demonstrates how convex duality can construct structured counterexamples motivated from the hidden convex geometry. The intuition for such construction is making the set $\mathcal{Q}_{X} = \{ReLU(Xu)\ | \ \Vert u \rVert_2 \leq 1\} $ meet with its supporting hyperplane with $n+1$ points.
>
> At last, we would like to emphasize that while the setting $d = 1$ could seem restrictive, understanding minimum-norm interpolators is an important problem that has been extensively discussed [1]~[7]. We believe it is an interesting problem whether we can construct counterexamples where the minimum-norm solutions differ the most, but we would like to leave it for future work.
>
> **2. Significance of the result in Theorem 4**
> The significance of Theorem 4 is not within the proof strategy, but the fact that we can describe a certain subset of the optimal set of a deep neural network in a precise manner. It is challenging to describe the optimal set of deep networks, because of its nonconvexity without a good theoretical structure to work with. Nevertheless, due to page limitations, we moved the Theorem to the appendix and mention such characterization is possible in the main paper.
>
> **3. Universality of the techniques**
> For the possible extensions to different activation functions, please see the general response.
>
> Also, we have extensions of the result to parellel three-layer neural networks (Theorem 3 of the main paper). Also, there are discussions of convex reformulations of parallel neural networks of depth L [11], and we believe that at least some of the results (e.g. the polytope characterization) could be extended to these settings.
>
> Overall, it is not true that these results could be extended to networks with arbitrary activation and architecture, but are not limited to only two-layer ReLU neural networks.

---

> ### Author Response · Authors · 2024-11-21
>
> **4. What happens if $m < m^{\ast}$? What is the scale of $m^{\ast}$?**
>
> When $m < m^{\ast}$, we cannot apply the methods discussed in the paper directly because there is no convex reformulation. We "could" start with a cardinality-constrained optimization problem,
> $$
> \min_{(u_i, v_i)_{i=1}^{P}} L(\sum_i D_iX(u_i - v_i), y) + \beta \sum_i \lVert u_i \rVert_2 + \lVert v_i \rVert_2 \quad \mathrm{subject} \ \ \mathrm{to} \sum_i 1(u_i \neq 0) + 1(v_i \neq 0) \leq m.
> $$
> but the critical issue is that the above problem is not convex, so it is not clear yet how to deal the case $m < m^{\ast}$.
>
> Precisely speaking, the scale of $m^{\ast}$ does not depend on $n, d$, but depend on the geometry of the dataset $(X, y)$. Specifically, consider $\mathcal{Q}_X = \{ReLU(Xu) \ | \ \lVert u \rVert_2 \leq 1\}$. Suppose we are solving the min-norm interpolation problem. Scale $y$ so that $ty \in \mathrm{Conv}(\mathcal{Q}_X \cup -\mathcal{Q}_X)$. The minimal number of points that are needed to express $ty$ as a sum of vectors from $\mathcal{Q}_X \cup -\mathcal{Q}_X$ becomes $m^{\ast}$. Hence, even for large $n, d$, depending on the dataset $m^{\ast}$ could be very small. It is an interesting open problem how $m^{\ast}$ would behave for random $(X, y)$, but unfortunately, I am unsure how things look like currently.
>
> **5. Can we still apply this technique for $\epsilon$ perturbations?**
> One thing we could do is using a convex set
> $(u_i, v_i)_{i=1}^{P}$ where $L(\sum_i D_iX(u_i - v_i), y) + \beta \sum_i \lVert u_i \rVert_2 + \lVert v_i \rVert_2 \leq p^{\ast} + \epsilon$
> where $p^{\ast}$ denotes the optimal value of the convex program. Here we will not have the optimal polytope characterization, so the connectivity behavior could differ and we may have that the set becomes connected even for lesser $m$. We think understanding how the connectivity behavior of the set changes as $\epsilon$ changes and understanding the (possible) phase transitional behavior is a very interesting direction of study, and also fits well with the initial description of mode connectivity.
>
> **6. Clarifications: $\mathcal{S}i$, optimal model fit, reversed triangular inequality, $\mathcal{P}^{\ast}_{\nu^{\ast}}$, the statement of Proposition 1, Dimensions in equation (7)**
>
> We no longer have complicated statements in the main paper. The optimal model fit and its uniqueness is stated before it is used. We fixed the triangular inequality and added more explanations for unclear statements, and specified the dimensions in equation (7). Please refer to the general response.
>
> **References**
> [1] Hanin, Boris. "Ridgeless Interpolation with Shallow ReLU Networks in $1 D $ is Nearest Neighbor Curvature Extrapolation and Provably Generalizes on Lipschitz Functions." arXiv preprint arXiv:2109.12960 (2021).
>
> [2] Boursier, Etienne, and Nicolas Flammarion. "Penalising the biases in norm regularisation enforces sparsity." Advances in Neural Information Processing Systems 36 (2023): 57795-57824.
>
> [3] Joshi, Nirmit, Gal Vardi, and Nathan Srebro. "Noisy interpolation learning with shallow univariate relu networks." arXiv preprint arXiv:2307.15396 (2023).
>
> [4] Ergen, Tolga, and Mert Pilanci. "Convex geometry of two-layer relu networks: Implicit autoencoding and interpretable models." International Conference on Artificial Intelligence and Statistics. PMLR, 2020.
>
> [5] Ongie, Greg, et al. "A function space view of bounded norm infinite width relu nets: The multivariate case." arXiv preprint arXiv:1910.01635 (2019).
>
> [6] Savarese, Pedro, et al. "How do infinite width bounded norm networks look in function space?." Conference on Learning Theory. PMLR, 2019.
>
> [7] Parhi, Rahul, and Robert D. Nowak. "Banach space representer theorems for neural networks and ridge splines." Journal of Machine Learning Research 22.43 (2021): 1-40.
>
> [8] Ergen, Tolga, and Mert Pilanci. "The Convex Landscape of Neural Networks: Characterizing Global Optima and Stationary Points via Lasso Models." arXiv preprint arXiv:2312.12657 (2023).
>
> [9] Bartan, Burak, and Mert Pilanci. "Neural spectrahedra and semidefinite lifts: Global convex optimization of polynomial activation neural networks in fully polynomial-time." arXiv preprint arXiv:2101.02429 (2021).
>
> [10] Ergen, Tolga, et al. "Globally optimal training of neural networks with threshold activation functions." arXiv preprint arXiv:2303.03382 (2023).
>
> [11] Ergen, Tolga, and Mert Pilanci. "Path regularization: A convexity and sparsity inducing regularization for parallel relu networks." Advances in Neural Information Processing Systems 36 (2024).
>
> [12] Garipov, Timur, et al. "Loss surfaces, mode connectivity, and fast ensembling of dnns." Advances in neural information processing systems 31 (2018).

---

> > ### Author Response · Authors · 2024-11-25
> >
> > Dear Reviewer eG7f,
> >
> > We believe that we have addressed your concerns in our responses and the revised manuscript. As the deadline is approaching, we would like to hear your feedback so we can respond before the discussion period ends. Please feel free to raise questions if you have other concerns. Thank you very much for your support, we sincerely appreciate that!
> >
> > Best regards, Authors

---

> ### Comment · Reviewer_eG7f · 2024-11-25
>
> I would like to thank the authors for addressing my questions and concerns in detail, I believe now I understand the contributions of the papers better after reading the authors' response.  I also checked the revised version of the paper, and it is much more readable comparing to the original version.
>
> Overall, I believe this paper has solid theoretical contributions on understanding the loss landscape of certain neural networks with L2-regularization, and would like to suggest a clear acceptance of this paper to ICLR2025.
>
> Finally, I have one last question: I think I'm able to understand the technical results in this paper, and it is based on a line of works [1-3]  that studies the convex formulation of neural networks as mentioned in related works part. To be honest, I'm not familiar with the technical results this line of work, so I can not accurately tell how "novel" the technical parts is. Thus, could the authors elaborate a bit more on the connections and differences between this paper and the line of works[1-3] (or maybe other works follows this line)?
>
> [1] Pilanci, Mert, and Tolga Ergen. "Neural networks are convex regularizers: Exact polynomial-time convex optimization formulations for two-layer networks."
> [2] Ergen, Tolga, and Mert Pilanci. "The Convex Landscape of Neural Networks: Characterizing Global Optima and Stationary Points via Lasso Models."
> [3] Mishkin, Aaron, and Mert Pilanci. "Optimal sets and solution paths of ReLU networks."
>
> Thank you and best regards,
> Reviewer eG7f

---

> > ### Author Response · Authors · 2024-11-25
> >
> > I can elaborate more on the technical novelty of this work.
> >
> > [1] first introduces that neural networks have convex reformulations, i.e. we have equivalent convex problems for two-layer ReLU neural networks with regularization, and there exists a solution mapping between the two. This is not our technical novelty, as it was done earlier. [3] is the main paper that we are motivated from: it describes the "solution set" (which is the set of optimal parameters of the neural network) of the neural network using this formulation, and characterizes as a polytope. [2] discusses that stationary points correspond to subsampled convex problems, which was discussed in different literature as well.
> >
> > Our main technical novelties are:
> >
> > 1) We associate the characterization in [3] with the dual optimum, leading to a different proof and a clear geometric intuition of the optimal solution set.
> > 2) The most novel proof is Lemma D.1. and Theorem D.1. in the proof. The proof is based on a nontrivial construction of a path from two different points in $\mathcal{P}^{*}(n+1)$ (the cardinality-constrained set), and we made a lot of effort to make this work. Such proof was never given in the above literature on convex networks, or in proving connectivity.
> > 3) Another technical novelty is section 3.3. The existence of non-unique minimum-norm interpolators has been discussed, but no work has discussed it using a geometric structure involving convex duality, that could further be applied to different problems.
> >
> > Also, we would like to highlight a conceptual novelty, which is that there has been discussions of the irreducible solution set and pruning a solution, but the existence of exact thresholds of topological change and their relation with the convex problem has never been discussed, both in the convex networks literature and nn theory literature (as far as I am concerned).
> >
> > Thank you very much for your time, constructive feedback and suggesting a clear acceptance.

---

> > > ### Comment · Reviewer_eG7f · 2024-11-26
> > >
> > > Thank you for the detailed explanation of the novelty of the paper. I believe I have a better understanding on the overall contributions of this paper after the rebuttal session and thus I decide to raise my score.
> > >
> > > Thank you and best regards, Reviewer eG7f

---

### Official Review · Reviewer_iEp8 · 2024-11-02

**Soundness:** 3
**Presentation:** 3
**Contribution:** 3
**Rating:** 8
**Confidence:** 3

**Summary:**

In this paper, the authors apply convex duality for two-layer ReLU networks to study mode connectivity and unique minimal-norm interpolator problems, while also working to generalize this framework. Specifically, the authors have:

* identified the staircase of connectivity that describes how connectivity evolves with width;
* constructed none-unique minimal-norm interpolator by breaking the uniqueness conditions;
* generalized the optimal polytope to the general cone-constrained group LASSO problem and applied it to more complicated architectures.

**Strengths:**

The authors apply the new technique of convex duality to problems of connectivity and minimal-norm interpolation, which have been studied previously using other methods. This approach yields both generalizations of existing results and new insights into these problems. Overall, I believe this paper is a strong demonstration of how convex duality can be leveraged in the theoretical study of machine learning. The abstract concepts are clarified through figures and examples.

**Weaknesses:**

I have some concerns with the presentation of this work. Specifically:

* If I understand correctly, the convex duality only applies to ReLU networks. This is not emphasized.
* I found Section 2 difficult to follow without prior knowledge of Pilanci & Ergen (2020). The relations between (1), (2), and (3) are mentioned but not explained (When do they have the same loss value? How do the solutions relate to each other?) Dimensions of $X$ and $y$ are not mentioned. $D_i$ is not explained.
* In Figure 1, is each red point truly a unique solution, or does it represent solutions equivalent under permutation (p-unique)? If they are p-unique solutions, readers may get the wrong impression.
* The lower half of Figure 2 is not explained.

**Questions:**

I wonder if the authors have any comment regarding the weaknesses.

---

> ### Author Response · Authors · 2024-11-21
>
> Thank you for your helpful comments on the paper. We have addressed them as the following:
>
> **1. Different activation functions**
>
> For the possible extensions to different activation functions, please see the general response.
>
> Though this is the case, this paper only discusses neural networks with ReLU activation - we now explicitly state that our scope is networks with ReLU activation in the Introduction and problem settings.
>
> **2. Presentation of Section 2**
> Thank you for pointing out the insufficient explanation on convex reformulations in Section 2. In the revised manuscript we added more explanation to Section 2. Specifically, we
>
> 1) clearly defined what $D_i, X, y$ and the shape of each matrix is in the problem setting and notations
> 2) We specifically stated that when $m \geq m^{\ast}$ for some critical threshold $m^{\ast} \leq n$, we have a convex reformulation.
> 3) We gave an exact notion what "equivalent convex problem" means, and gave how the two solutions are related.
> 4) We specifically stated that when $m \geq m^{\ast}$, strong duality holds and the dual problem also has the same optimal value as the primal problem.
>
> **3. Clarifications: Red points in Figure 1, the lower half of Figure 2**
>
> First, the red points in Figure 1 are actual optimal points in parameter space. Note that all permutations of each red point will be optimal: you could understand Figure 1 as all those permutations each plotted as red points, and as permutations are finite, we have finite number of red points.
>
> We clarified the lower half of Figure 2 in the revised manuscript.

---

> > ### Author Response · Authors · 2024-11-25
> >
> > Dear Reviewer iEp8,
> >
> > We believe that we have addressed your concerns in our responses and the revised manuscript. As the deadline is approaching, we would like to hear your feedback so we can respond before the discussion period ends. Please feel free to raise questions if you have other concerns. Thank you very much for your support, we sincerely appreciate that!
> >
> > Best regards, Authors

---

> > > ### Comment · Reviewer_iEp8 · 2024-11-25
> > >
> > > Dear authors,
> > >
> > > Thank you for the reply and the revision. My concerns are addressed. I have raised my score.
> > >
> > > Best regards,
> > > Reviewer iEp8

---

### Official Review · Reviewer_Rcks · 2024-11-02

**Soundness:** 3
**Presentation:** 3
**Contribution:** 3
**Rating:** 8
**Confidence:** 2

**Summary:**

In this work the authors analyze multiple aspects of the loss landscape of regularized two-layer neural networks with scalar output, including the structure of stationary points, the connectivity of optimal solutions and the non uniqueness of optimal solutions. The main proof strategy is to translate the problem into an equivalent convex problem and characterize its solution set through its dual form.
The authors show that the topology of the global optima goes through a phase transition as a function of the hidden layer width, which they term the staircase of connectivity.
This result is extended later to networks with vector-valued outputs, and parallel deep networks of depth 3.

**Strengths:**

- I found the "staircase of connectivity" very insightful, particularly how the connectivity properties of the optimal solutions are connected to critical widths $m^*$ and $M^*$. This finding explains how increasing the number of neurons affects the connectedness of optimal sets, and makes the observation of mode connectivity [Garipov et al. 2018] more precise.
- The paper generalizes its findings also to vector-valued networks and deep networks with skip connection, which provides a broader framework that can be applied across different architectures.

**Weaknesses:**

- I found the theoretical results, for instance on the staircase of connectivity, hard to interpret in practice and would benefit from more accessible explanations. While the toy example in Example 1 illustrates the concept, the absence of labels in Figure 2, as well as the notation-heavy formulation makes it difficult for readers to grasp the results intuitively.
- Although the toy examples are helpful, the paper lacks actual empirical validation of the theoretic results. I think it would add credibility to this work, if the staircase of connectivity concept would also be tested on actual neural network architectures trained on real data. It would be interesting to see how these results scale with different data distributions and larger models.
- Overall I found the work quite difficult to read due to the dense mathematical formalism. I also feel like the section on notations should not be in the appendix, but should - at least in a shortened version - be included in the main paper.

**Questions:**

1. Is there a way to bound or estimate the critical widths $m^*$ and $M^*$ in practice, for instance on real datasets?
2. In line 186: What does $h$ refer to in $\text{diag}[1 (Xh \geq 0)]$ ?
3. It is not very clear to me what lines 225-226 mean. Could you perhaps rephrase it? (That $\mathcal{P}^*_{\nu^*}$ does depend on $\nu^*$, but that the specific choice of it does not matter.)
4. Figure 2 bottom: The axis labels are missing and it is not very clear to me what the red and blue lines are supposed to represent.
5. In line 351 the author mention three interpolation problems of interest, but only discuss one problem on the minimum-norm interpolation problem. What are the other two interpolation problems and can you also extend your results to these problems?
6. The paper describes a path of nonincreasing loss that connects local to global minima. Could this insight be incorporated into practical training algorithms, such as initializing weights or guiding optimizers in large-scale training?

---

> ### Author Response · Authors · 2024-11-21
>
> Thank you for your helpful comments on the paper. We have addressed them as the following:
>
> **1. Accessible explanations of theoretical results**
>
> We agree that the paper would benefit from more intuition and have uploaded a major revision improving the presentation. Please see the general reponse.
>
> **2. Empirical validation of theoretical results with real-world datasets**
>
> Unfortunately, it is extremely hard to verify our findings with real-world datasets and large models. There are two reasons:
>
> 1) Computing the critical widths $m^{\ast}$, $M^{\ast}$ is challenging. Though we present how we can compute the critical widths in practice (see Remark D.1. or answer 4 of the rebuttal), the algorithm is exponential in the number of data (the complexity is actually $O((n/d)^{dn})$, though we have an algorithm to upper bound $m^{\ast}$ by first computing a solution to the convex training problem and pruning the solution(as introduced in [6]). A naive bound is $m^{\ast} \leq n$). As finding $m^{\ast}$ is equivalent to finding sparsest points within a convex set, we believe that the problem could be NP-hard (e.g. [1] shows that finding a sparsest point in an ellipsoid is NP-Hard), and there might not be any efficient (or practical) algorithm to find these widths.
>
> 2) Obtaining the optimal solution set is challenging for real-world dataset and large models. This is because even for two-layer networks, training neural networks to global optimality can be NP-Hard in certain scenarios ([2], [3]). Even in cases where we can train networks to global optimality, e.g. when the network is sufficiently wide enough so that it does not have spurious local minima, we could use the convex reformulations to obtain the solution set, but one problem is that the number of variables(which is in scale $O((n/d)^d)$) for the problem will be too large and finding the exact optimal solution set is impractical.
>
> Nevertheless, we think that this weakness demonstrates the theoretical strength of our results, as our results show how theoretical analysis can lead to scientific statements concerning neural network training where empirical experiments could never grasp (due to computational complexity).
>
> **3. Notations to the main paper**
>
> We summarized and moved the notation part to the main paper in the revised manuscript. Please refer to the general response.
>
> **4. A way to compute or estimate the critical widths $m^{\ast}$, $M^{\ast}$**
>
> There exists an algorithm to exactly compute the critical widths $m^{\ast}, M^{\ast}$. The algorithm is discussed in Remark D.1, and we have added that there exists an algorithm that can compute $m^{\ast}$ and one can refer to the appendix in the main paper.
>
> **5. How "the descent path to arbitrary global minimum" can motivate practical algorithms**
>
> The construction of a path that connects global minima can motivate new algorithms that search the optimal solution space. More specifically, the "meta-algorithm" that was used to prove that $\mathcal{P}^{*}(n+1)$ is connected could be directly used to construct a path between two points with low training loss.
>
> Also, we have a practical implication of the Theorem, which says that the loss landscape is benign and even though the landscape is nonconvex, naive methods may find good parameters well.
>
> **6. Clarifications: $\mathrm{Diag}(1(Xh \geq 0))$, $\mathcal{P}^{\ast}_{\nu^{\ast}}$, Figure 2 bottom, the three interpolation problems**
>
> We clarified these in the revised manuscript. Specifically, $\mathrm{Diag}(1(Xh \geq 0))$ is defined in the notations, sentences explaining $\mathcal{P}^{\ast}_{\nu^{\ast}}$ and the three interpolation problems were rewritten, and the bottom of Figure 2 is explained with labels.
>
> **References**
> [1] Natarajan, Balas Kausik. "Sparse approximate solutions to linear systems." SIAM journal on computing 24.2 (1995): 227-234.
>
> [2] Boob, Digvijay, Santanu S. Dey, and Guanghui Lan. "Complexity of training ReLU neural network." Discrete Optimization 44 (2022): 100620.
>
> [3] Froese, Vincent, and Christoph Hertrich. "Training neural networks is np-hard in fixed dimension." Advances in Neural Information Processing Systems 36 (2024).
>
> [4] Cover, Thomas M. "Geometrical and statistical properties of systems of linear inequalities with applications in pattern recognition." IEEE transactions on electronic computers 3 (1965): 326-334.
>
> [5] Haeffele, Benjamin D., and René Vidal. "Global optimality in tensor factorization, deep learning, and beyond." arXiv preprint arXiv:1506.07540 (2015).
>
> [6] Mishkin, Aaron, and Mert Pilanci. "Optimal sets and solution paths of ReLU networks." International Conference on Machine Learning. PMLR, 2023.

---

> > ### Author Response · Authors · 2024-11-25
> >
> > Dear Reviewer Rcks,
> >
> > We believe that we have addressed your concerns in our responses and the revised manuscript. As the deadline is approaching, we would like to hear your feedback so we can respond before the discussion period ends. Please feel free to raise questions if you have other concerns. Thank you very much for your support, we sincerely appreciate that!
> >
> > Best regards, Authors

---

> > > ### Comment · Reviewer_Rcks · 2024-11-25
> > >
> > > I would like to thank the authors for addressing my concerns and questions, particularly also explaining why it is challenging to empirically validate the staircase of connectivity.
> > >
> > > I think that the revised manuscript is a good theoretical paper and would suggest the acceptance of this paper to ICLR25.

---

### Official Review · Reviewer_UbdA · 2024-11-03

**Soundness:** 3
**Presentation:** 4
**Contribution:** 3
**Rating:** 8
**Confidence:** 5

**Summary:**

The authors present a deep and novel analysis of the loss landscape and a solution in the context of regularized neural networks. They also show that the topology of the global optima undergoes a phase transition as the width of the network changes.

**Strengths:**

The paper is well-written, easy to follow, and its results are clear and novel. The theoretical results stand out for their depth and clarity, as do the empirical results. The support of images is quite helpful when reading through some mathematical arguments or proofs of theoretical results.

**Weaknesses:**

Perhaps a deeper discussion on the topological implications of their results would be beneficial.

**Questions:**

No questions.

---

> ### Author Response · Authors · 2024-11-21
>
> We are very glad to see that the reviewer appreciated our work!
>
> **1. A deeper discussion on the topological implications of the results**
>
> The main topological implication of our theorem is that as the width increases, the topology of the solution set is complicated at first, but becomes simpler later on.
>
> An important implication of our theorems is that all sublevel sets are connected when the width is sufficiently large ($n+1$) Hence, our result on connectivity is a non-asymptotic improvement of [2], enabling us to discuss the topology of sublevel sets not only in the case $m \rightarrow \infty$.
>
> Another interesting direction to study the topology of the optimal set of neural networks is discussing its homology (i.e. we want to find out if the optimal set looks like a simple sphere, or something like a torus with a hole, etc.). We believe that our framework could be used to cound the number of holes that an optimal set of a neural network has, once we understand the homology of the cadinality-constrained set $\mathcal{P}^{*}(m)$ in the paper. We believe such characterization could make our understanding of neural network loss landscapes more complete, by excluding certain shapes as impossible or finding out that the "bottom of the loss landscape" looks like a certain shape.
>
> [1] Haeffele, Benjamin D., and René Vidal. "Global optimality in tensor factorization, deep learning, and beyond." arXiv preprint arXiv:1506.07540 (2015).
>
> [2] Freeman, C. Daniel, and Joan Bruna. "Topology and geometry of half-rectified network optimization." arXiv preprint arXiv:1611.01540 (2016).

---

> > ### Author Response · Authors · 2024-11-25
> >
> > Dear Reviewer UbdA,
> >
> > Thank you for appreciating our work. As the deadline is approaching, we would like to hear your feedback about the revised manuscript so we can respond before the discussion period ends. Please don't hesitate to raise questions if you have other concerns. Thank you very much for your support, we sincerely appreciate that!
> >
> > Best regards, Authors

---

> > > ### Comment · Reviewer_UbdA · 2024-11-25
> > > **Response**
> > >
> > > Thank you for the reply! My concerns are addressed.

---

### Author Response · Authors · 2024-11-21
**General Response**

We thank all reviewers for their time and thoughtful reviews. The manuscript has improved a lot thanks to the sincere reviews.

**Presentation**

Many reviewers suggested improving the overall presentation of the paper, making it more accessible and readible. We have a revised manuscript that we believe that we have improved the presentation by:

1) Clearly defining the problem of interest and notations in the main paper
2) Making the main results (Theorem 1, 2, Proposition 3) more accessible by removing mathematically heavy notations and explaining the main concepts verbally, deferring a rigorous treatment to the appendix
3) Clarifying any undefined variables.
4) Moving "most" relevant results from the appendix to the main paper. If the result was mathematically very complicated and is a direct extension/application of existing results (e.g. Prop F.5, Thm F.2, Prop F.1, F.2, F.3), we left the results in the appendix.
5) We tried to clarify every unclear sentences / sections / figures (especially the second part of Figure 2) that the reviewers pointed out.

Our submission is updated with the revised version.

**Different Activations**

Another common question was the scope of the analysis, in particular for different activation functions.

We would like to note that convex reformulations exist for many different activation functions: for example they exist for piecewise linear [1], polynomial activations [2], and threshold activations [3]. Our result are easily extended to piecewise linear activations, and similar results could be developed for different activations. Our results may not be trivially extendible to other activations such as threshold or polynomial, as their convex reformulations of  have a different form compared to ReLU.

We sincerely appreciate the reviewers' insightful questions and thoughtful feedback. We look forward to a meaningful discussion during the discussion phase!

**References**

[1] Ergen, Tolga, and Mert Pilanci. "The Convex Landscape of Neural Networks: Characterizing Global Optima and Stationary Points via Lasso Models." arXiv preprint arXiv:2312.12657 (2023).

[2] Bartan, Burak, and Mert Pilanci. "Neural spectrahedra and semidefinite lifts: Global convex optimization of polynomial activation neural networks in fully polynomial-time." arXiv preprint arXiv:2101.02429 (2021).

[3] Ergen, Tolga, et al. "Globally optimal training of neural networks with threshold activation functions." arXiv preprint arXiv:2303.03382 (2023).

---

### Meta-Review · Area_Chair_SqQk · 2024-12-16

**Metareview:**

This paper studies the loss landscape of regularized ReLU networks, focusing on the structure of stationary points, the connectivity of optimal solutions and the non uniqueness of optimal solutions. The authors start by considering a two-layer network with scalar output, and characterize the connectivity of the solution for different sizes of the network. Then, they consider extensions to minimal norm interpolation, vector-valued networks, and deep neural networks.

The reviewers found the results insightful (especially the 'staircase of connectivity' phenomenon) and well-integrated in the related literature. The proposed framework based on convex duality is rather general: it does not require large over-parameterization or specific scalings, which is a strong point. At the technical level, the origin of this framework is to be found in earlier work by Pilanci et al., which limits a bit the technical novelty. However, significant effort is required to tackle the settings considered here. All reviewers are positive about the paper, and so am I. Thus, I recommend it to be accepted.

**Additional Comments On Reviewer Discussion:**

All the points raised by the reviewers have been resolved during the rebuttal and discussion, so the opinion on the paper is uniformly positive.

---

### Decision · Program_Chairs · 2025-01-22

Accept (Oral)